# Ice fabrics in two-dimensional flows: beyond pure and simple shear

Daniel H. Richards[1], Samuel S. Pegler[1], and Sandra Piazolo[2]

[1]School of Mathematics, University of Leeds, United Kingdom
[2]School of Earth and Environment, University of Leeds, United Kingdom

**Correspondence:** Daniel H. Richards (d.h.richards@leeds.ac.uk)

**Abstract.** Ice fabrics - the distribution of crystal orientations in a polycrystal - are key for understanding and predicting ice flow dynamics. Despite their importance, the characteristics and evolution of fabrics produced outside of the deformation regimes of pure and simple shear flow has largely been neglected, yet they are a common occurrence within ice sheets. Here, we use a recently developed numerical model (SpecCAF) to classify all fabrics produced over a continuous spectrum of incompressible two-dimensional deformation regimes and temperatures. The model has been shown to accurately predict ice fabrics produced in experiments, where the ice has been deformed in either uniaxial compression or simple shear. Here we use the model to reveal fabrics produced in regimes intermediate to pure and simple shear, as well as those that are more rotational than simple shear. We find that intermediate deformation regimes between pure and simple shear result in a smooth transition between a fabric characterised by a girdle and a secondary cluster pattern. Highly rotational deformation regimes are revealed to produce a weak girdle fabric. Furthermore, we provide regime diagrams to help constrain deformation conditions of measured ice fabrics. We also obtain predictions for the strain scales over which fabric evolution takes place at any given temperature. The use of our model in large-scale ice flow models and for interpreting fabrics observed in ice cores and seismic anisotropy provides new tools supporting the community in predicting and interpreting ice flow in a changing climate.

## 1 Introduction

Mass loss from ice sheets is set to be the main contributor to sea-level rise this century (e.g. Shepherd et al., 2018). Reliably predicting sea-level rise depends on accurately modelling ice flow. One of the most important controls on ice-flow dynamics is the ice fabric, i.e. the collective distribution of crystal orientations within a given polycrystal. Strong alignment of the crystal orientations may cause the strain-rate response to an applied stress to vary by a factor of 9 in different directions (e.g. Pimienta and Duval, 1987). Hence, understanding the fabrics present in any flowing ice sheet, i.e. Antarctica or Greenland, is important for predicting ice-sheet flow and, in turn, the loss of ice over time.

To date, the analysis and discussion of ice fabrics has focused primarily on those formed under the specific deformation regimes of uniaxial compression and pure shear (both irrotational deformations), as well as simple shear. However, these regimes represent isolated points in a parameter space of deformation regimes that occur in nature. Kamb (1972) conducted experiments to produce fabrics at for intermediate deformations between uniaxial compression and pure or simple shear and produced illustrative pole figures across this space. He found that for these intermediate deformations, the fabric pattern produced exhibits a continuous transition between what is seen for the known end points. This transition is measured using the

*stress character*, the ratio of maximum and minimum in stress difference between principal axes. Kamb (1972) also examines the steady state characteristics of the fabric, finding it is primarily dependent on total strain and weakly dependent on stress. However the experiments are limited to strains of around $0.5$ and high temperatures ($-5$ to $0°$ C) which limits the application of this work to conditions seen in ice sheets.

The objective of the present paper is to use the fabric evolution model SpecCAF (Richards et al., 2021) to take a step away from the isolated conditions of irrotational deformation and simple shear where the model has been validated, and explore the continuous space of deformation regimes lying between these cases, and extrapolate beyond to deformation regimes more rotational than simple shear. One way to model ice flow is to simplify it to the two-dimensional $x$-$z$ plane (along the flow direction and the vertical, e.g Martín et al., 2009). This is done in order to understand vertical variation of the ice flow and compare to ice core profiles. Analysis of this flow shows that it commonly resides outside the regimes of pure and simple shear (we will illustrate this further in section 2.2.1 below). Therefore, an exploration of the fabrics arising in this generalised situation is a necessary step towards improving our understanding of ice fabrics: both to aid the interpretation of fabrics measured from ice cores and to predict future ice flow taking ice fabric effects into account.

To summarise, in this paper we seek to address a number of open questions. First, what fabrics are produced under any given (incompressible) two-dimensional deformation regime? Second, how do these fabrics change over the space of increasing vorticity and temperature, and can we use this information to aid in interpreting ice cores? Third, how do fabrics evolve at very high strains which have remained inaccessible to laboratory experiments, and at what strain does the fabric reach a steady state?

We address these questions by making use of a new continuum model SpecCAF (Richards et al., 2021), which is the first fabric evolution model to accurately predict ice fabrics produced in laboratory experiments, and is computationally efficient enough to be incorporated into large-scale ice-sheet models.

## 2  Background

### 2.1  Fabric Development

#### 2.1.1  Processes governing fabric development

The distribution of crystallographic orientations within a polycrystal is called the fabric or crystallographic preferred orientation (CPO). The distribution of the $c$-axes is the dominant control on the mechanical properties (such as viscous anisotropy) of ice (Duval, 1981). Ice deforms and flows primarily through dislocation glide, which occurs almost exclusively along the basal-plane. The orientation of the basal-plane can be described by its normal vector, the $c$-axis. Both the intensity and pattern of the fabric produced is dependent on the conditions of deformation, which will influence the relative activity of different mechanisms. As ice deforms, the fabric evolves through dislocation glide along the basal plane, which causes $c$-axes to rotate (Steinemann, 1958; Hondoh, 2000), rigid-body rotation, which simply rotates grains around the rotation axis, and recrystallization processes, which rearrange the grain boundary network.

There are two main recrystallization processes that affect the ice fabric. The first is *migration recrystallization*, which can include a combination of strain-induced grain boundary migration and nucleation (Doherty et al., 1997). Grain-boundary migration in a deforming crystalline material is mainly driven by differences in stored strain energy i.e. energy related to dislocation density either side of a grain boundary (e.g. Gottstein and Shvindlerman, 1999; Humphreys and Hatherly, 2004). Hence, in the case of such strain-induced migration recrystallization, the less strained grain grows at the expense of the more strained grain resulting in an overall decrease in the strain energy of the system (Drury et al., 1985; Drury and Urai, 1990). The dislocation density accumulated within a certain grain is primarily a response to its orientation relative to the deviatoric stress axes. For example, a grain favourably oriented for basal slip will accumulate fewer dislocations than a grain that is oriented unfavourably. Consequently, depending on the deformation regime, grains of certain orientation will grow at the expense of grains of less favourable orientations. Grains may also nucleate spontaneously in areas of high dislocation density and then grow if they are favourably orientated (Doherty et al., 1997). As a result, the effect of migration recrystallization is to produce $c$-axes clustered towards certain orientations in a polycrystal. It should be noted that, while the local stress axes are influenced by the environment around the respective grain (e.g. Grennerat et al., 2012; Piazolo et al., 2015), grains with $c$-axis oriented less favourably for slip relative to the far field stress axes will statistically have higher stored strain energy.

The second recrystallization process is *rotational recrystallization*. This occurs when dislocations recover into subgrain boundaries which, with increasing strain, will develop into grains (Drury et al., 1985). These dislocations tend to be concentrated closer to grain boundaries due to stress heterogeneity, as observed in shallow and deep polar ice (Kipfstuhl et al., 2006, 2009), and which can be thought of as an ice grain having a stressed outer 'mantle' and a less stressed inner 'core' (Faria et al., 2009). Therefore, new grains developing from subgrains will tend to occur near grain boundaries. The orientation of these new grains is similar to, but slightly different to, the parent grain. With increasing strain, the difference in orientation tends to increase (Halfpenny et al., 2006). This randomisation of orientations acts to diffuse concentrations in the fabric (Alley, 1992).

### 2.1.2 Observed fabrics

Ice fabrics can be observed through laboratory experiments, through ice cores from real-world locations, and, more recently, inferred through radar and seismic measurements. In the laboratory, the majority of experiments are performed by compressing a block of ice, resulting in an irrotational deformation: either pure shear if the block is confined in one direction or or uniaxial compression otherwise (e.g. Jacka and Maccagnan, 1984; Jacka and Li, 2000; Craw et al., 2018; Fan et al., 2020; Piazolo et al., 2013). The other oft-studied case is simple shear (Journaux et al., 2019; Qi et al., 2019). Laboratory experiments provide detailed fabric measurements in known conditions. However, experiments are mostly limited to single deformation regimes, as well as to strains of around $0.4$ for uniaxial compression (Fan et al., 2020) and 2 for direct simple shear (Qi et al., 2019).

Experiments have been performed for deformations intermediate to pure and simple shear, at temperatures close to the melting point of ice (Duval, 1981; Li et al., 1996; Budd et al., 2013). Fabrics from Duval (1981) combining uniaxial compression and simple shear show a broad cluster with 3 or 4 maxima inside it. Budd et al. (2013) shows, for an experiment with mostly simple shear combined with some pure shear and at an equivalent strain to that used later in this paper of $0.75$, the merging of a double cluster (from pure shear) and a single-maximum (from simple shear). There are no experiments exploring deformation

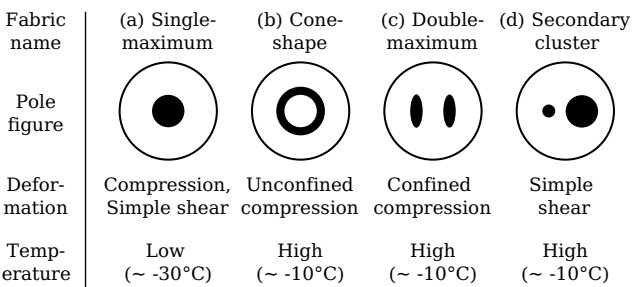

| Fabric name | (a) Single-maximum | (b) Cone-shape | (c) Double-maximum | (d) Secondary cluster |
|---|---|---|---|---|
| Pole figure | | | | |
| Deformation | Compression, Simple shear | Unconfined compression | Confined compression | Simple shear |
| Temperature | Low (~ -30°C) | High (~ -10°C) | High (~ -10°C) | High (~ -10°C) |

**Figure 1.** Illustration showing common fabrics or fabrics which develop in ice, illustrated by their pole figures, as well as the deformation regime and temperature they typically occur at. The pole figures show the distribution of $c$-axis orientations, with the compression axis at the centre. (a) shows a single maximum fabric, produced in uniaxial compression or simple shear at low temperatures (Qi et al., 2019). (b) shows a girdle fabric, produced in uniaxial compression at higher temperatures, when grain-boundary migration is active (Paterson, 1999). This can also be considered a girdle fabric when the cone-angle approaches $90°$. (c) shows a double-maximum fabric produced in pure shear (Budd et al., 2013). (d) shows a single-maximum with a secondary cluster, produced in simple shear at higher temperatures. (Kamb, 1972; Qi et al., 2019). Note: this simple shear deformation is rotated $45°$ to keep the principal deformation axes constant relative to the other figures.

regimes more rotational than simple shear. As a preliminary motivation in section 2.2.2, we will show that deformation regimes that lie in between pure and simple shear, as well as those that are more rotational than simple shear, occur widely in natural ice-sheet flows along both horizontal and vertical cross-sections.

Fabrics can also be analysed by taking ice cores in ice sheets. A detailed understanding of the fabrics produced over possible deformations and temperatures enables us to interpret the deformation regime and temperature history of ice cores. Initial studies of ice cores have concentrated on ice domes or divides (Gow, 1961; Holtzscherer et al., 1954; Johnsen et al., 1995). These locations are deliberately chosen because they have minimal deformation, and thus act as a good proxy for past climate data. At the centre of a dome, the ice will deform vertically in uniaxial compression, producing either a single-maximum or a girdle fabric (Fig. 1). Recently, ice cores have become available in locations with more complex deformation regime histories (Stoll et al., 2018; Treverrow et al., 2010). Stoll et al. (2018) show examples of a variety of fabric shapes such as girdles and single-maximum fabrics orientated in different directions as well as relatively faster fabric development with depth compared to ice cores at domes. Fabrics can also be measured from boreholes using sonic and optical techniques (Gusmeroli et al., 2012; Kluskiewicz et al., 2017).

Recently, data from radar and seismics has also been used to infer fabric properties (Matsuoka et al., 2003; Fujita et al., 2006; Booth et al., 2020). These methods can capture natural ice fabrics without expensive drilling, allowing data to be collected at more active locations such as ice streams (Jordan et al., 2020).

### 2.1.3 Fabric development

In experiments and observations, a number of common fabric patterns occur (Figure 1). Fabrics are commonly visualised by pole figures, showing a hemisphere where each point represents a possible orientation. As the $c$-axes are antipodally symmetric,

a hemisphere is sufficient to show all possible orientations. The colour then indicates concentrations of orientation at the direction. The mechanisms of basal-slip deformation, rigid-body rotation, migration and rotational recrystallization act in different ways and with different magnitudes depending on the deformation regime and temperature. For uniaxial compression, at low temperatures ($T \approx -30°$ C) basal-slip deformation dominates and this causes $c$-axes to rotate towards the axis of compression, producing the single-maximum pattern (Fig. 1a). At high temperatures, migration recrystallization is also active. This process acts to consume grains orientated towards the compression axis and, on its own, grows grains orientated in a ring $45°$ away from the compression axis (the orientation easiest for basal slip and hence likely to be with the least dislocations). Therefore, the balance of basal-slip deformation and migration recrystallization produces a girdle pattern, with an angle always $< 45°$ due to the interaction between the two processes (Fig. 1b). In pure shear, the grains produced by migration recrystallization instead form two clusters at $45°$ and hence a double-maximum fabric is produced (1c), as observed in experiments (Budd et al., 2013).

Recent experiments in simple shear produced either a single-maximum at low temperatures, or a single-maximum with an offset secondary cluster (Fig. 1d) at intermediate strains and high temperatures (Qi et al., 2019; Journaux et al., 2019). This pattern is similar to a double-maximum but the presence of vorticity in simple shear causes an imbalance in cluster strengths. For the stronger, primary cluster the vorticity acts to move $c$-axes in the opposite direction to the basal-slip deformation, resulting in a stable position. For the weaker, secondary cluster the vorticity and basal-slip deformation both rotate $c$-axes away, towards the compression axis. This results in the imbalance in cluster strengths illustrated in Fig. 1d.

## 2.2 Classifying flow regimes

### 2.2.1 General deformation regimes

There exists a significant variety of deformation regimes in the natural world. One way to classify a deformation regime is by the vorticity number (Passchier, 1991), which measures the ratio of vorticity magnitude to strain-rate magnitude:

$$\mathcal{W} = \sqrt{\frac{W_{ij}W_{ij}}{D_{ij}D_{ij}}}, \tag{1}$$

where $\mathbf{W} = \frac{1}{2}(\nabla \boldsymbol{u} - \nabla \boldsymbol{u}^T)$ is the anti-symmetric part of the velocity gradient (the spin-rate tensor) and $\mathbf{D} = \frac{1}{2}(\nabla \boldsymbol{u} + \nabla \boldsymbol{u}^T)$ is the symmetric part of the velocity gradient (the strain-rate tensor).

As a note for people unfamiliar, in Eq. (1) we have used both summation notation $W_{ij}$ and vector notation $\mathbf{W}$. $W_{ij}$ is a 2nd-rank tensor (shown by the number of indices) and the operation $W_{ij}W_{ij}$, indicating summation over the repeated indices, is the tensor inner product $\mathbf{W} : \mathbf{W}$.

Figure 2 illustrates the flow regimes associated with different vorticity numbers $\mathcal{W}$. The vorticity number is 0 for pure shear or uniaxial compression, 1 for simple shear and $\infty$ for rigid-body rotation. Ice in the natural world will experience deformation regimes with vorticity numbers from 0 to $\infty$. However, the most extensive analysis to date has focused on the specific cases of $\mathcal{W} = 0$ and $\mathcal{W} = 1$ due to the difficulty of producing other deformation regimes in experiments. Pure shear and simple shear also tend to dominate discussions regarding the interpretation of fabrics in ice-sheet flow. In the late 1990s and early 2000s it was recognised in the geological community that flow in rocks cannot be approximated by the isolated conditions of $\mathcal{W} = 0$

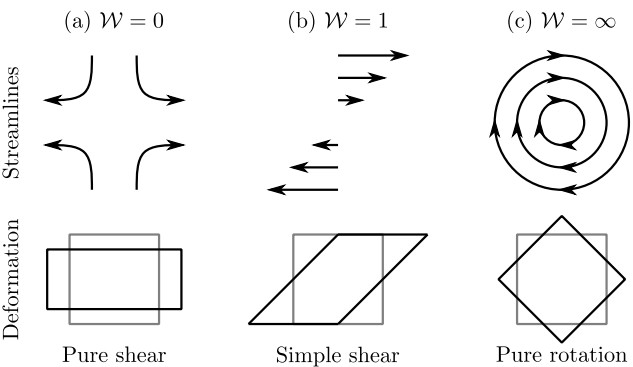

**Figure 2.** Schematics illustrating two-dimensional flow regimes at different vorticity numbers $\mathcal{W}$ (Eq. (1)): (a) Pure shear ($\mathcal{W} = 0$), (b) Simple shear ($\mathcal{W} = 1$), (c) A purely rotational deformation ($\mathcal{W} = \infty$). For each flow the streamlines and deformation regime produced are shown.

and $\mathcal{W} = 1$ alone (Jiang, 1994; Bailey and Eyster, 2003). Since then, many papers within structural geology have developed
conceptual models and analytical techniques to predict and recognise natural geological flows with vorticity numbers between
0 and 1 (Fossen and Tikoff, 1993; Tikoff and Fossen, 1995; Piazolo et al., 2002, 2004; ten Grotenhuis et al., 2002). In contrast,
such analysis is less common in discussions surrounding ice-core interpretation. This may be because such scenarios are a)
experimentally straightforward to achieve, and b) these cases can – as a first approximation - be associated with different ice
flow scenarios of an ice divide and the shallow ice approximation (although it should be noted, as we discuss below, that the
150 shallow ice approximation is only in simple shear near the base).

### 2.2.2 Two-dimensional deformation regimes in natural ice flow

As a first step towards exploring the fabrics produced by all possible deformation regimes, we will focus here on general
incompressible two-dimensional deformations. Although deformation regimes in the natural world will be three-dimensional,
exploring fabrics produced by two-dimensional deformation regimes is a natural first step away from the canonical regimes
of pure and simple shear. It is also common to limit the modelling of ice sheets to two dimensions; either in the vertical
cross-section (Pattyn et al., 2008; Martín et al., 2009) or through depth-integrated approaches (e.g Pegler, 2016; Joughin et al.,
2021).

In order to illustrate the range of vorticity numbers which are expected to occur in natural flows, we explore here a number
of scenarios. For an ice divide, the simulation shows that the vorticity number varies smoothly between 0 and 1 (Fig. 3). Due to
160 the vanishing of horizontal velocity at the central divide itself, the vorticity number is 0 there, corresponding to the regime of
pure shear along $x = 0$. Away from the divide the flow is dominated by a balance between gravity and the divergence of vertical
shear stresses (the shallow ice approximation). The vorticity number transitions continuously from 1 at the base, corresponding
to simple shear, towards close to 0 at the surface, corresponding to pure shear.

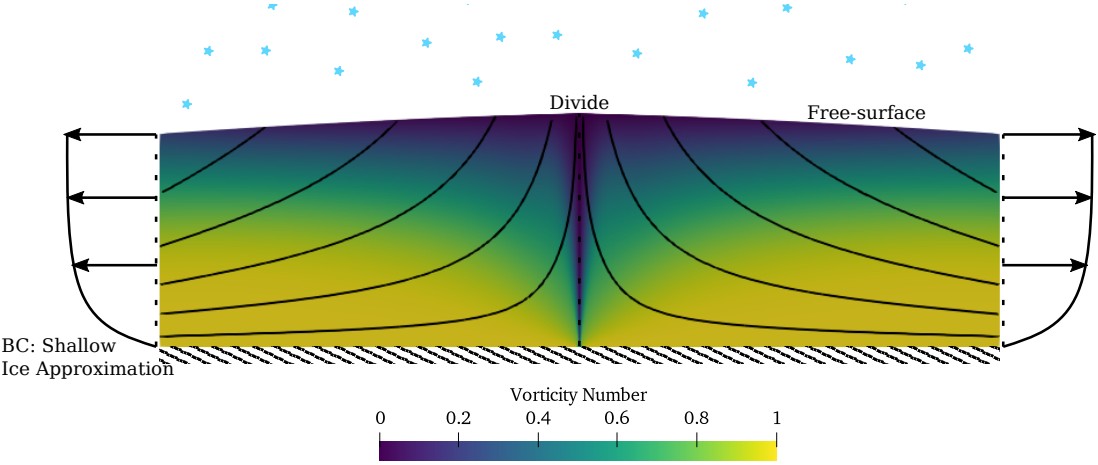

**Figure 3.** The vorticity number for ice flowing at a divide, showing a range from 0 to 1. The problem setup is from Martín et al. (2009) with isotropic ice. The simulation has an aspect ratio of 20, but only the region from $x = 0$ to $x = 10\,H$, where $H$ is the height of model domain, is shown. The domain has the velocity from the shallow-ice approximation imposed at the left and right boundaries, and the surface accumulation is set to match the outflow, corresponding to a steady state. No-slip is imposed at the base and a free surface is assumed at the top. The vorticity number is shown, alongside streamlines. This flow was computed using a full-Stokes solver written in FEniCS (Martin Alnæs et al., 2015) with $n = 3$ and solved using Taylor-Hood elements.

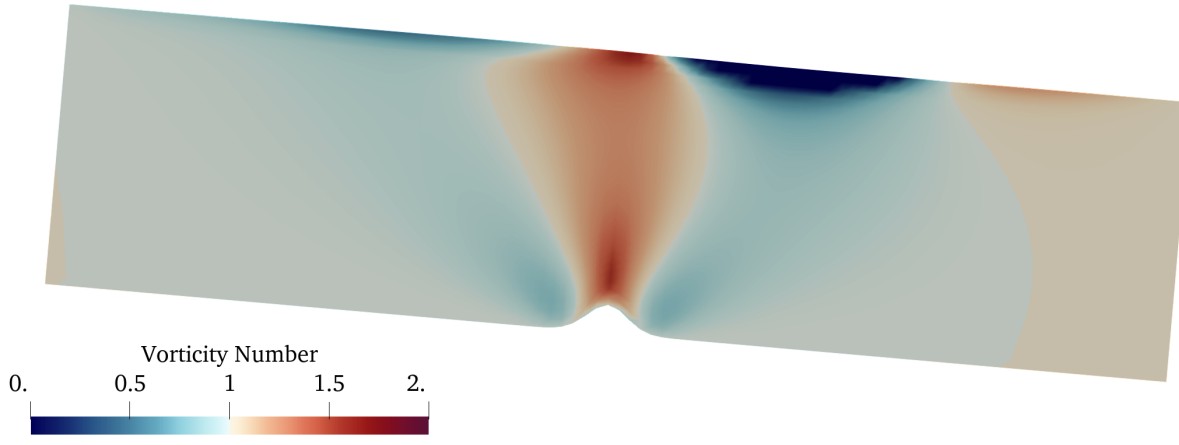

**Figure 4.** The vorticity number for ice flowing over a gaussian bump, performed in Elmer/ICE (Gagliardini et al., 2013) with $n = 3$. The ice is flowing from left to right, down a hill with angle $3.0°$. There is no slip at the base and a free surface at the top. The bump height $h = H/10$, where $H$ is the domain height. Over the bump the flow accelerates leading to vorticity numbers greater than 1.

    If we consider a 2D flow of ice over a Gaussian bump (Fig. 4), flowing from left to right down a hill of angle $3.0°$, we

find that vorticity numbers both between 0 and 1 and greater than 1, corresponding to deformation regimes more rotational

than simple shear, can occur. This indicates that, even in relatively simple configurations, vorticity numbers greater than 1 (in this case reaching 1.6) can potentially occur in the vertical cross-section. However, to-date, the fabrics produced for vorticity numbers above 1 have not been analysed.

The above examples focused on the two-dimensional flow in vertical cross-sections of ice-sheet simulations. To further explore the occurrence of vorticity numbers away from 0 or 1 in natural flows, we calculate an estimate of the vorticity number in the horizontal flow near the surface of the Antarctic Ice Sheet. To do this, we use surface velocity data from Antarctica (Mouginot et al., 2019), shown in Fig. 5. The vorticity number is calculated by combining surface velocity gradients with an estimate of the vertical shear rate ($\partial u/\partial z, \partial v/\partial z$). In order to estimate the lower-bound of the vorticity number near the surface, we estimate the upper bound on the vertical shear by using the shallow-ice approximation to assume no sliding at the base of the ice sheet. This is likely to be a conservative estimate of vorticity number in areas where it is known there is significant sliding at the base of the ice stream. We estimate the vertical shear for no basal slip, at a depth of $25\%$ into the ice-sheet, such that these vorticity numbers are at least valid to this depth: in regions with more slip at the base of the ice sheet, the vertical shear rate will be reduced and this estimate remains valid to a greater depth. The derivative $\partial w/\partial z$ is calculated using mass continuity and we have neglected the higher order contributions $\partial w/\partial x$ and $\partial w/\partial y$.

The resulting prediction for the vorticity number shown in the map of Fig. 5 indicates that there are widespread regions of Antarctica where the surface vorticity number near the surface is at intermediate values between 0 and 1, or at values greater than 1. The regions characterised by high vorticity numbers ($\mathcal{W} > 1$) typically occur in highly dynamic regions such as ice streams. In the majority of the ice sheet, the vorticity number will tend to 1 as depth increases due to the large vertical shear at the base. For ice streams and shelves, the vorticity number predicted here may also apply closer to the base.

## 2.3 Hierarchy of ice modelling spanning the microscale to the macroscale

The effects of deformation on the dynamics of ice covers a vast range of scales from the order of microns for studying grain-grain interactions, to continental scales of 1000s of kilometres when studying ice sheets. Consequently, different approaches must be used depending on the scale one seeks to work on, with micro-scale models serving to provide parametrisations of small-scale processes for use in larger scale models. At the scale of microns and millimetres, there exist several approaches for modelling the microstructure directly (e.g. Llorens et al., 2016; Kennedy and Pettit, 2015). This involves simulating grain-to-grain interactions with deformation regimes imposed via stress or velocity boundary conditions at the edges of the numerical domain. This is useful for improving our understanding of ice microstructure and fabric evolution. However, it cannot be scaled-up to be incorporated into ice-sheet models due to the numerical cost. At the largest scale, ice-sheet models typically neglect the effect of fabric entirely. The state-of-the-art for incorporating fabric evolution into large-scale models is to track the evolution of a 2nd-rank tensor representing the second moment of the orientation density function (to be defined below in Eq. (2)). As taking the second moment only retains information from terms up to order two in a spherical harmonic expansion (Montgomery-Smith et al., 2010), it is impossible with this approach to reproduce observed fabric patterns like a secondary cluster.

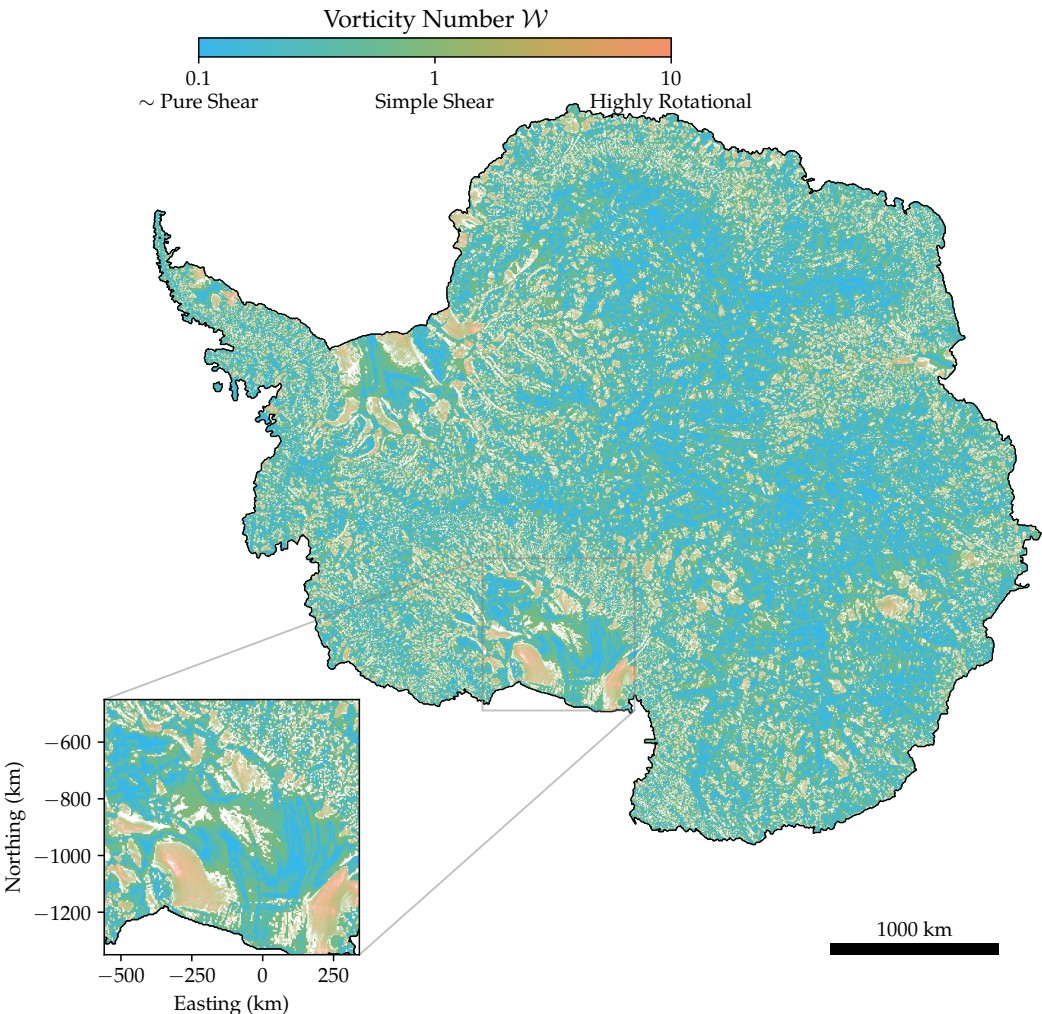

**Figure 5.** Figure to illustrate the range of vorticity numbers near the surface in Antarctica. This is calculated from the surface velocity data of Antarctica (Mouginot et al., 2019) after averaging over a $10 \times 10$ block and taking the mean value within each block. The calculation uses the horizontal velocity fields from the observed surface velocity combined with an estimate of the average vertical shear predicted to occur in the top $25\%$ of the ice sheet using the shallow-ice-approximation. The colour shows the vorticity number on a log scale. The transparency is used to hide areas with large uncertainty in the vorticity number. The inset shows the Ross Ice Shelf, with the easting and northing in Antarctic polar stereographic coordinates. This shows considerable variation across the continent, including deformation regimes not accessible in the laboratory.

In this contribution, we use the SpecCAF model from Richards et al. (2021), which forms a continuum model of the full
orientation density function with parameterisations of the underlying processes calibrated using laboratory experiments. In the hierarchy of ice modelling, SpecCAF seeks to model ice fabrics at a larger scale than models such as Llorens et al.

(2017). SpecCAF acts as a continuum model for fabric evolution which, through the use of spherical harmonic expansion, is sufficiently efficient computationally that it can be incorporated into ice-sheet models, while retaining all key processes. In this approach, grain-to-grain interactions are incorporated by considering ice as a continuum *mixture of orientations* (Faria, 2006).

Parameterisations then describe the effect of different microstructural processes on the orientation distribution function. Since it incorporates parameterisations of the mean effect of grain-grain interactions, there is no need to represent the strain and stress fields explicitly within the microstructure. However, these interactions are incorporated statistically within the model, and calibrated using laboratory experiments for which parameters such as strain, deformation rate and temperature are known. This model (SpecCAF) has been shown to predict fabrics accurately in both uniaxial compression and simple shear, and can

make predictions of fabric evolution with only the velocity gradient and temperature as inputs.

We note that, by itself, SpecCAF models fabric evolution only for given applied deformation. It does not include a viscosity formulation like the CAFFE model of Placidi et al. (2010). Thus, we take the velocity gradient as being prescribed, and consider the evolution of fabrics that occur as a result. In principle, the model could be coupled to any flow law (viscosity formulation) describing the effect of the fabric on the anisotropy. This could be considered in future.

## 2.4 Outline of paper

In section 3 we review the SpecCAF model and clarify its underlying modelling assumptions and comparison with other modelling approaches. In section 4.1 we determine the evolution of fabrics in general 2D deformation regimes, for the first time bridging the complete spectrum from pure shear to rigid-body rotation, across the range of temperatures seen in ice sheets. In section 4.2 we construct a complete regime diagram for two-dimensional deformation regimes documenting fabrics

that arise over the space of temperature, deformation regime, and strain, and explain the physical balances leading to these fabrics. Finally, in section 4.3 we investigate the time or strain scales over which ice fabric evolution takes place, as well as investigating the steady-state strength of ice fabrics, across the space of deformation regime and temperature. We then discuss the implication of these results for the interpretation of ice cores (section 5.3) and ice flow (sec 5.4).

## 3 Methods

### 3.1 The continuum approach

We begin by reviewing the SpecCAF model, which is developed, experimentally calibrated, tested and solved in Richards et al. (2021) and based on a mathematical continuum approach of Faria (2001, 2006) and Placidi et al. (2010). The SpecCAF model uses a continuum approach to represent the mass distribution of $c$-axes within a polycrystal $\rho^*(\boldsymbol{x}, t, \boldsymbol{n})$, termed the *orientation mass density*, defined according to

$$\rho(\boldsymbol{x}, t) = \int_{S^2} \rho^*(\boldsymbol{x}, t, \boldsymbol{n}) \, \mathrm{d}\boldsymbol{n}, \tag{2}$$

| Symbol | Name | Unit | Defined in |
|:---:|:---:|:---:|:---:|
| $\nabla \boldsymbol{u}$ | Velocity gradient | $\mathrm{s}^{-1}$ | |
| $\mathbf{D}$ | Strain-rate tensor | $\mathrm{s}^{-1}$ | |
| $\mathbf{W}$ | Spin-rate tensor | $\mathrm{s}^{-1}$ | |
| $\mathcal{W}$ | Vorticity number | - | Eq. (1) |
| $\rho$ | Density | $\mathrm{kgm}^{-3}$ | |
| $\rho^*$ | Orientation mass density | $\mathrm{kgm}^{-3}$ | Eq. (2) |
| $f^*$ | Orientation distribution function | - | Eq. (3) |
| $(\cdot)^*$ | Quantity defined over orientation space | | |
| $\boldsymbol{n}$ | Unit orientation vector | - | |
| $\nabla^*$ | Gradient over orientation space | - | Eq. (5) |
| $\boldsymbol{v}^*$ | Orientation transition rate | $\mathrm{s}^{-1}$ | Eq. (6) |
| $\mathcal{D}^*$ | Deformability | - | Eq. (7) |
| $\langle(\cdot)^*\rangle$ | Average over orientation space | | |
| $\lambda$ | Rotation recrystallization rate | $\mathrm{s}^{-1}$ | |
| $\beta$ | Migration recrystallization rate | $\mathrm{s}^{-1}$ | |
| $\iota$ | Ratio between basal-slip deformation and rigid-body rotation | - | |
| $\dot{\gamma}$ | Strain rate | $\mathrm{s}^{-1}$ | Eq. (9) |
| $(\tilde{\cdot})$ | Non-dimensionalised quantity | - | |
| $\gamma$ | Strain | - | Eq. (13) |
| $T$ | Temperature | $^\circ\mathrm{C}$ | |
| $J$ | J index | - | Eq. (14) |

**Table 1.** List of mathematical symbols used in this paper, including units and the equation if they are explicitly defined

where $\rho(\boldsymbol{x},t)$ is the mass density of ice, and $\boldsymbol{n}$ is a unit vector representing the direction of a $c$-axis. Here, $\rho^*(\boldsymbol{x},t,\boldsymbol{n})\,\mathrm{d}\boldsymbol{n}$ is the mass fraction of grains with orientations directed towards $\boldsymbol{n}$ within the solid angle $\mathrm{d}\boldsymbol{n}$. In accordance with the equation above, integrating $\rho^*$ over the space of possible orientations (the surface of a unit sphere $S^2$) gives the mass density of ice at that particular point in physical space, $\rho(\boldsymbol{x},t)$. The non-dimensional equivalent of this is the *orientation distribution function*:

$$f^* = \rho^*/\rho \tag{3}$$

SpecCAF incorporates the effect of basal-slip deformation, migration recrystallization and rotation recrystallization (all as functions of temperature), as well as rigid-body rotation, to develop an evolution equation for the orientation mass density $\rho^*$. As discussed above, the model evolves $\rho^*$ as the dependent variable, with the net effects of grain-to-grain interactions incorporated through parameterisations. This is similar to other continuum approaches, for example how explicit descriptions

of particle interactions are not included in the Stokes equations, yet are satisfactorily modelled statistically through parameterisation in the form of a constitutive relation. In the present case, rotation recrystallization is modelled as a diffusion of

concentrations of $\rho^*$, originally the idea of Gödert (2003). The effect of migration recrystallization is modelled through an orientation-dependent source term producing $\rho^*$ preferentially at orientations where grains would be likely to have a large basal shear stress, and vice-versa. Basal-slip deformation is incorporated in accordance with the 'deck-of-cards' analogy, and assuming linear dependence on the strain-rate tensor (Placidi et al., 2010). The resulting continuum model has been shown to reproduce all detailed features of the orientation distribution functions calculated from experimental samples (Richards et al., 2021).

The essential continuum approach was proposed previously by Faria (2001, 2006). Grains with the same orientations are called a *species*. There has been discussion in the literature (Gagliardini, 2008; Faria et al., 2008) on whether this theory implicitly includes a *Taylor assumption*, namely, that all ice grains forming the polycrystal experience the same strain-rate. The Taylor assumption has been shown not to be valid for ice (Castelnau et al., 1998). To summarise this debate, Gagliardini (2008) suggests that the assumption in Faria (2006) that the strain-rate of a species is independent of orientation is equivalent to every grain undergoing the same deformation (a Taylor assumption). However, Faria et al. (2008) rejected this assertion and replied that this assumption only requires that grains move with the surrounding material, with no direct constraint on the individual deformation of grains. In accordance with the continuum approach, the net effect of deformations on individual grains, which can vary from grain to grain, is incorporated via parametrisations of the overall net effect of these interactions. Since these parametrisations are calibrated using laboratory experiments (Richards et al., 2021), they represent net effects relevant to real samples in which grains do not all experience the same strain. Therefore, the continuum model does not impose a Taylor assumption on the grain deformation. Furthermore, care should be taken to attribute our calibrated parameters as applying specifically to the bulk interactions representing their net statistical effects in the model, as opposed to grain-grain interactions.

Despite the model not including the Taylor hypothesis, the term for basal-slip deformation in the equation below is similar to that which would be derived from a Taylor homogenisation of ice under a simple basal-slip only model (Gagliardini et al., 2009). The only exception is that the rate of visco-plastic deformation can vary relative to rigid-body rotation.

## 3.2 Model specification

The evolution equation for $\rho^*$ under the framework described above was first defined in Placidi et al. (2010):

$$\frac{\partial \rho^*}{\partial t} = -\nabla^* \cdot (\rho^* \boldsymbol{v}^*) + \lambda \nabla^{*2}(\rho^*) + \beta \big(\mathcal{D}^* - \langle \mathcal{D}^* \rangle\big)\rho^*, \tag{4}$$

where $\lambda$ and $\beta$ are parameters, to be defined below. Here, $\nabla^*$ is the gradient operator in orientation space, restricted to the surface of a sphere (i.e. the space of possible orientations) defined by:

$$\nabla^* \boldsymbol{v}^* = \frac{\partial \boldsymbol{v}^*}{\partial \boldsymbol{n}} - \left(\frac{\partial \boldsymbol{v}^*}{\partial \boldsymbol{n}} \cdot \boldsymbol{n}\right)\boldsymbol{n} = \frac{\partial v_i^*}{\partial n_j} - \frac{\partial v_i^*}{\partial n_l}n_l n_j. \tag{5}$$

The parameters $\lambda$ and $\beta$ represent the rates of rotational and migration recrystallization, respectively. The orientationally dependent term $\mathcal{D}^*$ will be defined below in Eq. (7). The term $\boldsymbol{v}^*$ defines the orientation transition rate, defined by Placidi et al.

(2010) as:

$$v_i^* = W_{ij}n_j - \iota(D_{ij}n_j - n_i n_j n_k D_{jk}). \tag{6}$$

This equation is broadly similar to the rotation of an individual $c$-axis in a discrete model. The term $W_{ij}n_j$ in Eq. (6) represents the effect of rigid-body on the fabric, and the second term models basal-slip deformation. The non-dimensional parameter $\iota$ represents the ratio of basal-slip deformation to rigid-body rotation.

The parameter $\lambda$ (s$^{-1}$) represents the rate of rotational recrystallization, modelled as a diffusional term. Migration recrystallization is modelled by an orientation-dependent source term, with the rate controlled by $\beta$ (s$^{-1}$). The orientation dependence is governed by the deformability, defined by:

$$\mathcal{D}^* = 5\frac{(D_{ij}n_j)(D_{ik}n_k) - (D_{ij}n_j n_i)^2}{D_{mn}D_{nm}}. \tag{7}$$

For a given stretching tensor $\mathbf{D}$, and for a basal plane with normal $\boldsymbol{n}$ this function represents the normalised strain-rate (or stretching) acting on the basal plane. Therefore, $\mathcal{D}^*$ will be greater at orientations where it is easier to slip along the basal plane. Because ice deforms primarily by slip along the basal plane, this is a good approximation for the accumulation of deformation energy in a physical grain, which drives migration recrystallization. The average of $\mathcal{D}^*$ is defined as:

$$\langle \mathcal{D}^* \rangle = \int_{S^2} \frac{\rho^*}{\rho} \mathcal{D}^* \, \mathrm{d}\boldsymbol{n} \tag{8}$$

If $\mathcal{D}^*$ is greater than the average value $\langle \mathcal{D}^* \rangle$ then this term acts as a source term for $\rho^*$ at this orientation. This parameterises grains growing or nucleating at this orientation. Note that the total production and consumption of $\mathcal{D}^*$ always balance. The factor of 5 in Eq. (7) is a convention.

## 3.3 Non-dimensionalisation

To apply Eq. (4) to spatially homogeneous fabrics and to compare to fabrics deformed in the laboratory we perform a non-dimensionalisation, where we non-dimensionalise by a characteristic density $\rho_0$ and strain-rate, which we define as:

$$\dot{\gamma} = \sqrt{\frac{1}{2}D_{ij}D_{ji}}. \tag{9}$$

This is the effective strain rate, corresponding to the second invariant of the strain-rate tensor $\mathbf{D}$. The non-dimensional variables are represented with tildes and are defined as:

$$f^* = \frac{\rho^*}{\rho_0}, \quad \tilde{\mathbf{D}} = \frac{\mathbf{D}}{\dot{\gamma}}, \quad \tilde{\mathbf{W}} = \frac{\mathbf{W}}{\dot{\gamma}}, \quad \tilde{\lambda}(T,\dot{\gamma}) = \frac{\lambda(T,\dot{\gamma})}{\dot{\gamma}}, \quad \tilde{\beta}(T,\dot{\gamma}) = \frac{\beta(T,\dot{\gamma})}{\dot{\gamma}}, \tag{10}$$

where for clarity we use $f^*$ to refer to the non-dimensional orientation distribution function (note that the strain rate we non-dimensionalise with in this paper is half the strain-rate we used in Richards et al. (2021), which was based on the experimental strain rate). Recasting Eq. (4) in terms of the non-dimensional variables above, we obtain:

$$\frac{\partial f^*}{\partial \tilde{t}} = -\nabla^* \cdot [f^* \tilde{\boldsymbol{v}}^*] + \tilde{\lambda}\nabla^{*2}(f^*) + f^*\tilde{\beta}(\mathcal{D}^* - \langle \mathcal{D}^* \rangle), \tag{11}$$

where

$$\tilde{v}_i^* = \tilde{W}_{ij}n_j - \iota(\tilde{D}_{ij}n_j - n_in_jn_k\tilde{D}_{jk})$$

is the non-dimensional form of the orientation transition rate (Eq. (6)). Richards et al. (2021) constrained the non-dimensional parameters $\tilde{\lambda}, \iota, \tilde{\beta}$ as functions of temperature. This was done by finding the parameters which gave a best fit to experimental results in simple shear. These parameters were then found to predict well fabrics produced in uniaxial compression. In this paper, we use a best fit from the entire inversion performed in Richards et al. (2021), rather than just from the inversion performed in simple shear. Furthermore, as the strain-rate we use to non-dimensionalise is half that used in Richards et al. (2021), the non-dimensional recrystallization parameters $(\tilde{\lambda}, \tilde{\beta})$ used in this paper are double those used in Richards et al. (2021). The non-dimensional parameters as functions of temperature are shown in Fig. 6, along with the data points used in the inversion from Richards et al. (2021) and $80\%$ and $95\%$ confidence intervals. In the accompanying supplement to this article a parameter sensitivity study can be found, reproducing the figures below with (a) $\iota_{\mathrm{max}}, \tilde{\beta}_{\mathrm{max}}, \tilde{\lambda}_{\mathrm{min}}$ and (b) $\iota_{\mathrm{min}}, \tilde{\beta}_{\mathrm{min}}, \tilde{\lambda}_{\mathrm{max}}$, with the max and min values taken from the $80\%$ confidence interval in Fig. 6. (a) and (b) give the strongest and weakest fabric, respectively.

Equation (11) and the parameters defined by the best fit lines in Fig. 6 combined, when solved with the spectral method defined in Richards et al. (2021), represent the SpecCAF model.

## 3.4 Pole figure and cross section representation

As a preliminary illustration of the model output and its representation, we show in Fig. 7 an example of model output obtained by solving the model at $T = -5°$ C, in simple shear ($\mathcal{W} = 1$). The principal axes are orientated at $\theta = \pm 45°$ directions of the pole figure. To visualise how the fabric changes with increasing strain, we plot slices of the pole figure at $y = 0$. The example pole figure in (a) is plotted at a strain of $\gamma = 0.345$. The value of $\rho^*$ at $y = 0$ is plotted in (c) for each strain. This shows how the fabric develops from an isotropic fabric. A secondary cluster can clearly be seen as a transient feature which has mostly disappeared by $\gamma = 0.8$.

As an example comparison of the model prediction and experimental observations, we have included a pole figure from laboratory experiments (Qi et al., 2019) in Fig. 7b. This is at the same temperature and strain as the model output in (a). There is very good agreement between the model and experiments. More experimental comparisons are detailed in Richards et al. (2021), showing that the model is able to generally capture both qualitative and quantiative features of fabrics observed in existing experiments.

# 4 Results

## 4.1 General fabric evolution: dependence on temperature and vorticity number

We explore fabric evolution across a complete, continuous range of vorticity numbers $\mathcal{W}$ for two-dimensional deformation regimes (spanning $\mathcal{W} = 0$ to $\infty$), and a continuous range of temperatures $T$ relevant to ice-sheet flow ($T = -30$ to $-5$ ° C).

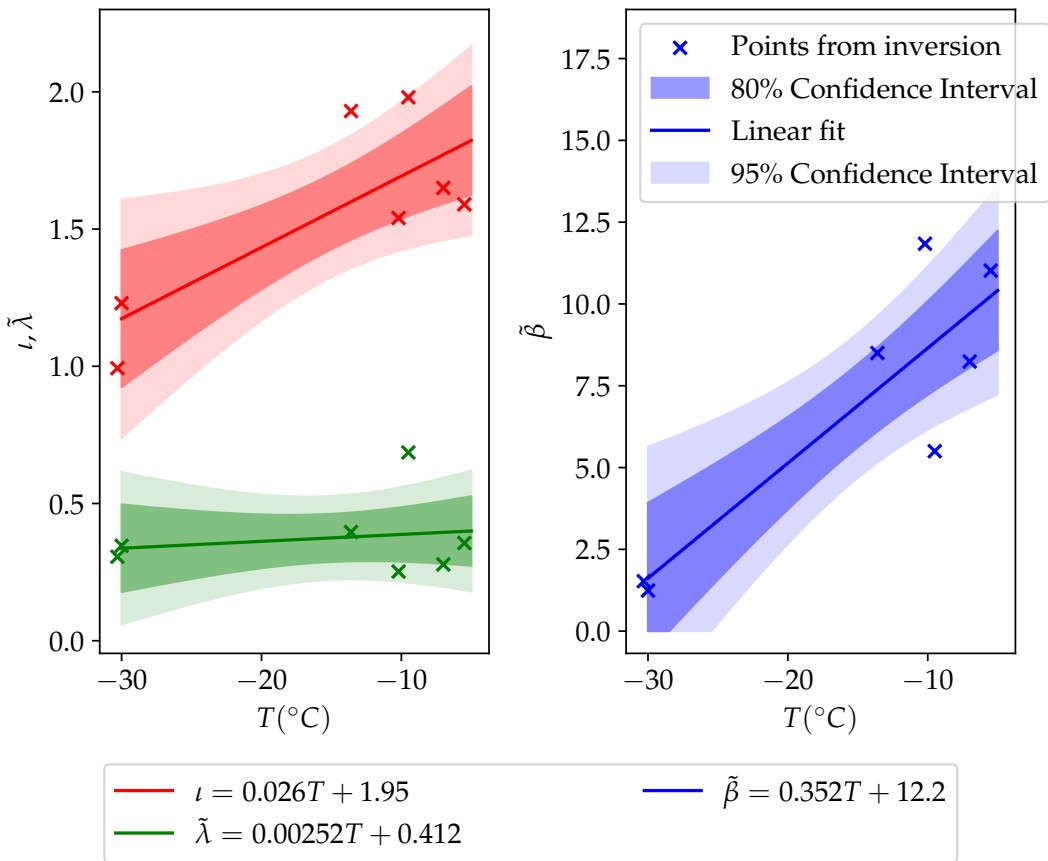

**Figure 6.** The model parameters $\iota, \tilde{\lambda}, \tilde{\beta}$ as functions of temperature, determined by linear regression to experimental data in both compression and simple shear, as conducted in Richards et al. (2021). The 95% and 80% confidence intervals are shown.

For all cases, we assume an initially isotropic fabric. To make comparisons, we will limit our analysis to fabrics undergoing a constant two-dimensional deformation and at a constant temperature. We apply a velocity gradient which varies with vorticity number. This is chosen to be:

$$\nabla\boldsymbol{u} = \begin{bmatrix} 1 & 0 & k \\ 0 & 0 & 0 \\ -k & 0 & -1 \end{bmatrix} \quad \text{such that} \quad \mathbf{D} = \begin{bmatrix} 1 & 0 & 0 \\ 0 & 0 & 0 \\ 0 & 0 & -1 \end{bmatrix} \quad \text{and} \quad \mathbf{W} = \begin{bmatrix} 0 & 0 & k \\ 0 & 0 & 0 \\ -k & 0 & 0 \end{bmatrix} \tag{12}$$

This gives $\dot{\gamma} = 1$, and the vorticity number $\mathcal{W} = k$. The vorticity number, defined in Eq. (1), gives the ratio of vorticity to strain-rate magnitude. This gives pure shear for $\mathcal{W} = 0$, simple shear for $\mathcal{W} = 1$ and rigid-body rotation as $\mathcal{W} \to \infty$. This velocity gradient is chosen such that the principal strain axes are unchanging as $\mathcal{W}$ varies. Because of this, the simple shear ($\mathcal{W} = 1$) condition is rotated 45° from the usual definition of $\partial u / \partial z = 1$, 0 otherwise, and this is seen in the pole figures.

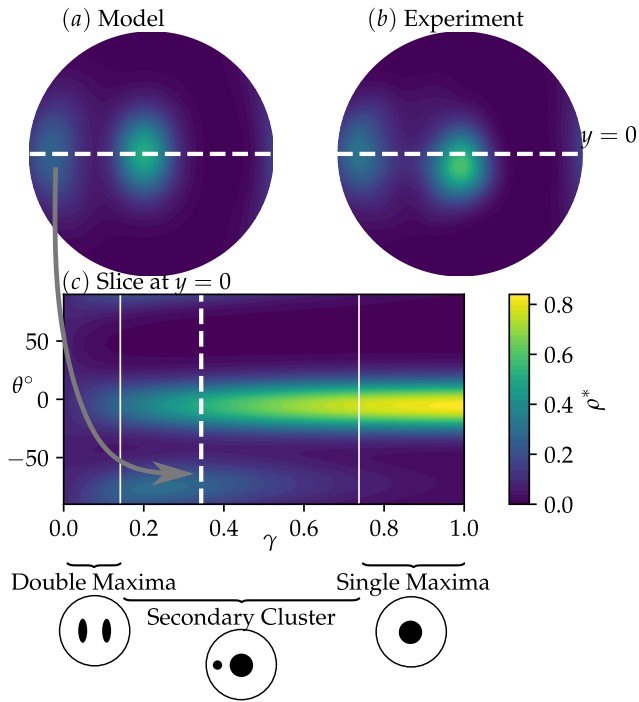

**Figure 7.** To illustrate the fabric we show a simulation in simple shear at $T = -5°$ C. (a) shows a pole figure from the model at an effective strain of $\gamma = 0.345$ (where the strain-rate is defined as in Eq. (9)). (b) shows a pole figure from laboratory experiments (Qi et al., 2019) at the same temperature and strain, showing good agreement. More comparisons like this can be found in Richards et al. (2021). The white dotted line in (a) and (b) shows $y = 0$. (c) shows $f^*$ at $y = 0$ against strain. Here the white dotted line highlights the strain at which the pole figure is plotted. $\theta$ is the polar angle. The grey arrow shows the secondary cluster transposed from the pole figure to the $\gamma - \theta$ diagram. Also highlighted is the classification of the different fabric types at different strains; from double-maximum to secondary cluster to the steady state single-maximum. For this figure only the principal strain axes are oriented at $\theta = \pm 45°$.

We define the strain as:

$$\gamma = \int \dot{\gamma} \, \mathrm{d}t = \int \sqrt{\frac{1}{2} D_{ij} D_{ji}} \, \mathrm{d}t, \tag{13}$$

i.e. based on the effective strain rate defined in Eq. 9. Different measurements (e.g. an axial strain) are sometimes used in experiments but these can be converted to the effective strain by normalising by the effective strain-rate of the deformation. With the velocity gradient fully defined, we explore the fabric dynamics produced across $T$-$\mathcal{W}$ space in Fig. 8. For each square we show the slice through the pole figure at $y = 0$ (explained in Fig. 7) up to a finite strain of $\gamma = 1$. The temperature range is from $-30°$ C to $-5°$ C, temperatures typical in ice sheets (Duval et al., 2010). The vorticity number ranges from $\mathcal{W} = 0.1$, very close to pure shear, and $\mathcal{W} = 10$ representing highly vortical flow with curved streamlines. This provides a detailed picture of how the fabric evolves with increasing strain, providing insights into deformation regimes between pure and simple shear as well analysis of fabrics produced by deformation regimes more rotational than simple shear. For low vorticity

numbers, a single-maximum can be seen at low temperatures which develops into double-maximum as strain increases. At high temperatures with a double-maximum, as $\mathcal{W}$ increases the clusters are moved by the rotational component of the deformation. In combination with basal-slip deformation, this results in one stable and one unstable cluster (see section 2.1.3). The $c$-axes of the unstable cluster rotate under basal-slip deformation and vorticity such that they are at an orientation where they are consumed by migration recrystallization as strain increases, leading to a single-maximum at high strains, for $\mathcal{W} \sim O(1)$.

Figs. S1 and S2 in the supplement show that variations in the parameters from Fig. 6 affect the strength of the primary cluster primarily, but do not affect the variation with vorticity number or the transition from one fabric type to another.

We also show in Fig. 8 analysis of fabrics produced in highly rotational ($\mathcal{W} > 1$) deformation regimes, which we have shown to occur (Figs. 4,5). Figure 8 shows that the fabric is strongest for $\mathcal{W} = 1$, and weakens as vorticity increases past this. For example, for $\mathcal{W} = 10$ there is only a very weak fabric produced. Furthermore, at large vorticity numbers oscillation can be seen in the fabric pattern.

To further analyse the limit of very large vorticity numbers we show the fabric produced as $\mathcal{W} \to \infty$ in Fig. 9. This fabric is seen for any vorticity number above $\mathcal{W} \approx 50$. To measure fabric concentration the $J$ index is often used (Bunge, 1982), defined by

$$J = \int_{S^2} f^{*2} \, d\boldsymbol{n}. \tag{14}$$

Although the $M$-index can also be used to measure fabric strength and may be more reliable (Skemer et al., 2005), the $J$-index can be calculated very efficiently, enabling exploration of the parameter space used in this paper. The $J$ index of this fabric is 1.16, very close to completely isotropic ($J = 1$). It is unlikely this weak girdle fabric would be distinguishable from an isotropic fabric in a physical sample, where the fabric is determined by sampling a limited number of grain orientations. Sensitivity analysis in the supplement reveals no change in the fabric pattern and only a small change in fabric strength, ranging between 1.11–1.23 for the min and max parameters.

## 4.2 Fabric regime diagrams for cluster angle and fabric type

To distil all the complex information shown in Fig. 8 and make this information more easily accessible, we present results showing a regime diagram of fabric patterns (Fig. 10) as well as the the angle between the primary cluster and the closest principal strain axes (i.e. the axis of compression, Fig. 11). To define whether a fabric is a double-maximum, secondary cluster or single-maximum we take the ratio of the two largest peaks in the fabric. If the 2nd largest peak is less than $10\%$ the strength of the largest peak, it defines as a single-maximum. If the strength of the 2nd largest peak is between $10\%$ and $90\%$ of the largest peak, it is defined as a secondary cluster. If it is $> 90\%$ it is defined as a double-maximum. Contour lines of primary cluster angle at $20°$ and $50°$ are also shown. This shows the different fabric types (Fig. 1) across the space of temperature, vorticity number and finite strain.

Figure 10a shows the initial fabric after a finite strain of only $\gamma = 0.3$. There are three regimes at this finite strain. For approximately $\mathcal{W} < 0.7$ a double-maximum is produced. There is a small region, at high vorticity numbers and primarily at low temperatures but extending into high temperatures, at which a single-maximum is produced. Otherwise a secondary

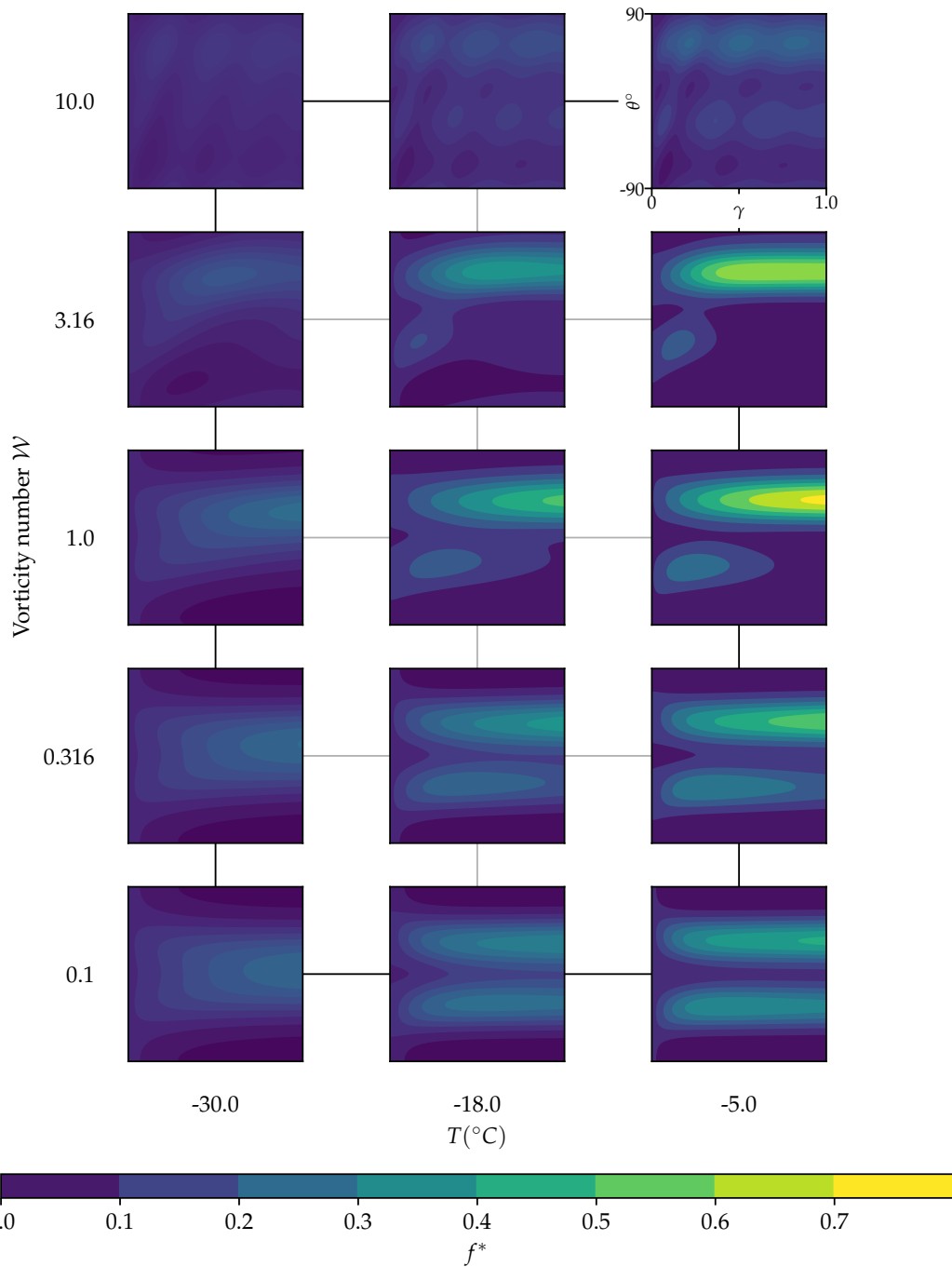

**Figure 8.** Slices of the pole figure showing the value of the orientation distribution function $f^*$ at $y = 0$ for an array of temperatures and vorticity numbers. All plots go to a strain of $\gamma = 1$. The colour limits are the same for all plots.

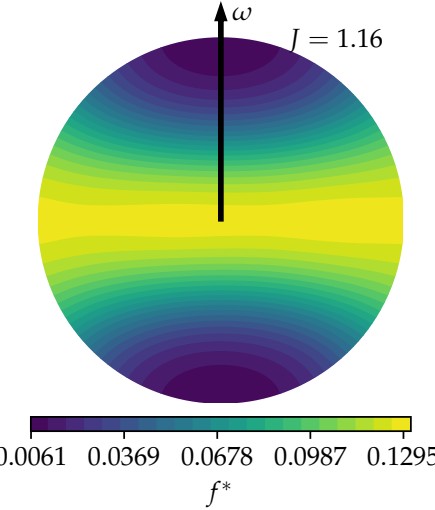

**Figure 9.** Pole figure for $\mathcal{W} \to \infty$ and $T = -5^\circ$ C at steady state. A very weak girdle fabric is produced, with the girdle coincident with the axis of vorticity, shown by an arrow. This fabric has a $J$ index of 1.16, where $J = 1$ is an isotropic fabric.

cluster is produced, this occurs for relatively high vorticity numbers and is more dominant at higher temperatures, as expected. At higher finite strain the double-maximum pattern becomes less prevalent, only occurring at lower vorticity numbers or not at all. This highlights the transient nature of this pattern. As a reminder, the double-maximum is the 2D equivalent of a girdle fabric (Fig. 1). The double-maximum fabric is only present up to a finite strain of about $\gamma = 0.5$. In Fig. 10c, at $\gamma = 1.0$ the parameters space is dominated by single-maximum and secondary cluster patterns. Ice fabrics which develop at higher temperatures $T > -20^\circ$C are dominated by secondary cluster patterns, with the exception of around $\mathcal{W} \approx 3$, where a single-maximum occurs because the secondary cluster is too weak. At lower temperatures $T < -25^\circ$C a single-maximum is produced because migration recrystallization is not active enough for multiple clusters to be produced. This balance between a single-maximum fabric and a secondary cluster fabric continues as the finite strain increases, with a single-maximum also becoming more prevalent at high temperatures for vorticity numbers around 1 (Fig. 10e and f).

Figures S5 and S6 in the supplement show Fig. 10 with the strongest and weakest possible fabric based on the 80% confidence intervals in Fig. 6. The overall picture is similar. The variation with vorticity number is approximately unchanged and boundaries between the regimes shift by roughly $\pm 7^\circ$ C.

The angle between the primary cluster and the closest principal axis of deformation (i.e. the axis of compression) is shown in Fig. 11 at six separate finite strain values, across $\mathcal{W}$-$T$ space. Even at a low finite strain of $\gamma = 0.3$ there is already an established difference in angle across the parameter space (Fig. 11a). Low temperatures and low vorticity numbers have the primary cluster most closely aligned with the compression axis. The angle then increases as both temperature and especially vorticity number increase. As strain increases the variation in angle increases. However, for a finite strain greater than 0.5 the

angle is mostly invariant with strain. Across the strain and temperature range, an angle of around $40°$ implies simple shear ($\mathcal{W} \approx 1$), whereas if the primary cluster and compression axis are coincident, this suggests pure shear at $T \approx -30°$ C .

Figures S7 and S8 in the supplement show Fig. 11 with the strongest and weakest possible fabric based on the 80% confidence intervals in Fig. 6. At low vorticity numbers and temperatures, the angle between the primary cluster and compression axis is slightly sensitive to variations in parameters, but outside of this space the angle is roughly unchanged.

To illustrate the difference in pole figure patterns at the same finite strain but different temperatures and deformation regimes we plot pole figures at a finite strain of $\gamma = 2$ (Fig. 12) overlaid onto a regime diagram of fabric patterns. The pole figures are centred at the vorticity number and temperature they are simulated at. Figure 12 highlights fabrics are still variable despite being in the same regime. The fabric at $\mathcal{W} = 1, T = -5°$ C is much stronger than the fabric at $\mathcal{W} = 10, T = -30°$ C or $\mathcal{W} = 0.1, T = -30°$ C, however they are all single-maximum. We also note the difference in angle of the primary cluster across the parameter space: approximately $0°$ for $\mathcal{W} = 0.1, T = -30°$ C but increasing as $\mathcal{W}$ and $T$ increase, as shown in Fig. 11.

## 4.3  Analysis of fabric evolution timescales

A variable that is central to the interpretation of ice core fabrics is the timescale or, equivalently in the non-dimensional problem, the finite strain over which fabric evolution occurs. In Fig. 13 we explore fabric evolution timescales. Fig. 13a shows $J$-index, representing fabric strength, at steady-state. We also show the strain at halfway to steady state (Fig. 13b) and the $J$-index at a finite strain of $0.5$ (Fig. 13c). An example showing the evolution of the $J$ index with strain at a single temperature and vorticity number is shown in in Fig. 13d.

Fig. 13a shows that as the steady-state fabric always increases in strength. Furthermore, the steady-state fabrics are strongest at a vorticity number of around 0.7 across almost the whole temperature space. Comparing to the fabric strength at $\gamma = 0.5$ (Fig. 13c) highlights how far fabrics are from reaching steady state at this strain, the fabric is approximately $\frac{2}{3}$rds of the steady-state value at $\gamma = 0.5$. This also shows that at this strain, the strongest fabrics occur not for vorticity numbers around 1, but for higher vorticity numbers of $\sim 3$.

Figure 13b shows the finite strain at which the $J$-index is halfway to its steady-state value. This can be considered the half-life over which fabric evolution occurs. Measuring the strain at halfway to steady-state gives insight into the timescale over which fabric development occurs and is more robust measure than estimating the strain at steady state, which we found was sensitive to parameter variations such as those shown in the supplement. Vorticity numbers closest to 0 and the coldest temperatures have the highest halfway strain. Under these conditions, neither rigid-body rotation nor migration recrystallization are active to a significant degree: the fabric evolution is dominated by basal-slip deformation. Fabrics also take longer to develop closer to a vorticity number of 0.7, due to the fact that this vorticity number has the strongest fabrics generally. For $\mathcal{W} > 1$ the strain to reach half strength decreases, as the fabrics are shown to be generally weaker as vorticity number increases.

Figures S11 and S12 in the supplement show Fig. 13 with the strongest and weakest possible fabric based on the 80% confidence intervals in Fig. 6. These figures show that the halfway strain is fairly insensitive to changes in the parameters, with the maximum varying by around $\pm 20\%$. The maximum halfway strain remains at $\mathcal{W} = 0.1, T = -30°$ C for both Fig. S11 and

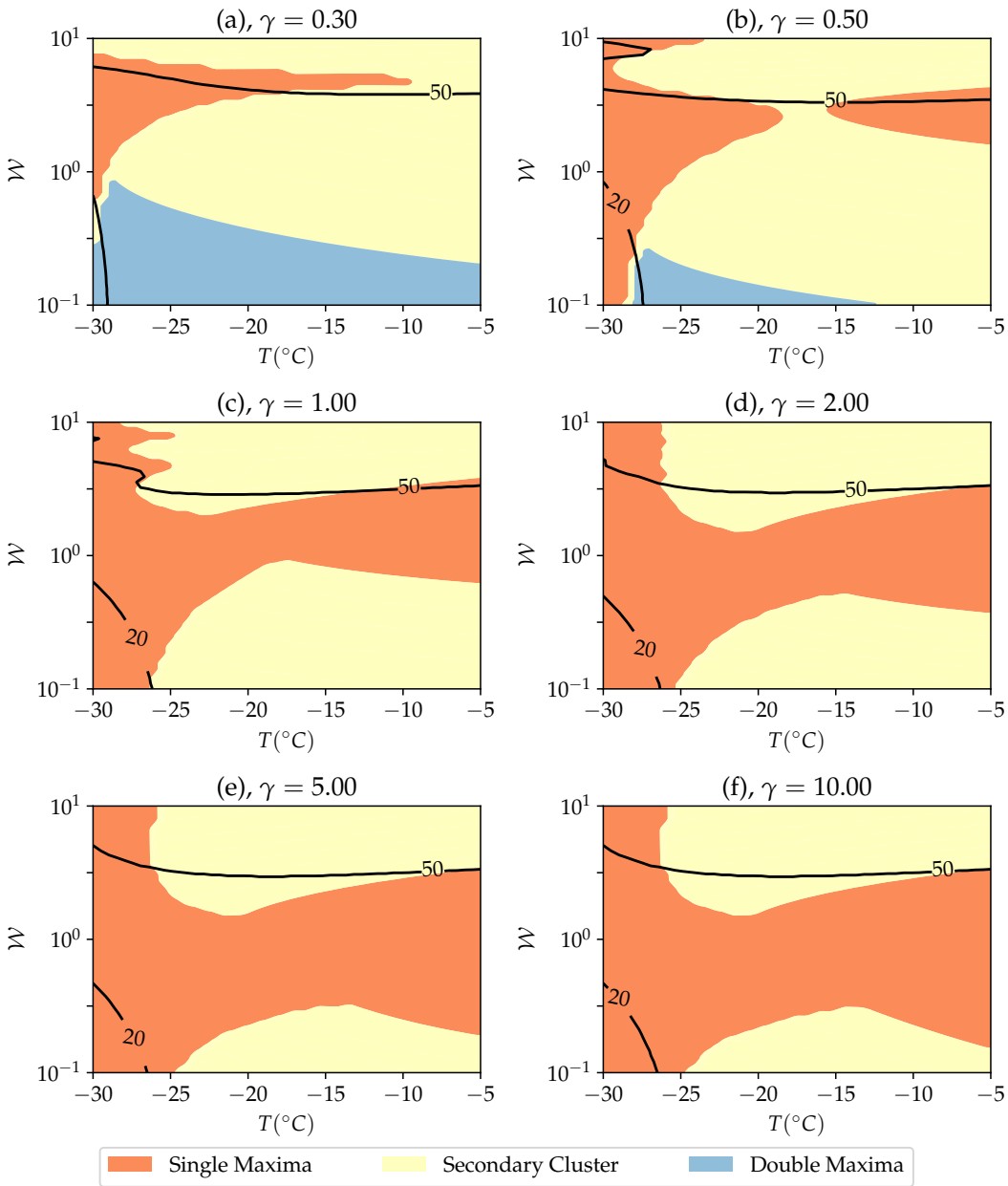

**Figure 10.** Regime diagram of the different fabric patterns which occur (defined in Fig. 1). The angle of the primary cluster in Fig. 11 at $20°$ and $50°$ is also overlaid. The diagrams are shown for discrete strain values. The resolution of this figure is $50 \times 50$ across the parameter space.

S12. The $J$ index at steady state is more sensitive to changes in parameters but the general picture of how this variable changes across the $T$-$\mathcal{W}$ space is similar, with the upper bound showing generally less variation across the parameter space.

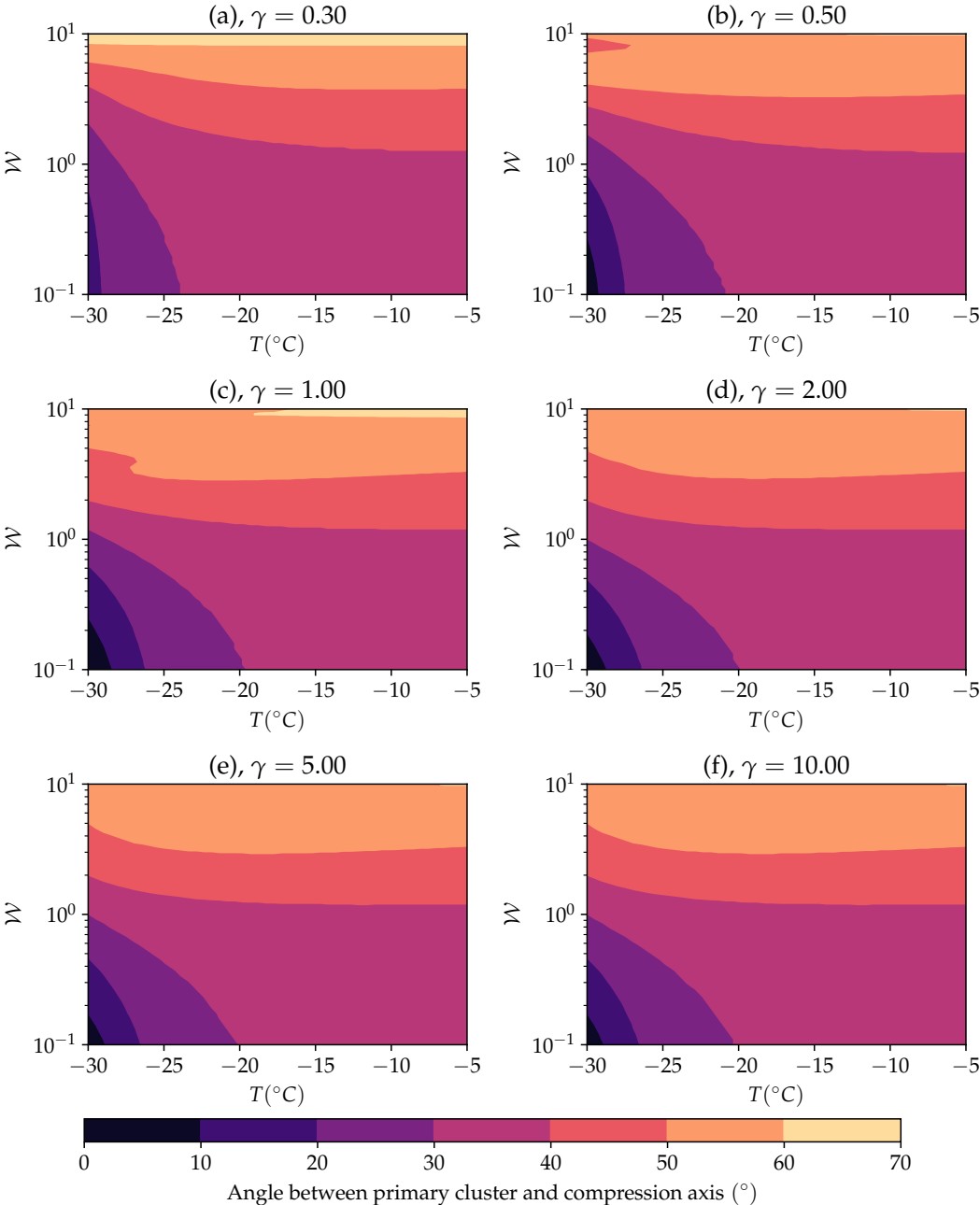

**Figure 11.** Contour plots showing the angle (in degrees) of the largest cluster from the compression axis. Panels are shown for progressively increasing finite strain values. The resolution of this figure is $50 \times 50$ across the parameter space.

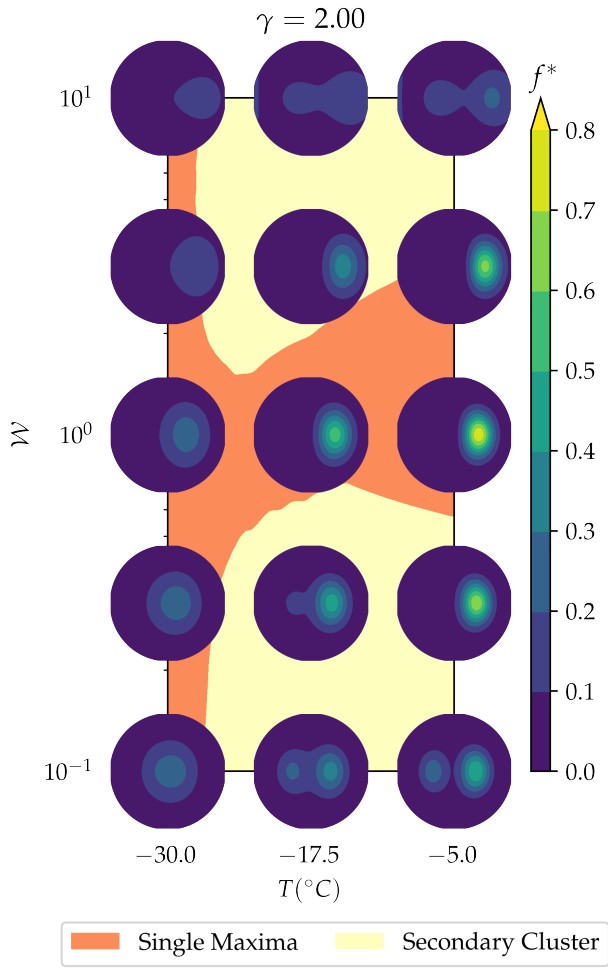

**Figure 12.** Pole figures overlaid onto the regimes at $\gamma = 2$. The pole figures are centred at the vorticity number and temperature they occur.

## 5 Discussion

The analysis presented here gives predictions for the fabric patterns produced over the whole range of vorticity numbers and temperatures arising for incompressible two-dimensional deformation regimes, a first for fabric modelling. We have limited the analysis here to fabrics produced under a constant deformation regime and temperature. Although ice in the natural world will
undergo changing deformation regimes, our analysis is a first step to provide insights into fabrics produced for deformation regimes away from pure and simple shear. Furthermore, the fabrics analysed here are highly relevant for ice deformed in the laboratory, which is in most cases deformed at constant temperature and vorticity number.

Ice in the real-world will undergo three-dimensional deformations yet it is common to model ice sheets in two dimensions; either along the vertical cross-section, such as in Fig 3 or Martín et al. (2009), or through depth-integrated approaches (e.g

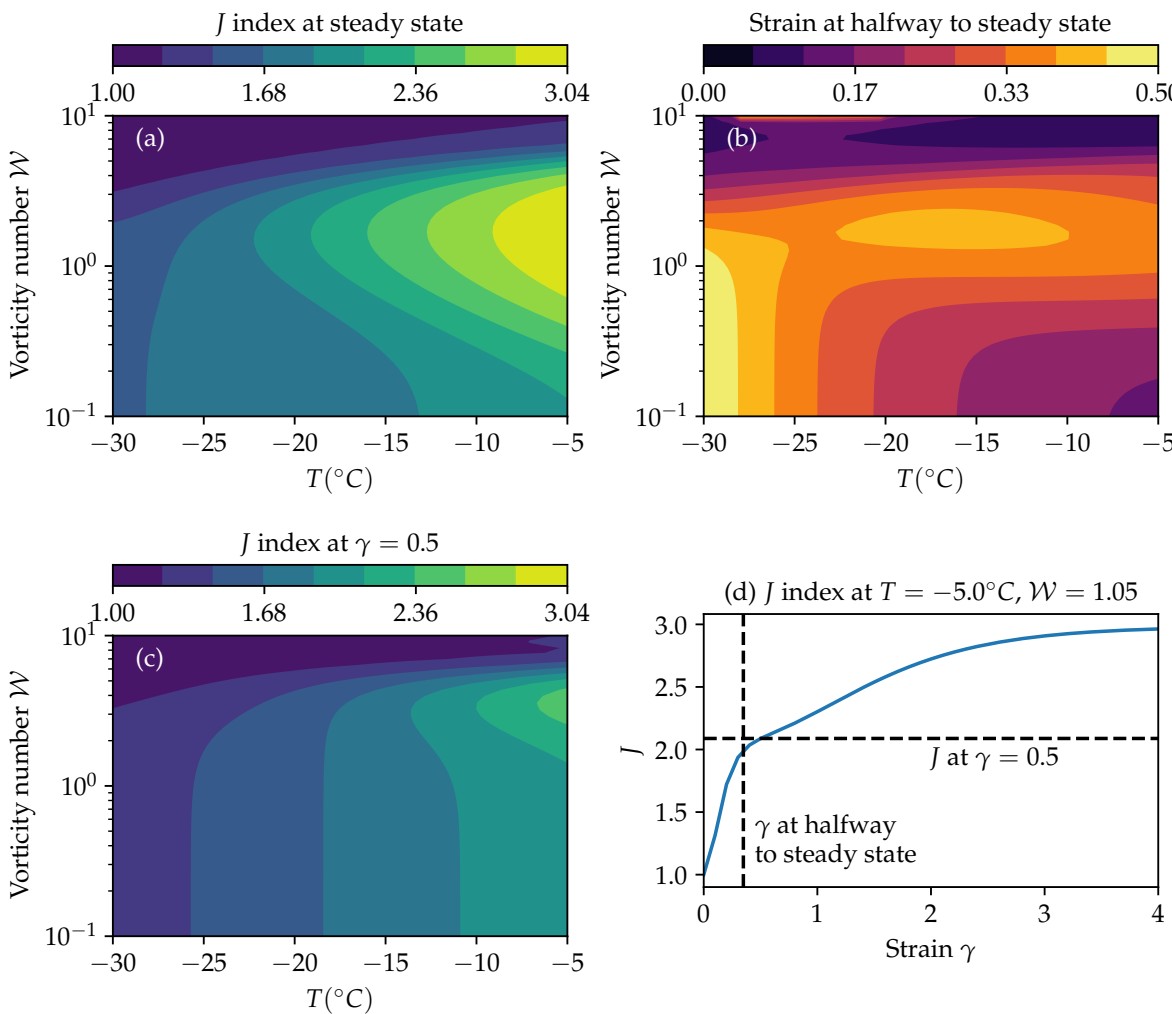

**Figure 13.** Properties of steady-state fabrics across the $\mathcal{W}$–$T$ parameter space. (a) shows the $J$ index at the steady state. (b) shows the finite strain required to reach halfway to the steady-state value of $J$. (c) shows the $J$ index at a strain of $\gamma = 0.5$. (d) shows the plot of $J$ against strain $\gamma$, illustrating the strain at halfway to steady state and $J$-index at $\gamma = 0.5$.

Joughin et al., 2021). However, caution must be used when applying the conclusions of this paper to areas with highly three-dimensional deformations, such as curved ice streams or other areas where there is both vertical and horizontal deformation. Nevertheless, exploring general two-dimensional deformations is a good first step away from the isolated, two-dimensional conditions of pure and simple shear alone.

## 5.1 Fabric patterns across deformation regime and temperature space

Previous work has focused on modelling fabrics produced at single deformation regimes (Llorens et al., 2016), or modelling the deformation experienced by ice at a divide (Bargmann et al., 2012). Due to the computational efficiency of our model, we have been able to perform thousands of simulations across the parameter space of temperature and vorticity number to show how fabrics vary.

Our work generally shows a smooth transition between the two deformation regimes of pure and simple shear, as can be seen in Fig. 8 and agreeing with the high temperature experiments of Kamb (1972). Such intermediate deformation regimes are important in the real-world (Figs. 3,4,5) and must be taken into account when considering ice fabrics, rather than focusing on the isolated cases of pure and simple shear.

The weak fabric seen for highly rotational flows had not previously been studied at all to date. This result is interesting as it reveals for the the first time the fabric produced by rotational deformation regimes: a weak girdle fabric. Throughout these deformation regimes we have kept the magnitude of $\mathbf{D}$, the strain-rate tensor, constant. Therefore, at high vorticity numbers the weak fabric seen in Figs. 8 and 9 is not caused by a lack of deformation. Instead it is due to the rotational component acting to quickly smear any cluster produced by basal-slip deformation or migration recrystallization to orientations where the cluster is consumed by migration recrystallization. The end result as $\mathcal{W} \to \infty$ is a very weak girdle fabric with the girdle aligned to the axis of vorticity.

The regime diagram in Fig. 10 shows the fabric patterns produced across the space of vorticity number, temperature and strain. From experimentally deformed ice, the vast majority of fabrics produced are girdle fabrics (Fan et al., 2020). However, ice deformed in the laboratory in compression can only reach strains of up to $\gamma = 0.5$ and Fig. 10 highlights how double-maximum fabrics - the two-dimensional equivalent of a girdle - are only present up to these strains. Above $\gamma = 0.5$, secondary-cluster and single-maximum patterns are more prevalent. Importantly, as suggested by Kamb (1972), secondary cluster fabrics, which are commonly only considered in simple shear ($\mathcal{W} = 1$), can occur at very low vorticity numbers (Fig. 10c,d).

Experimental results from Budd et al. (2013) show a secondary-cluster fabric for $W = 0.85, \gamma = 0.34$ and $T = -2°$ C. Extrapolating Fig. 10a would suggest this agrees with our model. Budd et al. (2013) also have another laboratory data point at $\mathcal{W} = 0.52, \gamma = 0.72$ and $T = -2°$ C which shows a single-maximum pattern. Visual extrapolation of the regimes in Fig. 10 would predict a secondary-cluster fabric. However, it is close to the parametric boundary where the fabric switches to becoming a single-maximum fabric. It should also be noted SpecCAF has only been constrained up to $T = -5°$ C, and, as ice approaches the melting point, the parameters are likely to depend non-linearly on temperature, so a simple extrapolation cannot be relied upon.

## 5.2 Finite strains required for fabric evolution

For the interpretation of ice fabrics, it is essential that we know the timescale (or in the non-dimensional case here, total finite strain) over which fabrics evolve to steady state. In this paper we have presented the first assessment of fabric timescales (i.e. strains), by examining the 'half-life' for fabrics to reach steady state (Fig. 13). In compression, experiments can only reach

a maximum effective strain of $\gamma \approx 0.5$. It is often assumed that a strain of around $\gamma = 0.2$ represents a steady state in the mechanical properties (e.g. Fan et al., 2020). However, recent experiments show that the fabric continues to evolve past this (Piazolo et al., 2013; Qi et al., 2017). It is also known that fabrics require higher finite strain to reach steady state in simple shear (Journaux et al., 2019). Qi et al. (2017) note that fabric data is required at higher strains than has been achieved to date by compression experiments to link to high-strain natural environments. The results in our paper fill this missing gap. The analysis of fabric regimes extends to very high strains of $\gamma = 10$. For low $\mathcal{W}$, in experiments and in Fig. 8, a fabric quickly develops at very low strain $\gamma \approx 0.2$, (also seen in Craw et al., 2018). Although the fabric pattern does not change as strain increases, it cannot be said to be in steady state because the concentration of orientation at the clusters continues to increase. It should be noted that any change in fabric intensity will directly affect the mechanical properties such as the degree of viscous and seismic anisotropy (Duval et al., 1983; Matsuoka et al., 2003).

Figure 13a shows that, for very low temperatures ($T \approx -30°$ C) the strain to reach halfway to steady state is the highest, around $\gamma = 0.5$. As can be seen in Fig. 13c the true steady-state can not be reached to much later as the $J$ index approaches the steady-state value slowly. From this we can remark that reaching true steady states at these temperatures in laboratory experiments will be impossible for deformations close to pure shear. Although Fig. 13a shows that, above very low temperatures ($T > -24°$ C), deformation regimes closer to simple shear take longer to reach steady state. However these strains are more achievable in the laboratory as fabrics can be deformed in torsion, allowing very high strains such as $\gamma > 1.5$ to be reached (Journaux et al., 2019). Furthermore, the strain to reach halfway to steady state is primarily dependent on vorticity number $\mathcal{W}$, rather than temperature.

## 5.3 Consequences for ice core interpretation

Analysis and interpretation of ice cores remain key for understanding the processes occurring in the natural world, for both understanding the past climate history (Dansgaard et al., 1969) as well as understanding ice sheet dynamics (Schytt, 1958). The regime diagrams we have constructed (Figs. 10,11) can be used as a toolkit to interpret ice cores. For example a single-maximum fabric with an angle less than $20°$ between the compression axis and primary cluster centre implies the core has undergone mostly compression at low temperature. If other constraints such as knowledge the deformation regime history or temperature has been constant to good approximation, the dominant deformation regime and temperature can be further constrained. As our work here is for constant temperature and vorticity number, any ice core fabric interpretation will inherently assume that an ice core that has been deformed primarily at a dominant temperature and deformation regime over its recent history. Since temperature varies with depth in an ice sheet (Paterson, 1999), this method is likely to be most reliable for ice cores where the ice is primarily moving horizontally i.e. far from ice divides. We have also assumed an initially uniformly distributed orientation for the fabric (corresponding to initially randomly distributed orientations). Although deformation regime history in the natural world is likely to be complex, this is a reasonable assumption because ice formed from surface accumulation will initially have a random distribution of orientations (Montagnat et al., 2020).

The fabric pattern is a robust way to interpret ice cores, as it requires no assumption about the deformation regime direction. The regime diagram in Fig. 10, which is based on the fabric pattern only, is complex but allows insights to be drawn. For exam-

ple, the presence of double-maximum fabrics (two equal strength clusters) implies that the fabric has undergone a relatively low strain. A secondary cluster fabric implies that the fabric is likely to be at intermediate strains, and at temperatures $T > -20°\,\mathrm{C}$. A very weak secondary cluster fabric implies a rotational deformation regime $W > 3$. Furthermore, we have shown that the presence of a fabric that appears to be isotropic could be indicative not only of no deformation, but also of a highly rotational

deformation regime.

  If other information about the fabric history is known, Figs. 10 and 11 can be used in combination to extend this knowledge. For example, if there are independent constraints on the orientation of the deformation regime axis, then the angle of the primary cluster can be used to interpret ice cores as well. As can be seen in Fig. 11 the angle between the primary cluster and the compression axis is relatively invariant with strain. Knowing this angle can therefore give a good estimate of the vorticity

number and temperature.

### 5.4 Implications for ice flow properties and modelling

Viscous anisotropy of ice is controlled by the fabric and is a key control of the flow field (e.g. Alley, 1988). This anisotropy is dependent on the pattern, direction and strength of the fabric. The current approach common in ice-sheet models is to represent anisotropy with an enhancement factor that scales the viscosity, either globally (Graham et al., 2018) or locally (Placidi et al.,

2010). The model SpecCAF used in our paper can, in principle, be coupled with any anisotropic viscosity formulation to include directional variation in viscosity. Martín et al. (2009) has coupled a fabric model to an anisotropic viscosity, but the fabric evolution model used did not include recrystallization, which is key processes in controlling fabric evolution and its approach to final steady state. Their model also neglects the considerable effect of temperature dependence. The high sensitivity of fabric strength and patterns to temperature shown in our paper, in agreement with experiments (Qi et al., 2019), may lead

to further interesting flow features on top of those caused by anisotropy alone. This can only be captured with a coupled fabric model including a temperature dependent fabric. Temperature can also affect the flow through viscous heating (e.g Hindmarsh, 2004) and a temperature dependent fabric model coupled to anisotropic viscosity such as Gillet-Chaulet et al. (2005) would allow us to understand what proportion of the effects of temperature are a consequence of viscous heating and what proportion arise through temperature induced changes to the fabric.

Our analysis of fabrics here can also give insight into where anisotropy may be most important in an ice sheet. Areas with strong fabrics will be highly anisotropic, with the viscosity varying in different directions. Anisotropic flow is not well studied but initial simulations with coupled anisotropic flow show that it can explain hitherto unexplained observations such as syncline patterns observed under ice divides (Martín et al., 2009). Our analysis of fabrics produced in highly rotational deformation regimes showing an almost isotropic fabric (as seen in Figs. 8 and 13b) implies that in the regions of Antarctica

where the flow is highly rotational (i.e. those highlighted in Fig. 5), the fabric will evolve towards a state with limited anisotropy (directional variation in viscosity). However areas of approximately simple shear ($\mathcal{W} = 1$) show the strongest fabrics and hence the strongest effect of anisotropy. This means viscous anisotropic affects are expected to be widespread, further motivating the need to fully represent them in models. In future work it would be possible to explore how seismic wave velocities vary with the different fabrics modelled in this paper, again comparing to real world observations (e.g. Smith et al., 2017).

## 6 Conclusions

Accurately predicting ice fabric evolution is pivotal for the correct interpretation of ice core fabrics as well as the reliable prediction of ice losses in a changing climate. Our work extends the ability to predict fabric evolution from the deformation regimes of pure and simple shear to general two-dimensional deformation regimes. This represents a step towards understanding fabrics in fully general conditions: key for understanding viscous anisotropy and, in turn, large-scale ice-sheet flow modelling. We have shown that deformation regimes outside of pure and simple shear are important in common flow scenarios seen in ice sheets. Future work could use the modelled fabric to predict seismic properties, to compare to real world observations.

The regime diagrams presented are a useful tool to help with the interpretation of ice core data. In combination with other information, such as the plane of deformation regime or an estimate of the temperature at which a core was deformed, these regime diagrams can be used to determine the primary deformation regime and temperature undergone by an ice core. We show that for two-dimensional deformations, the double-maximum fabric is not present at high strains when only a small amount of vorticity is present in the deformation regime $\mathcal{W} > 0.1$. This is important as many laboratory experiments are performed for $\mathcal{W} = 0$. Future work could investigate whether this conclusion extends to three-dimensional girdle fabrics.

Highly rotational deformation regimes were investigated for the first time and showed a weak girdle fabric aligned to the axis of vorticity. We have also shown how the timescale for fabric evolution, shown by the halfway strain to reach steady state, changes over the parameter space. Laboratory experiments around simple shear may be able to reach the strains required to get close to steady state, however compression experiments, especially at low temperatures, cannot achieve the required strains for steady state.

Our predictions of ice fabric evolution over a wide range of deformation regimes provide insights into how and where viscous anisotropy will be important for ice-flow dynamics. Intermediate deformation regimes between pure and simple shear produce the strongest fabrics, suggesting anisotropy will be most important in these regions. Similarly, as highly rotational deformation regimes produce a weak fabric, there is likely to be a less dominant affect of anisotropy in such regions. Our understanding of these issues could be further improved by combining our model with an anisotropic viscosity formulation to model the coupled fully anisotropic flow of ice is an important future step for accurately predicting ice flow.

*Code availability.* A python implementation of SpecCAF, alongside code to reproduce the figures in the Results section, is available at http://doi.org/10.5281/zenodo.7086132

*Author contributions.* All authors designed the research and edited the manuscript. DR developed the model code, performed the analysis and visualization and wrote the draft.

*Competing interests.* The authors declare that they have no conflict of interest.

*Acknowledgements.* We thank Sergio Faria for his email correspondence and review which helped to improve this manuscript. We also thank

Ed Waddington, Maurine Montagnat, Fabien Gillet-Chaulet, and one anonymous reviewers for their helpful and insightful reviews which also improved the manuscript. Finally we thank Kaitlin Keegan for her editorial handling. This worked was funded by the UK Engineering and Physical Sciences Research Council (EPSRC) grant EP/L01615X/1 for the University of Leeds Centre for Doctoral Training in Fluid Dynamics.

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
