# Peer review of "Ice fabrics in two-dimensional flows: beyond pure and simple shear"

_The Cryosphere, 2021_

## Referee Comment (RC3)

**General comments**

The authors have used a numerical model to explore changing CPO orientation and strength in ice deforming in flow regimes intermediate between pure and simple shear and with varying degrees of vorticity. They are able to present detailed data representing ice fabrics at much higher strains and under a much wider range of conditions than it is possible to achieve with laboratory experiments, which is an important contribution to the field. Their examination of steady-state fabrics at high strains is particularly valuable.

The manuscript could be made more impactful by adding more thorough comparisons between model results and experimental results from the cited literature. Some more clarity on the model setup would also be useful.

**Specific comments**

**l.30:** This statement (that looking at the c-axis alone is sufficient) needs some justification. Chauve et al. [2017] have found that non-basal slip systems are very significant at high homologous temperatures. Are you assuming that at lower temperatures they are no longer significant? Please clarify.

**l.50:** Jun et al. [1996], and Budd et al. [2013] also performed experiments with a combination of shear and compression.

**l.76, and Fig.1 caption:** There are referenced experimental examples for all of these fabrics aside from pure shear. I suggest citing Kamb [1972], or a more modern reference if one exists (I'm not aware of any).

**l.113:** I'm unsure what $W > \infty$ means. What is more rotational than pure rotation? Is there a section of the Ross Ice Shelf that is continually spinning in circles? Please make this clearer for easily confused readers like me.

**Section 2.4:** This is almost a rewording of your stated research questions in lines 18-25, but not in an obvious way. It would be clearer to refer more explicitly to which question you intend to answer in which section. As I've interpreted it, you've addressed your first question (which deformations are present in the natural world) in section 2.2.2, and the following two (how fabrics evolve, and how steady-state depends on temperature and deformation) will be addressed in the following sections. In that case, it would be clearer for the reader if you reiterate the second two questions here with similar wording as in the introduction, and do the same with the first question where you address it above.

**Section 3:** I am not entirely clear on what is being physically represented in the model. E.g., are non-basal slip systems being incorporated, even just by an empirical parameter? How are grain boundary interactions being represented, if at all? To find the answers, the reader must decipher a page of equations. I'm aware this has been covered in more detail in Richards et al. [2021], but a brief explanation at the beginning of your methods section referring back to section 2.1.1 and stating how the specific mechanisms you have described are represented in your model would be very useful.

**Figure 5:** It would be comforting for an experimentalist reading this paper to see some real experimental results alongside the model results, to show the agreement between the two on the presence of double clusters, and their strength. Fig. 5 shows your results from a simulation of simple shear at $-5°\,\mathrm{C}$, and this is a scenario which has been tested experimentally by Qi et al. [2019]. Perhaps you could take some results from Qi et al, where they have provided detailed c-axis plots and J-indices, and present them alongside your model data plotted in a similar fashion. Again, I know this comparison appears in some capacity in Richards et al. [2021], however it would make this current paper much more convincing to see specifically the agreement on double clusters.

**Fig. 8 and Fig. 9:** What is the resolution of the data used to make these contour plots (i.e., how many $W$ and T values were tested)? This information would be useful for interpreting parts of the figure. E.g., in Fig. 9c, it would be good to know how many data points make up the "wiggles" between the single cluster and secondary cluster zones at high $W$.

**Section 5.3:** The observation made earlier in the text, that highly rotational fabrics can produce a CPO which would appear isotropic when sampled at the resolution of most commonly used techniques, is important for ice core interpretation. It should be mentioned in this section.

**l. 321** It is unclear what is meant by "other constraints". Other data on strain history and temperature in the area?

**l. 325:** "We also assume an initial random orientation for the fabric". Although I assumed that was the case, this information would have been useful to know much earlier. If it was explicitly stated above, I missed it!

**l. 326:** It is true that ice formed from accumulation will have an initially random fabric, but as you say the fabric can adjust to the stress conditions very quickly (within strain of 0.2). It would be interesting to know if a pre-existing fabric affects the evolution of a CPO once the stress conditions change. That's a completely separate paper of course, but worth pointing out as an area of future research...

**Section 6:** I think there are two important findings mentioned in the text which have not found their way to the conclusions: firstly, that the $W=0$ case which is very commonly found in experiments results in a CPO which is different to that found if there is even a small amount of rotation, implying that the most common experimental scenario is not representative of most real scenarios (line 312). Secondly, that the double-cluster/cone CPO which we see so often in experiments is not present in your results in steady-state, and appears not to persist beyond the highest strains which we can reach with experiments (line 234). These deserve a brief mention here.

**Technical corrections**

**Fig. 5 caption:** "which", not "witch"

**l.274** should read "fabric patterns"

**l.50** "strain" rate response

**References**

W. F. Budd, R. C. Warner, T. H. Jacka, J. Li, and A. Treverrow. Ice flow relations for stress and strain-rate components from combined shear and compression laboratory experiments. *Journal of Glaciology*, 59(214):374–392, 2013. ISSN 00221430. doi: 10.3189/2013JoG12J106.

T. Chauve, M. Montagnat, S. Piazolo, B. Journaux, J. Wheeler, F. Barou, D. Mainprice, and A. Tommasi. Non-basal dislocations should be accounted for in simulating ice mass flow. *Earth and Planetary Science Letters*, 473:247–255, 2017. ISSN 0012-821X. doi: 10.1016/j.epsl.2017.06.020. URL http://dx.doi.org/10.1016/j.epsl.2017.06.020.

L. Jun, T. H. Jacka, and W. F. Budd. Deformation rates in combined compression and shear for ice which is initially isotropic and after the development of strong anisotropy. *Annals of Glaciology*, 23(May 2021), 1996. ISSN 02603055. doi: 10.1017/s0260305500013501.

B. Kamb. Experimental Recrystallization of Ice Under Stress. *Flow and Fracture of Rocks, American Geophysical Union Geophysical Monograph*, 16:211–241, 1972. doi: 10.1029/gm016p0211.

C. Qi, D. J. Prior, L. Craw, S. Fan, M. G. Llorens, A. Griera, M. Negrini, P. D. Bons, and D. L. Goldsby. Crystallographic preferred orientations of ice deformed in direct-shear experiments at low temperatures. *Cryosphere*, 13(1):351–371, 2019. ISSN 19940424. doi: 10.5194/tc-13-351-2019.

D. H. Richards, S. S. Pegler, S. Piazolo, and O. G. Harlen. The evolution of ice fabrics: A continuum modelling approach validated against laboratory experiments. *Earth and Planetary Science Letters*, 556:116718, 2021. ISSN 0012821X. doi: 10.1016/j.epsl.2020.116718. URL https://doi.org/10.1016/j.epsl.2020.116718.

---

## Referee Comment (RC4)

This paper present an application of specCAF, a numerical model of fabric development based on a continuum theory by Faria and Placidi, and described in Richards et al. (2021). Compared to previous works on ice fabric evolution, this paper discuss the fabric patterns obtained for a wide range of vorticity numbers, including highly rotational flows, using synthetical 2D experiments. To justify this approach, the authors have computed the vorticity number from observed horizontal surface velocities in Antarctica.

They obtain big (>1) vorticity numbers in large portions of the ice-shelves with curved stream lines, and a conclusion of the paper is that in such regime the fabric should remain nearly isotropic.

My main comment, is that I remain very sceptical about this conclusion and the interpretations of the results for fabrics in natural flow. The authors claim that most previous studies have focused on pure and simple shear, this is true, but they forgot to mention that the justification is that something between pure shear and uni-axial compression in the **« vertical «** direction is supposed to dominate in the upper ice layers while simple shear (**parallel to the bed**) is supposed to dominate in the lowest layers, at least in the central parts of the ice sheets where ice cores have been drilled and direct fabric observations are available. It's not clear from section 2.2 how the spin and strain – rate tensors are computed for the observed Antarctic horizontal surface velocities ? It is assuming plane strain in the horizontal plane ? I don't think that an horizontal 2D plane strain would be a good approximation of the natural conditions in ice shelves. I still would expect to have a compression component in the vertical direction, so the interpretation of the results presented here in term of fabrics in natural conditions need better justifications.

I read the comments from the other reviewers and the author responses. The debate between Gagliardini and Faria has not really been clarified and I think that this papers could be a good opportunity to clarify the assumptions behind the continuum approach and how it compares with homogenisation models. Two points seems to require clarifications.

First, the classical approach in ice flows model is to solve the Stokes equations (or some shallow approximations) for a given flow law, i.e. a relation between the macroscopic strain-rates and stresses, that are then solution of the problem. It is not clear here how such a relation could be obtained from specCAF. Faria (2006a,b) gives some homogenisation rules to compute the macroscopic stresses, but it seems that is has never really been used. Instead Seddik and others (2008, 2011), using the CAFFE model, parameterized an « enhencement » factor as a function of the polycrystal deformability that depends on the fabric. Using the same argument as for the strain rates, i.e. the volume contains an infinitely large number of grains, Seddik and others (2008) claim that the stress tensor do not depend on the orientation. So it is not clear, (i) how both the stresses and strain-rates at the level of the species (i.e. using Faria's terminology in is reply to Gagliardini) can be equal to the macroscopic equivalent, but still with a viscosity tensor that would depend on the orientation, and (ii) if the macroscopic stresses computed this way would be solution of the continuum model, i.e. the balance equations that are derived in Faria's papers?

Second, an anisotropic model must be able to describe how the fabric evolves. Here, the model includes several processes, including rotation of the ice crystals due to basal-slip deformation. The equation used to take into account this effect (Eq. 5) at the scale of the species in the continuum approach, is based on equations that have been derived for single crystals. According to the description of their model (Richards et al., 2021) : *« If this equation is applied to an individual grain, it describes the c-axis rotation rate (Gödert and Hutter, 1998; Svendsen and Hutter, 1996) under the Taylor hypothesis (neglecting grain-grain interactions). However, since we are using a continuum model that assumes a large number of grains within any solid angle of orientation, any grain-grain interactions are smeared-out (Faria et al., 2008). In this continuum model, we do not therefore require the Taylor hypothesis. »,* From that I understand that the continuum approach would give a fabric evolution similar to an homogenisation model that uses the Taylor hypothesis ? So maybe, strictly speaking the continuum model do not use the Taylor hypothesis because it does not have grains, but at the end the equations that are

used for the species (i.e. the orientations) come from single crystals models ? As the model has been calibrated against experiments, this could potentially affect the interpretation of the relative contributions of the different recrystallisation mechanisms that are included in the model ?

I have few other detailed comments listed below :
- Sec. 2.2 : see my main comment, the procedure to compute the vorticity number needs to be better explained and justified especially if it's only done in 2D. Ice is incompressible, so tr(D) must be zero is this enforced? Also it's not clear of me on which length scale the velocity gradients are computed, directly using a finite difference from the original grid resolution?
- Sec 2.3 : *« At the other end of the scale, models such as presented by Gillet-Chaulet et al. (2005) track the evolution of tensorial descriptions of the fabric, without including migration recrystallization. These cannot accurately reproduce detailed fabric patterns but are computationally cheap enough for integration into large-scale models (Gagliardini et al., 2013). »*. Gillet-Chaulet et al. (2005) only present the flow relation, i.e. the anisotropic tensorial relation between the macroscopic stresses and strain-rates, so there is no fabric evolution at all. The equations for the fabric evolutions are presented in Gillet-Chaulet et al. (2006). The fact that it do not includes migration recrystallisation is not a limitation of the procedure itself. Seddik et al. (2011) also derive an equation for the evolution of the orientation tensors from the CAFFE model ; so in principle migration recrystallisation, as it is represented here, could be included within the same framework.
- Sec. 3.2 : explain what is *y* here and in Fig. 5 and what are the deformation principal axes with respect to this reference frame for the pole figures.
- Sec. 3.2 : give the expression for the computation of the strain (\gamma) from the strain-rates.
- Page 9, last line : *« Furthermore the pre-factor »,* I'm not such which pre-factor ?
- Fig. 5 : Maybe the schema for the single maxima is a bit misleading at it shows a single maxima in the vertical direction, while it is directed at 45 degrees.
- Line 209 : give the definition of the J-index before using it.
- Sec. 5.4 : *« The model SpecCAF used in our paper can be coupled with an anisotropic viscosity formulation to include directional variation in viscosity »*. Provide more details on the exact procedure, i.e. how the stresses are computed with SpecCAF, and the assumptions that would be required for this step.
- Sec. 5.4 : *« This has been done with simplified fabric evolution models which do not include recrystallization and temperature dependence (Martin et al., 2009). »* This gives the impression that Martin et al. use the continuum model while they are using an homogenisation model with the static (uniform stresses) assumption. Also, from the CAFFE model, Seddik et al. (2008,2011) derive an anisotropic flow law where stresses and strain-rates remain colinear. So if the same method is used here (depending oon the previous comment), it is not so clear that this model would also produce the syncline patterns in the isochrones that are mentioned few lines latter.

References:
- Faria, S.H., 2006. Creep and recrystallization of large polycrystalline masses. I. General continuum theory. Proc. R. Soc. A. 462, 1493–1514. https://doi.org/10.1098/rspa.2005.1610
- Faria, S.H., 2006. Creep and recrystallization of large polycrystalline masses. III. Continuum theory of ice sheets. Proc. R. Soc. A. 462, 2797–2816. https://doi.org/10.1098/rspa.2006.1698

- Gillet-Chaulet F., O. Gagliardini, J. Meyssonnier, T. Zwinger, J. Ruokolainen, 2006. *Flow-induced anisotropy in polar ice and related ice-sheet flow modelling*, J. Non-Newtonian Fluid Mech. **134**,
- Seddik H., R. Greve, T. Zwinger and L. Placidi, 2011. *A full-Stokes ice flow model for the vicinity of Dome Fuji, Antarctica, with induced anisotropy and fabric evolution*, The Cryosphere, **5**, 495-508

---

## Author Response (AR1)

**1st review**

We again thank the reviewer for her helpful and thorough review of the manuscript, especially her suggestion of a parameter sensitivity investigation. We have implemented all these changes, detailed below in purple, and believe the manuscript is much better now thanks to these changes. We have added these purple comments to our original (blue) "reply to reviewers comments" to show where we have made changes in the manuscript.

This paper presents some simulations of ice fabrics in conditions relevant for the Antarctic ice sheet. The simulations are made by means of a numerical model inspired by the work of Placidi et al. (2010), that simulates the rotation of individual ice crystals included in a orientation mixture that is submitted to a given strain field, and includes a parametrisation of the effect of dynamic recrystallization on this rotation. This model has been applied recently in Richards et al. 2020 (EPSL) to reproduce laboratory observations.

1. We wish to thank the reviewer for her thorough review of the manuscript and numerous helpful comments. We understand the rationale behind the comments raised by the referee and provide responses to each below. In a number of instances relating to the nature of the model, we acknowledge that more explanation in the present paper is warranted and should be incorporated in a revised version. We also argue here that certain concerns raised by the reviewer are not rendering the model as invalid or inappropriate for the first order analysis and prediction of ice fabrics that this contribution is focussed on. We also assure the reviewer that the model has been tested by direct comparison with experiments, propose more quantification of confidence in the model, and clarify the motivation for focusing here on 2D flow.

Please find our comment-by-comment responses below. To summarise our responses:

a. We agree that all assumptions underlying our model need to be clearly stated. This can be satisfactorily addressed through added discussion in the paper and through reference to the existing experimental validation in Richards et al. 2021. In particular, we emphasise the validity of the assumptions is demonstrated by reproduction of experimentally produced fabrics (Richards et al. 2021). Please see below for details. **Discussion of the assumptions has been added extensively in section 2.1.1 which is almost entirely new.**

b. We do not agree with the referee's comments that the model makes a Taylor type assumption. This has been discussed before in Faria (2008). The model does not attempt to simulate individual ice crystals, only the evolution of the orientation distribution function. However, we acknowledge that this discussion is important and easily addressed in revision. **Clarification of this has been added in new subsection 3.1**

c. We agree that a focus on 2D deformations with constant deformation history, limits direct application of the simulations provided to a complete interpretation of fabrics retrieved from ice cores, and this was not our intention to convey. Nonetheless the presented results provide a necessary stepping-stone towards such an application. In a revised manuscript we will emphasise this notion in the motivation for our work, highlighting the importance of analysing fabrics in more complex conditions in the future. The primary motivation of this paper is instead to use the already validated fabric model to take a first step away from the isolated conditions of pure and simple shear, to identify new properties of fabrics occurring continuously across linear space between them (as well as to rotational deformations, which lie on the same spectrum indexed by

the vorticity number). While the model can accommodate general 3D and changing deformation/temperature history, this is beyond the intended scope of the solutions we intend to present in the current paper, as the parameter space is too large to explore within the scope of a single paper, yet can be incorporated and explored in subsequent work. To clarify this, we propose changing the title to 'Ice fabrics in **two-dimensional** flows: beyond pure and simple shear' alongside more clarification in the text of the reasons for beginning with 2D deformation, and also the rationale for considering surface velocities of Antarctic for basic motivation (see below). **Alongside changing the title, we have added 2 new figures, showing the vorticity number from a 2D vertical cross section of ice at a divide (Fig 3) and over a bump (Fig 4) respectively. Based on these new figures, section 2.2.2 is almost entirely new to highlight that the investigation presented in our manuscript can be motivated by the range of 2D deformation regimes occuring through the depth of a vertical cross-section of ice. In addition, we are now explicit that we are limiting our analysis to 2D deformations (as shown by the title)**

This paper suffers from a lack of clear explanation of the strong assumptions that are included in the numerical simulations and the associated parametrisation.

Such assumptions, that I detail below, can have a significant impact on the results, and, since they are not clearly stated, they are not tested either, and this undermines the credibility of the study.

2. The model was presented, calibrated and tested in Richards et al. (2021) (EPSL) in direct comparison with laboratory experiments. This provides validation of the approach and exhibits predictions (such as secondary clusters observed in simple shear) that have not been successfully predicted even by previous detailed microstructural models. In a revised version of the manuscript, this fact will be emphasised more explicitly.

**We have now emphasised this in section 3.1, around line 233.**

- It would be first necessary to recall the way the strain and stress interactions between grains are dealt with in the model.

3. It should be noted that the explicit modelling of grains or grain-grain interactions is not applicable to the model used in this contribution. The rationale of the continuum model is mathematically similar to the Navier-Stokes equations, which do not attempt to represent the motion of fluid particles, but instead describe the spatial average of a bulk of quantities describing them at a larger scale; this is the basis for all continuum approaches, and we are applying here the same principles for fabric modelling – in this regard the model does not neglect the microstructural interactions per se (e.g. it does not make any assumption of uniform distributions of stresses at the microscale, see below) because their mean emergent bulk effects are encapsulated by the model parameters that we have rigorously constrained empirically through direct comparison with laboratory data. All parameterisations in the model are formulated to represent the change in the ODF (orientation density function representing the fabric), not specific individual grain behaviour.

The success of our validation against experiments, including its ability to reproduce fabric structures that have not been predicted even by complex discrete models, shows clearly that the general continuum modelling approach taken is indeed justified, though we do understand that a more thorough discussion of the nature of the model and its assumptions, is helpful to include.

Again, we wish to reiterate that we do not wish to refrain from a clear explanation of the nature of the model, and we will therefore endeavour to clarify the model validation and the rational of an continuum model better in a revised version of the paper, and more clearly, alongside references to additional details and intended scope (see below).

**As mentioned above, we have now clarified the assumptions of the model in the new section 2.1.1. We have also clarified SpecCAF's place in the hierarchy of fabric models in the new section 2.3**

Unlike stated in Richards et al. 2020, the model, that derives from previous works of Faria et al. (2006-I,II,III), assumes an homogeneous strain rate, meaning that each crystal is submitted to the same strain rate. This hypothesis, apparently not clearly stated in any of those works, has been shown by Gagliardini (2008) in its response to Faria et al. (2006) to correspond to a Taylor-type of approximation, meaning uniform strain.

4. Faria (2008), in his reply to Gagliardini's comment, showed that the theory does *not* make a Taylor-type approximation. In essence, Gagliardini draws a false equivalence between averaging over grain-to-grain interactions up to polycrystal quantities in Lebensohn et al. 2004, and averaging operators over the abstract orientation space. The theory proposed in Faria et al. (2006-I,II,III) does not impose any constraint whatsoever on the deformation of individual grains. Please see Faria (2008) for a more detailed discussion of this point.

**We have now added an explicit clarification of the discussion between Gagliardini and Faria in section 3.1, paragraph 3.**

Such an approximation can be clearly recognised as such, and then it is possible to evaluate its impact on the simulation of the mechanical response of the polycrystal, as done by Castelnau et al. (1996). In particular, Castelnau et al. 1996 showed that this approximation was not satisfactory for a highly anisotropic material such as ice since it requires the activation of non-basal slip systems at a non realistic level. By doing so, it strongly reduces the level of strain heterogeneities between crystals, the latter being the main driving force for dynamic recrystallisation. We can expect this approximation to impact the modelling of this mechanism.

Since Castelnau et al. work, it appeared clear that in situation where the full stress and strain field heterogeneities can not be taken into account, an homogeneous stress approximation is more adapted to simulating the mechanical response of ice (see maybe, for instance, the work of Pettit and co-authors).

In Richards et al. 2020, it is mentioned that the fact that the model considers a large number of grains for each orientation specie, reduces (or annihilates) the dependency on the grain orientation on the mechanical state and response (strain and stress). Gagliardini (2008) showed, based on Lebensohn et al. 2004 work, that this is not true and that only the dependency on the neighbourhood is reduced by considering many grains for each orientation.

5. We appreciate the comments here as they have highlighted that we need to extend the explanation of the model and include more details from Faria (2008).

**We have now added an extensive clarification of the discussion between Gagliardini and Faria in section 3.1, paragraph 3.**

- The way the dynamic recrystallization is simulated is also based on important assumptions, not always in agreement with laboratory or field observations. It would be necessary to explicitly mention these approximations, and justify their use.

6. Thank you, we agree that the explanation of assumptions should be elaborated on in revision. Justification is provided by validation against existing laboratory experiments.

**As mentioned above, we have now clarified the assumptions of the model and the nature of the continuum approach in the new section 2.1.1**

First, in the main part of ice sheets, where temperature and strain rates are low, the main recrystallization mechanisms is continuous (or rotation) recrystallization, characterized by a low driving force for grain boundary migration (see for instance De la Chapelle et al. 1998). In such a regime, the fabric is supposed to evolve only slightly owing to recrystallization, and to remain mainly dominated by deformation (see also Montagnat et al. 2012, for the Talos Dome core).

It would therefore be important to evaluate, in some appropriate locations, the relative influence of the simulated rotation recrystallization versus migration recrystallization in the obtained fabrics. If migration recrystallization, the way it is simulated here, has too much weight on the resulting fabric in location where rotation recrystallization is expected to dominate, the model can be questioned.

7. We agree, this analysis of the contribution of different recrystallization mechanisms would be useful. At low temperatures our model also predicts a fabric mainly produced through deformation, in agreement with the referee's comment. The relative importance of the recrystallization mechanisms and its implementation in the model using dimensionless parameters is explained extensively in Richards et al. 2021. In a revised version a summary will be provided.

**We have added a new parameter sensitivity analysis in the supplementary materials (discussed in more detail below) which examines the sensitivity of the results to changes in the parameters. We haven't added a specific investigation into the effect of rotation recrystallization vs migration recrystallization. Such an investigation is not in the scope of the current contribution. In addition, the effect of rotation recrystallization is to diffuse i.e. weaken the modelled fabric but is not to change the actual pattern generated.**

In areas where migration recrystallization dominates (high temperature / high strain rate), the grain boundary migration kinematic dominates the softening process, so that the fabric and microstructure end up resulting from the stress state, and loose track of the deformation history (see what happens at the bottom of the GRIP, NEEM, Dome C ice cores for instance, or also in high shear conditions, Hudleston 1977 for instance, or even Hudleston 2015, see also Alley 1992). Can we expect, in such conditions, an evolution of fabric with strain?

8. Laboratory experiments (Qi et al. 2019, Journaux et al. 2018, Craw et al. 2018, Piazolo et al. 2013), performed at high temperatures (>-10C) and very high strain-rates, clearly show an evolution of fabric with increasing strain. See Fig. 5 of Richards et al. 2021 (EPSL) for a collation of these experiments plotted against strain.

- Second, concerning the physical mechanisms. Migration recrystallisation is supposed, in the presented model, to be governed by a "deformability" related to the total deformation accumulated in the grain.

Dynamic recrystallization mechanisms (nucleation and GBM) are related to the local accumulated dislocations in the form of geometrically necessary dislocations (responsible for local misorientations), and GNDs are not correlated with the total amount of strain experienced by the grains. It has been recently shown by Harte et al. 2020 for Ni-based alloy by coupled EBSD observations and Digital Image Correlation strain measurements (stored energy is different from cumulated strain).

In various experiments performed on ice, or full-field modeling, it was shown that there is no relationship between the amount of deformation (measured by Digital Image Correlation for instance) and the Schmid factor of a grain. There is therefore no "hard grains", or "soft grains", since the local behavior is much more controlled by the grain interactions and the resulting stress redistribution. The uniform strain assumption neglects this aspect too.

9. While in the continuum model approach taken the modelling of migration recrystallization includes assumptions and simplifications, the model (as noted above) is not aiming to simulate each grain or grain to grain interactions, but rather the mean effect of migration recrystallization on the orientation distribution function. Therefore, the representation of processes in the model should not be expected to correspond to grain behaviour, but rather their bulk mesoscopic representation (as represented by the dependent variable evolved by the model, the ODF). We again highlight the fact that in Richards et al. (2021, EPSL) the model was shown to predict the distribution function of fabrics from experimental results, indicating that these assumptions are justified. The model also predicts detailed features such as secondary clusters in simple shear, which even full-field approaches such as Llorens (2016) have struggled to reproduce. This is evidence that these assumptions are justified in terms of capturing the essential effect on the distribution function. We also note that, as stated above, the uniform strain assumption on grains does not apply, and we will make sure to clarify this better in revision

**We have incorporated the above in the new sections 2.1.1 and 2.3**

My point of view concerning these approximations made relatively to dynamic recrystallization is that they can be useful and justified in the simplified numerical modeling approach used in this work. Nevertheless, it has to be clearly mentioned that they ARE approximations, and their effects should be tested.

10. In light of the referee's comments, we realise that we need to be more explicit in this paper about the assumptions of the model. Testing of the model was already conducted in the previous paper Richards et al. 2021 (EPSL), but we agree that a more thorough discussion of assumptions is warranted here and can be easily incorporated in revision.

**This is added in sections 2.1.1 and 2.3**

- The way the boundary conditions are selected is very unclear to me. Considering that fabric is being formed during deformation in depth of the ice sheet, how can a surface velocity map be representative of the in-depth flow conditions? Can the authors be clearer about that?

11. We do not aim to match deformations to the conditions of flow in ice sheets exactly, but rather to use the surface velocity data as motivation for investigation of a range of vorticity numbers away from pure and simple shear. We thank the referee for highlighting that this was unclear in the paper, and can be addressed in revision. Please see below our response to comment No. 24 for details.

**We have added 2 new figures, showing the vorticity number from a 2D vertical cross section of ice at a divide and over a bump respectively. Based on these new figures, section 2.2.2 is almost entirely new to highlight the results can be motivated by considering a 2D cross-section of ice, and that we are limiting our analysis to 2D deformations. We have also updated the surface vorticity number plot to include estimates for du/dz, dv/dz from the SIA approximation 25% into the ice-sheet, so the vorticity number shown in (now Fig 5) can now be said to be estimating the fully 3D vorticity number near the surface of the ice-sheet. Nevertheless, we have updated the manuscript to be explicit that our analysis is primarily motivated to help with the analysis of 2D cross sections through ice (as shown on Figures 3 and 4).**

The 2D approximation is also strong. It was shown by the Elmer-Ice community to be OK in the case of specific types of flow, like divides (where there is little divergence or convergence). Can it holds for more complex situations such as fast ice streams? What effect could it produce on the fabric evolution? This should be justified and tested.

12. We are using the 2D approximation only as a stepping-stone to explore new fabric patterns and features beyond and intermediate to 'pure shear' and 'simple shear' (and to rotational deformations, which lie on the same spectrum). This is a deliberate choice for the scope of the present paper as a focus on a well-defined continuous space of fabrics indexed by a single parameter $W$ (the vorticity number) and temperature $T$. In principle the model could be extended to more general deformations, but this is not the aim of this contribution, and would require more parameters to classify (e.g. an extra parameter representing the relative importance of vertical shear would be a natural next step). In this regard, two-dimensionality is not an approximation or limitation, but a focus to allow systematic and controlled exploration of a new research question as an initial step in the exploration of ice fabric evolutions. We appreciate that the title of the paper and the abstract may have suggested otherwise. Considering the comments by the reviewer we propose softening these statements, and to incorporate the words "two-dimensional" into the title.

**We have updated the title to Ice fabrics in *two-dimensional* flows: beyond pure and simple shear. We have also included 2 new figures showing the vorticity number through a 2D vertical cross section of ice at a divide and over a bump respectively, showing how a continuous variety of vorticity numbers appears through the depth of an ice sheet. Based on these new figures, section 2.2.2 is entirely new to highlight the results can be motivated by considering a 2D cross-section of ice, and that we are limiting our analysis to 2D deformations.**

- What "highly-rotational" conditions represent "in reality"? Does that correspond to area where a block of ice rotates freely on itself? Can that happen in the depth of ice sheets? If yes, where?

13. An ice block rotating freely on itself corresponds to a vorticity number of infinity. Vorticity numbers greater than 1 are in general common, and the surface velocity data is a way of illustrating that. As an example, for flow around a cylinder the vorticity number will be greater than 1 in the region directly above and below the cylinder, and this situation is typical for flows involving obstructions and junctions. The identification of the essential form of fabric arising in this limit is a novel result of the present work.

- About the capacity of the model to predict steady-state fabrics. Steady-state fabrics depend strongly on the mechanical state the ice is experiencing, and the flow history. I therefore don't understand how could the model be realistically predictive considering the strong assumptions made (1) on the mechanical state (Taylor-type of approximation) and (2) on the recrystallization mechanisms.

14. As explained in our responses no. 3 and 4., the Taylor type approximation does not apply. Furthermore, the model is validated against experimental results. **See the response in purple just below.** In order to test the predictability of the model, it would be necessary to test how robust it is to variations in the parameters, and to the 2D approximation, and to the use of surface velocity vorticity. Such a robustness test was already missing in Richards et al. 2020.

15. We agree that this would be a useful addition to this paper, and we are currently working on this and will add this into the discussion soon, with a view towards adding this as a supplement. It is worth noting that with the results, especially the steady-state analysis, we are not seeking to draw conclusions based on precises values but on the general pattern and change with vorticity number and temperature, and we would not expect this to change with variations in the parameters.

**We have added this parameter sensitivity as a supplement. To do this we have taken the parameter fit from the inversion from both compression and simple shear in Richards et al. 2020 (rather than just simple shear as before) and calculated the 80% and 95% confidence intervals around this – shown in a new Fig 6. We have then reproduced the figures with the strongest possible fabric (maximum basal-slip deformation and migration recrystallization parameters, minimum rotation recrystallization parameter) and weakest possible fabric. This is then used to reproduce all the main figures in the results. We thank the reviewer for this suggestion.**

**In light of the parameter sensitivity investigation we decided to slightly modify our the focus of our results relating to the strain to reach steady-states. Rather than plotting steady-state time based on 90% of the convergence, which we found to be sensitive to parameter variations due to their effect on the tight criterion at which a steady state is reached, we instead report the halfway-strain to reach steady-state. This can be thought of as a half-life for fabric evolution, as explained in section 4.3.**

**Specific comments:**

- Abstract: "a definitive classification of all fabric patterns". This sentence lacks humility... in particular owing to the lack of clarity of the text regarding the assumptions made (see my comments above), and their effects on the obtained simulation results. On top of that, the 2D simulations highly limits the ability to provide this full classification, and also the fact that strain states were deduced from surface observations, very likely not relevant for flow in the depth of the ice sheet.

16. As mentioned in our introductory statement, based on the reviewers comments we appreciate that the limitation to 2D deformations limits the applicability to general ice cores. Therefore, we propose changing the title to 'Ice fabrics in two-dimensional flows: beyond pure and simple shear', rewording this sentence, and clarifying the immediate caveats towards any direct application to interpretation of ice cores in our discussion.

**We have updated the abstract to remove this sentence and highlight our focus on 2d deformations.**

"Highly-rotational fabrics... produce a weak fabric". Can we expect a fabric to produce a fabric? Not clear to me.

17. We apologise for this typo. It should read "Highly-rotational deformations... produce a weak fabric".

**We have corrected this**

- Part 2.1: The presentation of the processes made in this part is simplistic regarding the many other observations and analyses that exist in the literature (see my comments above). It is OK if it is clearly presented as assumptions made to simplify the processes and better introduce them into the modeling approach. It is a very classical approach to simplify the physics in order to be able to take it into account in a modeling approach. But it needs therefore to be clearly stated, justified, and tested when the results are presented.

18. We agree with the reviewer that more explanation of the model, and the underlying theory, would be helpful, and we are happy to address this in detail in revision.

**We have added this in section 2.1.1**

What is the "real situation" responsible for some "rigid-body rotation"?

19. See our response no. 13, in general this arises from any vorticity in the flow-field.

- Part 2.2: Various studies were done in the past that include torsion and compression, or shear and compression, and therefore that consider a more complex scheme that pure or simple shear. None of them are mentioned in part 2. I can suggest Budd et al. (2013), Duval 1981 for instance, but others are mentioned in Hudleston 2015.

20. We thank the reviewer for mentioning these papers which we will include in the literature review (in the introductory part as well in appropriate locations in the discussion section.

**We have added reference to these papers (also Jun et al. 1996) in section 2.1.2, including reviewing their results (paragraph 3 of this section). We have also compared the fabric patterns of Budd et al. 2013 to those predicted by Fig 11 in section 5.1 paragraph 5, with general agreement.**

At domes, in fact close to domes since deep ice cores are never exactly at the dome location, if girdle is observed it is that not only compression occurs, but also lateral extension. This can signify that the core was cored slightly on the flank, or that dome has moved with time (see for instance NEEM, Vostok, EDML, NorthGRIP). For nearly every deep ice core drilled close to a dome, a shear component was observed close to the bedrock, that participated to strengthen the single-max fabric (see for instance Talos Dome).

21. Thank you for this insight which can be used as an alternative motivation for exploring conditions away from pure and simple shear. We aim to use this paper to provide a clear exploration of these conditions, like laboratory experiments provide for compression/simple shear.

**Interestingly, this can be clearly seen in our new Fig 3. where the region where W=0 is only seen very close to the divide.**

Can we consider ice deep in the ice sheet to be fully unconfined?

22. Do you mean fully confined? As we have limited our analysis to this. As stated in 21. and 12. we are not aiming to fully represent ice sheet deformations but provide a systematic look at deformations away from pure and simple shear.

Please cite Gusmeroli et al. 2012 for sonic measurements of fabrics.

23. We will add this. **Now added in Section 2.1.2, paragraph 4**

- Part 2.2.2: How do you extrapolate surface velocity measurements to get access to in-depth flow history? What are the limitations? Where can it be used, and where it can't, and why?

24. Ice shelves form near-plug flows in which the surface velocity represents the velocity throughout the depth of the ice flow. Ice streams also form near-plug flows and will likely be two-dimensional in their flow properties except for flow close to the base, where basal conditions may generate localised three-dimensional flow. We have done some quick calculations with the shallow ice approximation with no-slip at the base, for ice flowing down an incline and n=3. These suggest the ice will remain at greater than 90% of the surface velocity to a depth of around 56% into the ice sheet. In all situations, the surface velocity provides the leading-order *direction* through the depth of the flow in accordance with all standard thin-layer regimes of ice flow (shallow ice, stream, or shelf), but can be subject to vertical shear. Therefore, while our motivation (which we will better clarify in revision) is nonetheless primarily to indicate deviation from pure and simple shear *per se*, it is in fact reasonable to expect surface velocity to imprint on the depth of an ice sheet flow to a certain depth (in many cases, such as ice shelves and ice streams, almost the full depth); this point was left largely implicit and can be detailed more explicitly in revision. Exploring the effects of vertical shear on the fabrics we have determined in this paper would make for an interesting extension that would encompass a broader range of ice-sheet flow deformations, with the analysis here a necessary first step.

**We have updated the figure showing the vorticity number of Antarctica to include estimates for the vertical derivatives. This is done by assuming the vertical profile is following the shallow-ice approximation. We then estimate the derivative down to a depth of 25% into the ice-sheet. We also calculate dw/dz by continuity. Dw/dx, dw/dy can are small by order of magnitude analysis so this figure now represents and estimate for the 3D vorticity number up to 25% into the ice-sheet. Compared to the previous figure, including the vertical derviatives makes vorticity numbers closer to 1 and reduces the calculated error. In regions where there is basal slip, the shown vorticity number will be valid deeper into the ice sheet.**

- Part 3: See my comment above, please provide here the main assumptions that are made in this model, from a mechanical point of view (how are the mechanical strain and stress field distributed in the microstructure, what is the flow law considered, how are the interactions taken into account, what are the boundary conditions, etc...), and from a physical point of view (what are the assumptions made to formulate the recrystallisation mechanisms, and why).

25. Please see above for a discussion of the model assumptions. It is worth noting again that the idea of strain/stress in a microstructure, and explicitly describing flow laws and grain-grain interactions do not apply. We are happy to provide more discussion of the assumptions in the model.

**In relation to this comment, section 2.3 clarifies the position of SpecCAF in the hierarchy of modelling approaches, and we clarify that the model is a fabric evolution model only.**

Some assumptions made, like the parametrisation with the deformability for instance, or the one for the temperature effect, are very strong and very likely control the results. It would be clearer to emphasise them and test their relative impact.

26. As stated in our response no. 15 above we will test the variability of the temperature parameterisation in a supplement.

**See the supplement where we test the results against variations in the parameters as functions of temperature. We also hope we have but more emphasis on the assumptions implicit in the model in sections 2.1.1 and 3.1.**

As it is presented, it appears to me as if the model was a parametrisation of the rotation of crystals, under homogeneous imposed strain, and not a mechanical modeling (such as Elmer-Ice or VPSC) able to provide interactions between the stress and strain field and the fabric evolution (see Martin et al. 2009 for instance).

27. Please see our response no. 3 & 4 noting the statement 'a parametrisation of the rotation of crystals, under homogeneous imposed strain' does not apply. The referee is correct in stating that the model presented here does not involve coupling between fabric evolution and the flow field, rather the fabric is solved from an imposed velocity gradient and temperature. A full coupling is not the intention of the present paper and is not required to address questions of what kinds of fabrics arise from given deformations. It should be noted in comparison to Martin et al. (2009), where second-order orientation tensor representations of the fabric are used, the model presented here is able to model much more detailed features in the distribution function than is possible than in this previous approach.

**We have clarified the model does not represent 'homogenous imposed strain' in our discussion of the Taylor type approximation (section 3.1).**

- Part 4: the limitations associated with the 2D formulation are not mentioned. Can it be applied in every stress and strain configurations considered? See my comment above.

28. See our response no. 12. We agree more explanation would be useful and we are happy to provide this.

**In line 28 we mention that the 2D formulation is a first step away from pure and simple shear. At the start of the discussion we have added a paragraph discussing the limitations**

- Part 4.3 and discussion: to my point of view, in order to test the robustness of the results presented, the authors should provide results within which the parametrisation is modified, and the effect of the assumptions made tested. In particular, the steady-state obtained is highly dependent on the way the recrystallisation is modeled, on the parameters that control the effect of temperature. By changing them slightly, are the steady-state still reached in the same conditions?

29. See our response no. 15

**We have added a parameter sensitivity study in the supplement. We have revised the steady state figure (and consequently the discussion) to show the halfway time to steady-state, rather than the steady state time which the reviewer rightly predicted would be sensitive to parameter variations. As**

**can be seen in the supplement the halfway time is not sensitive to changes in the parameters. Because of this we have changed the discussion in section 5.2**

- Part 5.2: I don't think that the model can be, as it is, predictive in terms of relation between finite strains and steady-state fabric owing to the fact that it neglects the complexity of the deformation history along flow lines, that it considers a homogeneous state of strain. Taylor-type of approximation, by neglecting the strong anisotropy of ice, very likely underestimate the fabric development rate (see Castelnau et al. 1996). It therefore seems to me hardly transferable to ice core interpretation.

30. We again highlight the fact that the model has been shown to successfully predict the fabric development of experimentally produced fabrics in both endmembers of pure and simple shear. However, we agree that assuming a constant deformation history restricts direct application of the simulations we show in this paper to ice core interpretation. Within the scope of the present paper, we do not wish to model a real ice flow history, just to show the evolution from an isotropic fabric towards final steady states, and to document the properties of the final steady states, as a demonstration and first order analysis.

**We have added a discussion of the Taylor-assumption and clarified the model does not require this assumption in section 3.1. We have also added an experimental pole figure to Fig 7. (from Qi et al. 2019), to highlight the good agreement between the model and experimental fabrics.**

Instead of citing Faria et al. 2014, please refer to some of the original work that deserve the credit, since Faria et al. 2014 is a review.

31. Thank you, we will be sure to change to the earlier papers. **We have replaced this with a reference to Schytt, 1958 on line 437**

- Part 5.4:

Before Minchew et al 2018, you could refer to Russell-Head and Budd, 1979, Alley 1988, Van der Veen and Whillans 1990, etc...

32. Thank you for highlighting these highly relevant additional references (to be incorporated in a revised version of the manuscript). **This has been replaced by a reference to Alley 1988 (line 542)**

By the way, the work of Minchew et al. 2018 seems to contradict the hypothesis of an evolutive effect of migration recrystallization, and go in favor of the fact that the fabric, in conditions where this recrystallization regime is dominant, is dominated by the state of stress (also mentioned by Alley 1992). Indeed, it shows that in shear zones, the fabric is very rapidly steady.

33. It should be noted that Minchew et al. 2018 does not model the fabric, but rather infers the enhancement factor by making a series of assumptions. They find values of between 6 and 10 for the fabric enhancement fit their model. This variation of 40% of the enhancement factor leaves plenty of room for the development of the fabric to continue to take place. We also note simple shear experiments at high temperature (Qi et al. 2019, Journaux et al. 2019) show fabric development with strain at such high temperatures. These laboratory results require far fewer assumptions to reach this conclusion.

**2nd review**

We again thank the reviewer for their helpful and positive review. Based on the discussion below, we have added much more explanation of the model and discussion of the assumptions required for it. This is in section 2.1.1, 2.3, and 3.1. As mentioned below, we have emphasised the fact the model agrees well with laboratory experiments by including an experimental pole figure into Fig 7 (previously Fig 5). We have added below in purple comments to our original "reply to reviewers comments" to highlight where and what changes have been made in the revised manuscript.

**General comments**

The authors have used a numerical model to explore changing CPO orientation and strength in ice deforming in flow regimes intermediate between pure and simple shear and with varying degrees of vorticity. They are able to present detailed data representing ice fabrics at much higher strains and under a much wider range of conditions than it is possible to achieve with laboratory experiments, which is an important contribution to the field. Their examination of steady-state fabrics at high strains is particularly valuable.

The manuscript could be made more impactful by adding more thorough comparisons between model results and experimental results from the cited literature. Some more clarity on the model setup would also be useful.

We thank the reviewer for their helpful and positive review of our submission. We agree with the changes suggested here and below. In particular, we agree that more explanation of the model would be helpful, and this is discussed also in our response to referee 1, which we summarize here. The proposed additions include clarifying the assumptions in the model, in particular distinguishing that we model the evolution of the distribution function of c-axes as a continuum representation of microscale processes, as opposed to explicitly accounting for individual grains.

As noted by the referee below, reviewing the model assumptions will cover some similar ground to the validation/calibration which was the focus of Richards et al. 2021, but we now understand that a more detailed review of the model, the assumptions taken, and its validation is warranted and important to reinforce here also, and this can be straightforwardly implemented in revision.

We have added more comparisons to experiments in Fig 7 (previously Fig 5), by adding an experimental pole figure. We have also compared the fabric patterns of Budd et al. 2013 to those predicted by Fig 11 in section 5.1 lines 406-411, showing general agreement.

**We have also clarified the model setup by clarifying the assumptions in section 2.1.1, clarifying the models place in the hierarchy of ice models in 2.3, and adding more discussing of the model theory in 3.1.**

**We have also included a parameter sensitivity supplement, exploring the sensitivity of the results to changes in the parameters based on 80% confidence intervals (Fig 6.).**

**Specific comments**

l.30: This statement (that looking at the c-axis alone is sufficient) needs some justification. Chauve et al. [2017] have found that non-basal slip systems are very significant at high homologous temperatures. Are you assuming that at lower temperatures they are no longer significant? Please clarify.

Thank you for highlighting this. Indeed, we do not wish to imply that non-basal slip systems are unimportant, but that, for the aim of modelling the fabric, the c-axis provides a sufficient leading order approach to capture experimentally produced fabrics, as confirmed in Richards et al. 2021, for example. In other words, while a description of the model is reduced to tracking of the c-axis distribution, some amount of the effects of non-basal slip sliding for higher temperatures is still likely captured by the empirical calibration (Richards et al. 2021). We will modify the referenced statement to make this clear.

**Reworded to: Basal-slip is generally dominant in ice (Duval et al., 1983), although non-basal slip systems are active, especially at high temperatures (Chauve et al., 2017).**

l.50: Jun et al. [1996], and Budd et al. [2013] also performed experiments with a combination of shear and compression.

Thank you, we will include these references also.

**We have added reference to Budd and Jun in section 2.1.2, including reviewing their results (lines 71-80). We have also compared the fabric patterns of Budd et al. 2013 to those predicted by Fig 11 in section 5.1 lines 406-411, with general agreement.**

l.76, and Fig.1 caption: There are referenced experimental examples for all of these fabrics aside from pure shear. I suggest citing Kamb [1972], or a more modern reference if one exists (I'm not aware of any).

Thank you again, we agree and will add these references.

**We have referenced Kamb 1972 (line 99 and Fig 1. caption). We have also referenced Budd et al. 2013 as it contains an example of pure shear.**

l.113: I'm unsure what W >∞ means. What is more rotational than pure rotation? Is there a section of the Ross Ice Shelf that is continually spinning in circles? Please make this clearer for easily confused readers like me.

We apologise, this was a typo. It should have said W>1 rather than W> ∞, which we will be sure to correct.

**This part has been removed.**

Section 2.4: This is almost a rewording of your stated research questions in lines 18-25, but not in an obvious way. It would be clearer to refer more explicitly to which question you intend to answer in

which section. As I've interpreted it, you've addressed your first question (which deformations are present in the natural world) in section 2.2.2, and the following two (how fabrics evolve, and how steady-state depends on temperature and deformation) will be addressed in the following sections. In that case, it would be clearer for the reader if you reiterate the second two questions here with similar wording as in the introduction, and do the same with the first question where you address it above.

We agree with the comments here and will make these changes.

**We have changed this section to reference the sections in which we answer the open questions, as suggested above**

Section 3: I am not entirely clear on what is being physically represented in the model. E.g., are non-basal slip systems being incorporated, even just by an empirical parameter? How are grain boundary interactions being represented, if at all? To find the answers, the reader must decipher a page of equations. I'm aware this has been covered in more detail in Richards et al. [2021], but a brief explanation at the beginning of your methods section referring back to section 2.1.1 and stating how the specific mechanisms you have described are represented in your model would be very useful.

We agree and plan to add more description of the model to cover this. Our initial intention was to defer to the validation in Richards et al. 2021, but we now appreciate that a thorough review of the model and a more detailed account of its physical representations is worthwhile to include here also,

As remarked above, the effect of non-basal slip systems on the c-axis distribution is captured empirically within the model by the experimentally calibrated temperature-dependent parameters iota(T), beta(T) and gamma(T). It was not our intention to imply that these slip mechanisms are not important at high temperatures, and we will be sure to clarify this in revision.

Regarding the representation of grain boundary interactions, we remark that the model is a continuum model of the evolution of the distribution function of c-axes, so only the bulk effect of grain boundary interactions on the distribution function is modelled. The fact that, once calibrated, we can reproduce experimental results – without additional fitting parameters - demonstrates that the approach taken is sufficient to model this distribution function.

**In section 3.1, before the equations we have added a discussion of how the processes in section 2.1.1 are represented in the model, along with other discussion on what assumptions are implicit in the model**

Figure 5: It would be comforting for an experimentalist reading this paper to see some real experimental results alongside the model results, to show the agreement between the two on the presence of double clusters, and their strength. Fig. 5 shows your results from a simulation of simple shear at −5◦ C, and this is a scenario which has been tested experimentally by Qi et al. [2019]. Perhaps you could take some results from Qi et al, where they have provided detailed c-axis plots and J-indices, and present them alongside your model data plotted in a similar fashion. Again, I know this comparison appears in some capacity in Richards et al. [2021], however it would make this current paper much more convincing to see specifically the agreement on double clusters.

Thank you for this suggestion. We are happy to provide this by adding a pole figure from Qi et al. 2019 into this figure and adjusting the strain so it matches.

**We have added an experimental pole figure from Qi et al. 2019 to this figure (now fig. 7)**

Fig. 8 and Fig. 9: What is the resolution of the data used to make these contour plots (i.e., how many W and T values were tested)? This information would be useful for interpreting parts of the figure. E.g., in Fig. 9c, it would be good to know how many data points make up the "wiggles" between the single cluster and secondary cluster zones at high W.

The resolution of these diagrams is 100x100. The boundary between regimes exhibiting a single-maxima versus a secondary cluster is not exactly sharp and we plan to update this figure to broaden the current sharp boundaries to highlight the more gradual transitions.

**It should be noted that the resolution in the updated figures is 50x50, we have decided to keep the boundaries as they are. We have added this information to the figure captions.**

Section 5.3: The observation made earlier in the text, that highly rotational fabrics can produce a CPO which would ap- pear isotropic when sampled at the resolution of most commonly used techniques, is important for ice core interpretation. It should be mentioned in this section.

Thank you for highlighting this, we will certainly take your advice and highlight this more explicitly.

**We have added this sentence to section 5.3: Furthermore, we have shown that the presence of a fabric which appears to be isotropic could be indicative not only of no deformation, but also of highly rotational deformation regimes.**

l. 321 It is unclear what is meant by "other constraints". Other data on strain history and temperature in the area?

Other constraints could include, for example, knowledge that the deformation history was constant to good approximation over a certain distance, in which case the vorticity number and temperature could be estimated more precisely. We will be sure to list potential additional constraints in more detail.

**We have added an extra clarification in the text (section 5.3) : "other constraints such as knowledge the deformation regime history or temperature has been constant to good approximation"**

l. 325: "We also assume an initial random orientation for the fabric". Although I assumed that was the case, this information would have been useful to know much earlier. If it was explicitly stated above, I missed it!

Thank you for highlighting this, we will update the manuscript to clarify this earlier. **Added to the start of the results, section 4.1**

l. 326: It is true that ice formed from accumulation will have an initially random fabric, but as you say the fabric can adjust to the stress conditions very quickly (within strain of 0.2). It would be interesting to know if a pre-existing fabric affects the evolution of a CPO once the stress conditions change. That's a completely separate paper of course, but worth pointing out as an area of future research...

Indeed, this is an interesting question, we agree with you that this would be a separate paper – in fact, this is one avenue we are planning to explore.

Section 6: I think there are two important findings mentioned in the text which have not found their way to the conclusions: firstly, that the W =0 case which is very commonly found in experiments results in a

CPO which is different to that found if there is even a small amount of rotation, implying that the most common experimental scenario is not representative of most real scenarios (line 312). Secondly, that the double-cluster/cone CPO which we see so often in experiments is not present in your results in steady-state, and appears not to persist beyond the highest strains which we can reach with experiments (line 234). These deserve a brief mention here.

We thank the reviewer for highlighting the importance of these findings and will be sure to emphasise this point. We agree that our finding that slight rotation makes the double maxima fabric unstable at high strains, is interesting. . We remark also that although the steady-state fabrics produced for W=0.1 and W=0 are very different after sufficient strain and in steady state, the initial fabrics at low strains are similar in accordance with experimental observations.

**We have this into the conclusions highlighting that double-maxima only persist for W=0: "We show that for two-dimensional deformations, the double-maxima fabric is not present at high strains when only a small amount of vorticity is present in the deformation regime W>0.1. This is important as many laboratory experiments are performed for W=0. Future work could investigate whether this conclusion extends to three-dimensional cone-shape fabrics."**

**Based on the comments by the first reviewer, we introduced a parameter sensitivity study (figure 6). The time/strain to full steady-state was found to be sensitive to variations in the parameters so we have instead plotted the halfway time to steady-state (a kind of fabric half-life). Because of this, we have reduced our discussion of steady-state fabric strains.**

**Technical corrections**

Fig. 5 caption: "which", not "witch"

l.274 should read "fabric patterns"

l.50 "strain" rate response

Apologies, thank you for highlighting these typos, we will update the manuscript to correct them. **These have all been corrected**

**References**

W. F. Budd, R. C. Warner, T. H. Jacka, J. Li, and A. Treverrow. Ice flow relations for stress and strain-rate components from combined shear and compression laboratory experiments. Journal of Glaciology, 59(214):374–392, 2013. ISSN00221430. doi: 10.3189/2013JoG12J106.

T. Chauve, M. Montagnat, S. Piazolo, B. Journaux, J. Wheeler, F. Barou, D. Mainprice, and A. Tommasi. Non-basal dislocations should be accounted for in simulating ice mass flow. Earth and Planetary Science Letters, 473:247–255,2017. ISSN 0012-821X. doi: 10.1016/j.epsl.2017.06.020. URL http://dx.doi.org/10.1016/j.epsl.2017.06.020.

L. Jun, T. H. Jacka, and W. F. Budd. Deformation rates in combined compression and shear for ice which is initially isotropic and after the development of strong anisotropy. Annals of Glaciology, 23(May 2021), 1996. ISSN 02603055. doi: 10.1017/s0260305500013501.

B. Kamb. Experimental Recrystallization of Ice Under Stress. Flow and Fracture of Rocks, American Geophysical Union Geophysical Monograph, 16:211–241, 1972. doi: 10.1029/gm016p0211.

C. Qi, D. J. Prior, L. Craw, S. Fan, M. G. Llorens, A. Griera, M. Negrini, P. D. Bons, and D. L. Goldsby. Crystallographic preferred orientations of ice deformed in direct-shear experiments at low temperatures. Cryosphere, 13(1):351–371, 2019. ISSN 19940424. doi: 10.5194/tc-13-351-2019.

D. H. Richards, S. S. Pegler, S. Piazolo, and O. G. Harlen. The evolution of ice fabrics: A continuum modelling approach validated against laboratory experiments. Earth and Planetary Science Letters, 556:116718, 2021. ISSN 0012821X. doi: 10.1016/j.epsl.2020.116718. URL https://doi.org/10.1016/j.epsl.2020.116718.

**3rd review**

We again thank the reviewer for the helpful and thorough review of the manuscript, especially the suggestion of a parameter sensitivity investigation. We have implemented all suggested changes, detailed below in purple, and believe the manuscript is much better now thanks to these changes. We have added in purple comments to our original "reply to reviewers comments" to show where we have made changes in the manuscript.

**Responses to Reviewer Fabien Gillet-Chaulet**

We thank the reviewer for their constructive review. We agree with the majority of the suggestions and appreciate all the comments raised. We believe the comments can be addressed in revision by including:

a.  A complete clarification and review of the numerical model used and its underlying assumptions, including its place in the hierarchy of complexity for fabric evolution models. We agree with the reviewer and comments by reviewer Montagnat that the discussion between Faria (2008) and Gagliardini (2008) can be clarified here. Finally, we will be more explicit in the text to clarify that the model presented here is a fabric evolution model only; consequently, it is not set-up, nor are we seeking within the scope of the present work, to solve a coupled full-Stokes system. We do agree that such a model including a coupled Stokes system would be a good next step, hence we are happy to add in the discussion a future perspective of how this can be done. **To address this, we have included a new section 2.3 that provides discussion of the model's place in the hierarchy of ice modelling approaches. Clarification of the discussion between Faria (2008) and Gagliardini (2008) has been included in Section 3.1. We have also clarified that the model is for fabric evolution only in 2.3 (last paragraph).**

b. Clarification that the discussion of vorticity numbers derived from two-dimensional strain rates in Fig. 1 was primarily intended to motivate the basic need to look beyond the limiting endmembers of simple shear and compression. We are not suggesting there is a direct link between the predicted 2D fabrics and these regions, and appreciate that the motivation to show Fig. 1 was not made sufficiently clear in the original submission. As correctly noted by the referee, there is the potential for significant three-dimensional deformations. We admit that the previous version did not clarify this sufficiently. As stated in our reply to reviewer Montagnat, we aim to use this figure primarily as a motivation for exploring deformations away from pure and simple shear, and 2D deformations are a logical first step away from this. Furthermore, following a comment here we realise that a clearer motivation for our 2D analysis arises from considering a vertical cross-section (in the (x,z) plane) of an ice sheet, for which simulations are often performed. This would encompass regions involving pure shear and simple shear and all intermediate cases between these endmembers (as noted by the reviewer), which will arise through the depth of the ice sheet. Again, not all regions of the ice sheet will conform to this regime precisely due to the presence of horizontal deformations, but the case of 2D deformations we consider provides a necessary first step for the systematic documentation of fabrics, which we would like to emphasise in revision. **We have now changed the title to "Ice fabrics in two-dimensional deformations: beyond pure and simple shear" in order to clarify the focus. We have also added 2 new figures (figures 3 and 4), showing the vorticity number from a 2D vertical cross section of ice at a divide and over a bump respectively. Based on these new figures, we have included a new discussion in section 2.2.2 to highlight that the results can be motivated simply by the diversity of deformation regimes occurring through the depth of a 2D cross-section of ice alone, and that we are limiting our analysis to 2D deformations.**

This paper present an application of specCAF, a numerical model of fabric development based on a continuum theory by Faria and Placidi, and described in Richards et al. (2021). Compared to previous works on ice fabric evolution, this paper discuss the fabric patterns obtained for a wide range of vorticity numbers, including highly rotational flows, using synthetical 2D experiments. To justify this approach, the authors have computed the vorticity number from observed horizontal surface velocities in Antarctica.

**We have changed our background motivation, and introduced new figures 3 and 4, showing the vorticity number through a 2D vertical cross sections from an ice divide and flow over a bump, to highlight that we are motivating our analysis of two deformations primarily by the 2D vertical cross section (see summary point b above). We have also updated our analysis of the surface vorticity number to include the vertical derivatives (discussed in more detail below) so now it can be said to represent the 3D vorticity number (rather than just the surface) down to a depth of 25% into the ice-sheet.**

They obtain big (>1) vorticity numbers in large portions of the ice -shelves with curved stream lines, and a conclusion of the paper is that in such regime the fabric should remain nearly isotropic.

We remark that the surface vorticity number from Antarctica is merely intended as an illustrative example. For any complex flow field ice will experience deformations away from pure shear and simple shear. The presented work acts as a first and currently unexplored step towards deformations away from these endmember flow regimes, and as a first step we limit the analysis to 2D. In revisions, the later will be made explicit.

As mentioned in our reply to Reviewer Montagnat, we seek to correct the statement that curved streamlines necessarily lead to vorticity numbers >1. However, according to our analysis vorticity numbers >1 should lead to a weak fabric.

**We have removed the statement about curved streamlines.**

My main comment, is that I remain very sceptical about this conclusion and the interpretations of the results for fabrics in natural flow. The authors claim that most previous studies have focused on pure and simple shear, this is true, but they forgot to mention that the justification is that something between pure shear and uni-axial compression in the **« vertical «** direction is supposed to dominate in the upper ice layers while simple shear (**parallel to the bed**) is supposed to dominate in the lowest layers, at least in the central parts of the ice sheets where ice cores have been drilled and direct fabric observations are available.

We agree and, on reading this (and a similar comment received from reviewer Montagnat), appreciate that we – in the initial submission – failed to be sufficiently clear about the purpose of Fig. 1. It is used for motivating the analysis from first order observations. Indeed, vertical strain likely applies widely due to thickness variation of the ice sheet, and shear will indeed apply (particularly the lower 50%) of central parts of the ice sheet (with no slip at the base), and hence the direct application of fabric predictions in 2D cannot be attributed directly to these regions, at least without further quantification of the role of three-dimensional deformations.

Our intention with Fig. 1 was only to provide a basic quantitative indication of the diversity of deformation styles in natural ice flows beyond the idealised situations of simple shear and compression (whether two-dimensional or three-dimensional) on which experimental analysis has focused to date. It was not our primary intention to attribute the fabrics arising from two-dimensional deformations directly to these regions. The essential indication of the diversity of deformation styles is nonetheless helpful to motivate the study, particularly, we believe, for the benefit of highlighting the limitations of current experiments. We will address this in revision. As remarked above, our results here provide a first step towards documenting the full range of fabrics that can apply, since (for example) pure shear in the vertical can be included with just one additional parameter alongside the horizontal vorticity number W. Given that the exploration of 2D fabrics is already highly rich, it is sensible to retain the scope of 2D deformation alone for one paper before this additional complication is added.

That said, it is still interesting to discuss where two-dimensional deformations may apply to good approximation in the case of natural ice flows. As highlighted by the referee, we would expect, for example, that in approximately horizontally one-dimensional ice sheet flow, the *vertical* cross-section of the flow will experience a spectrum bridging simple shear at the base and pure shear near the surface. Indeed, this spectrum corresponds directly to the range of deformations we explore in this paper. In this case, horizontal deformations would affect this profile, and more work would be needed to address these more complex situations. Nonetheless, the motivation based on deformations experienced in the vertical plane is straightforward, and we would like to include it in revision with appropriate explanation of caveats; we are grateful to the reviewer for highlighting it here.

As an incidental point, we also remark that in a vertical cross-section of a horizontally one-dimensional flow, the relevant endmember for pure shear is the two-dimensional (confined) version that we report here, not uniaxial (radially symmetric) compression, the latter being the focus of experiments of compressed or extended cylindrical samples of ice. In fact, the fabrics produced in confined (two-dimensional) compression differ significantly from those in uniaxial compression, and it is the two-dimensional form included in our analysis here that is the one which is the most relevant endmember to

discuss in the context above. Uniaxial compression by contrast requires radial spreading of a compressed cylinder of ice. This situation does not readily correspond to anything in natural ice flows (perhaps flow at the centre of an ice dome would be one, very rare, instance of this). We will clarify this important point in a revised manuscript, giving yet further motivation for our work.

**We again thank the reviewer for pointing this out. Based on this comment we have adjusted our motivation to primarily focus on the 2d vertical cross section (summary point b). We present fig 3 showing the range of vorticity numbers at an ice divide, going between 0 and 1. We also show ice flowing over a bump (a modification of Exp F from Pattyn et al. 2008) to highlight vorticity numbers >1 (figure 4).**

It's not clear from section 2.2 how the spin and strain – rate tensors are computed for the observed Antarctic horizontal surface velocities? It is assuming plane strain in the horizontal plane? I don't think that an horizontal 2D plane strain would be a good approximation of the natural conditions in ice shelves. I still would expect to have a compression component in the vertical direction, so the interpretation of the results presented here in term of fabrics in natural conditions need better justifications.

**Considering this helpful comment, we have updated the plot of vorticity numbers in Antarctica to include dw/dz (calculated using div.u =0) and the du/dz and dv/dz (calculated by assuming a shallow-ice approximation velocity profile with n=3 and calculating the derivative in the top 25%). Accordingly, the vorticity number shown in this plot is three-dimensional and is valid for the top 25% of the ice-sheet. This is now explicitly stated/clarified in the figure caption and text.**

We certainly agree with the referee. In this section it was not our intention to assert that the surface velocities represent the full deformation field, but merely to illustrate that two-dimensional vorticity numbers away from 0 and 1 (including >1) derived from horizontal velocity fields alone motivates analysis of fabrics beyond those that have been analysed from existing laboratory configurations. As noted above, a one-dimensionally flowing ice shelf would indeed involve a pure shear flow in the vertical along-flow (flow line) cross-section (x,z). In such a case the *vertical* compression is equal in magnitude to the horizontal extension (by incompressibility). Although we had not mentioned it previously, this range of two-dimensional deformations arising in the vertical cross-section of a horizontally one-dimensional ice flow (in both central and floating regions of the ice sheet) will generally involve pure shear in the vertical cross-section, simple shear near the base, and a mixture of pure shear and simple shear elsewhere; these are precisely the regimes and spectrum of deformations we have studied. As noted above, this is a further and potentially clearer and more straightforward motivation for our analysis of two-dimensional fabrics than the illustration of horizontal surface deformations alone. Hence, we aim to include the aforementioned arguments/reasoning in the revisions.

In the late 1990 and early 2000 it was recognized in the geological community that flow in rocks cannot be approximated by endmember plane strain flow models alone. There is now an extensive literature within structural geology which developed conceptual models and analytical techniques to predict and recognize natural flow with vorticities between 0 and 1. In contrast, in the ice flow community, such analysis is not yet common place – here pure shear and simple shear has dominated discussions for both flow and fabric development models/interpretations. This may be mainly due to the fact that such endmember scenarios are a) experimentally straightforward to achieve and are the only experiments in the literature so far and, b) the two endmembers can – as a first approximation - be associated with different "ice flow scenarios". In the revisions, we suggest to include a short review of the geological vorticity literature.

**(see description of revision above)**

I read the comments from the other reviewers and the author responses. The debate between Gagliardini and Faria has not really been clarified and I think that this papers could be a good opportunity to clarify the assumptions behind the continuum approach and how it compares with homogenisation models. Two points seems to require clarifications.

We agree with the reviewer that our paper provides a nice opportunity to clarify the assumptions behind continuum modelling of fabrics, particularly the relationship between the model and those for single crystals, and its relationship to the process of homogenisation (we elaborate below).

**We thank the reviewer for this suggestion, and have clarified this extensively in section 3.1, 3rd paragraph**

First, the classical approach in ice flows model is to solve the Stokes equations (or some shallow approximations) for a given flow law, i.e. a relation between the macroscopic strain-rates and stresses, that are then solution of the problem. It is not clear here how such a relation could be obtained from specCAF. Faria (2006a,b) gives some homogenisation rules to compute the macroscopic stresses, but it seems that is has never really been used. Instead Seddik and others (2008, 2011), using the CAFFE model, parameterized an « enhencement » factor as a function of the polycrystal deformability that depends on the fabric. Using the same argument as for the strain rates, i.e. the volume contains an infinitely large number of grains, Seddik and others (2008) claim that the stress tensor do not depend on the orientation. So it is not clear, (i) how both the stresses and strain -rates at the level of the species (i.e. using Faria's terminology in is reply to Gagliardini) can be equal to the macroscopic equivalent, but still with a viscosity tensor that would depend on the orientation, and (ii) if the macroscopic stresses computed this way would be solution of the continuum model, i.e. the balance equations that are derived in Faria's papers?

The model considered here is for fabric evolution only for given deformations, which (for this purpose) does not require coupling to a flow model. While not the focus of the present paper, we nonetheless agree with the reviewer that methods for coupling SpecCAF with an anisotropic viscosity, to simulate the coupled fabric/full Stokes flow, are worth discussing, and we would like to do this in revision. We nonetheless emphasise that we are concerned here only with fabric evolution, and the details of this discussion, while worth discussing, do not concern the results of the present paper where the focus is on predicting fabric evolution under different specified strain fields per se, not its coupling to ice flow.

**We have clarified that SpecCAF is designed to numerically model fabric evolution only in section 2.3, paragraph 3**

Second, an anisotropic model must be able to describe how the fabric evolves. Here, the model includes several processes, including rotation of the ice crystals due to basal-slip deformation. The equation used to take into account this effect (Eq. 5) at the scale of the species in the continuum approach, is based on equations that have been derived for single crystals. According to the description of their model (Richards et al., 2021) : *« If this equation is applied to an individual grain, it describes the c-axis rotation rate (Gödert and Hutter, 1998; Svendsen and Hutter, 1996) under the Taylor hypothesis (neglecting grain-grain interactions). However, since we are using a continuum model that assumes a large number of grains within any solid angle of orientation, any grain-grain interactions are smeared-out (Faria et al., 2008). In this continuum model, we do not therefore require the Taylor hypothesis. »,* From that I understand that the continuum approach would give a fabric evolution similar to an homogenisation model that uses the Taylor hypothesis? So maybe, strictly speaking the continuum model do not use the Taylor hypothesis because it does not have grains, but at the end the equations that are used for the species (i.e. the orientations) come from single crystals models? As the model has been calibrated against experiments,

this could potentially affect the interpretation of the relative contributions of the different recrystallisation mechanisms that are included in the model?

We agree the fabric evolution due to basal-slip deformation derived from the continuum is similar to that produced by the Taylor hypothesis, and this could have some effect on the values of the parameters (ι, β, λ). The equation comes from assuming a linear dependence on D (the strain-rate tensor) as Placidi (2010) does. The term [Dijnj – Djknknj] is then valid for any plastic spin induced by deformation and is not necessarily linked to the Taylor bound but appears in other fields, such as fibres rotating in a flow (Dafalias, 2001).

**We have added the following clarification at the end of section 3.1: "Despite the model not including the Taylor hypothesis, the term for basal-slip deformation in the equation below is similar to that which would be derived from a Taylor homogenisation of ice under a simple basal-slip only model (Gagliardini et al.,2009), with the exception that the rate of viscoplastic deformation can vary relative to rigid-body rotation."**

The continuum framework also allows us to include the effect of migration recrystallization on the fabric. We note that no other fabric evolution model has been able to reproduce the detailed features seen in experiments (which also occur in the natural world), even full-field models which are much more computationally expensive.

We agree that care should be taken on interpretating the contribution of different recrystallization mechanisms on the grain-scale from the model parameters, as the parameters represent the contribution to the change in the distribution function and are do not directly correspond to grain behaviour.

**We have also clarified this in section 3.1: "However, care should be taken in attributing our calibrated parameters as applying specifically to grain-grain interactions rather than the bulk interactions representing their net statistical effects in the model."**

I have few other detailed comments listed below:

- Sec. 2.2 : see my main comment, the procedure to compute the vorticity number needs to be better explained and justified especially if it's only done in 2D. Ice is incompressible, so tr(D) must be zero is          this enforced? Also it's not clear of me on which length scale the velocity gradients are computed, directly using a finite difference from the original grid resolution?

**As mentioned above, we have updated the plot of vorticity numbers in Antarctica to include dw/dz (calculated through div.u =0) and the du/dz and dv/dz (calculated by assuming a shallow-ice approximation velocity profile with n=3 and calculating the derivative in the top 25%). Accordingly, the vorticity number shown in this plot is three-dimensional and is valid for the top 25% of the ice-sheet.**

Thank you for highlighting this. The vorticity number is calculated based on the 2D horizontal velocity gradients derived from the observations of surface velocity, even though dw/dz can also be calculated from the surface velocities due to incompressibility as you say. As other derivatives (du/dz, dv/dz, dw/dx, dw/dy) cannot be estimated but are likely to be non-zero, hence we decided not to include dw/dz as it would underestimate the vorticity number. The derivatives are found using second order accurate central

differences on the original grid resolution and then averaged over a 10x10 block as described in section 2.2.2.

The figure below shows the surface vorticity number including the calculated contribution from dw/dz, which makes very little difference.

[Figure]

- Sec 2.3 : « *At the other end of the scale, models such as presented by Gillet-Chaulet et al. (2005) track the evolution of tensorial descriptions of the fabric, without including migration recrystallization. These cannot accurately reproduce detailed fabric patterns but are computationally cheap enough for integration into large-scale models (Gagliardini et al., 2013).* ». Gillet-Chaulet et al. (2005) only present the flow relation, i.e. the anisotropic tensorial relation between the macroscopic stresses and strain-rates, so there is no fabric evolution at all. The equations for the fabric evolutions are presented in Gillet-Chaulet et al. (2006). The fact that it do not includes migration recrystallisation is not a limitation of the procedure itself. Seddik et al. (2011) also derive an equation for the evolution of the orientation tensors from the CAFFE model ; so in principle migration recrystallisation, as it is represented here, could be included within the same framework.

Thank you, we will correct this reference to 2006. Migration recrystallization as represented here is a 4th order process, so cannot be represented by frameworks solving for the 2nd order orientation

tensor. If an evolution equation for the 2nd order orientation tensor is derived by taking the 2nd moment from the CAFFE fabric evolution equation, the term for migration recrystallization depends on the 6th order orientation tensor. Furthermore, the $2^{nd}$ order orientation tensor does not contain sufficient information to distinguish between ODFs produced by migration recrystallization (such as cone shapes or secondary clusters) and simpler fabrics such as single maxima, due to the limited information.

We note further that migration recrystallisation requires the temperature-dependent pre-factor β to have been defined to be used in simulating fabrics. A new development in SpecCAF (Richards et al. 2021) was to provide this through a regression analysis with laboratory data. This, in addition to the solution providing the full ODF field, allows the important process of migration recrystallisation to be implemented.

**We have corrected this reference, and added the above points in blue into our almost entirely new section 2.3**

> Sec. 3.2 : explain what is *y* here and in Fig. 5 and what are the deformation principal axes with respect to this reference frame for the pole figures.

Thank you for highlighting this, we will clarify the strain γ here. The deformation is the same as defined in eq (10) in section 4.1. The principal axes are orientated at 45 degrees relative to the to the x and z (out of the page) directions of the pole figure. We will define the strain and grad u earlier to avoid confusion.

**We have clarified that the strain we use is the effective strain, and defined the principal axes in the caption.**

- Sec. 3.2 : give the expression for the computation of the strain (\gamma) from the strain-rates.

We define the strain-rate in Section 4.1, as above to avoid confusion we will define it earlier.

**Added as eq. 8**

- Page 9, last line : *« Furthermore the pre-factor »,* I'm not such which pre-factor ?

We mean the factor of sqrt(2)/2, we will clarify this in the text.

**This has been removed as we have redefined the strain-rate earlier, so the pre-factor is not needed**

- Fig. 5 : Maybe the schema for the single maxima is a bit misleading at it  shows a single maxima in the vertical direction, while it is directed at 45 degrees.

We are happy to change this.

**The pole figure single maxima is now aligned at the vertical direction as we have included the experimental pole figure alongside.**

- Line 209 : give the definition of the J-index before using it.

Thank you for highlighting this.

**Done**

- Sec. 5.4 : « *The model SpecCAF used in our paper can be coupled with an anisotropic* viscosity formulation to include directional variation in viscosity ». Provide more details *on the exact procedure, i.e. how the stresses are computed with SpecCAF, and the assumptions that would be required for this step.*

We believe the reviewer has slightly overestimated the scope of SpecCAF. SpecCAF is limited to fabric evolution, and we make no attempt here to compute the stresses (through a viscosity formulation). As it is purely a fabric evolution equation, it can in principle be combined with a variety of viscosity formulations should one wish (see below).

**We have clarified that SpecCAF is for fabric evolution only in section 2.3, paragraph 3**

Sec. 5.4 : « *This has been done with simplified fabric evolution models which do not* include *recrystallization and temperature dependence (Martin et al., 2009).* » This gives *the impression* that Martin et al. use the continuum model while they are using an homogenisation model with the static (uniform stresses) assumption. Also, from the CAFFE model, Seddik et al. (2008,2011) derive an anisotropic flow law where stresses and strain-rates remain colinear. So if the same method is used here (depending oon the previous comment), it is not so clear that this model would also produce the syncline patterns in the isochrones that are mentioned few lines latter.

SpecCAF could be combined with either the Static viscosity formulation or the viscosity formulation from the CAFFE model. When we comment on Martin et al. (2009), we refer only to the fabric evolution part of the model and not the viscosity formulation. We will be sure to clarify this in the text.

We agree it is an interesting open question whether a co-linear (or other alternative viscosity formulations/homogenisations) could produce the syncline patterns seen in Martin et al. (2009).

**We have updated this sentence to hopefully avoid any confusion that Martin et al. Are using a continuum model: "Martín et al. (2009) has coupled a fabric model to an anisotropic viscosity, but the fabric evolution model used did not include recrystallization and temperature dependence.**

References:

Martín, C., Gudmundsson, G.H., Pritchard, H.D., Gagliardini, O., 2009. On the effects of anisotropic rheology on ice flow, internal structure, and the age-depth relationship at ice divides. Journal of Geophysical Research: Earth Surface 114. https://doi.org/10.1029/2008JF001204

Gagliardini, O., 2008. Comment on the papers 'Creep and recrystallization of large polycrystalline masses' by Faria and co-authors. Proceedings of the Royal Society A: Mathematical, Physical and Engineering Sciences 464, 289–291. https://doi.org/10.1098/rspa.2007.0187

Faria, S.H., Kremer, G.M., Hutter, K., 2008. Reply to Gagliardini's comment on 'Creep and recrystallization of large polycrystalline masses' by Faria and co-authors. PROC R SOC A 464. https://doi.org/10.1098/rspa.2008.0181

References:

- Faria, S.H., 2006. Creep and recrystallization of large polycrystalline masses. I. General continuum theory. Proc. R. Soc. A. 462, 1493–1514. https://doi.org/10.1098/rspa.2005.1610
- Faria, S.H., 2006. Creep and recrystallization of large polycrystalline masses. III. Continuum theory of ice sheets. Proc. R. Soc. A. 462, 2797–2816. https://doi.org/10.1098/rspa.2006.1698
- Gillet-Chaulet F., O. Gagliardini, J. Meyssonnier, T. Zwinger, J. Ruokolainen, 2006. *Flow-induced anisotropy in polar ice and related ice-sheet flow modelling*, J. Non-Newtonian Fluid Mech. **134**,
- Seddik H., R. Greve, T. Zwinger and L. Placidi, 2011. *A full-Stokes ice flow model for* the vicinity of Dome Fuji, Antarctica, with induced anisotropy and fabric evolution, The *Cryosphere, 5*, 495-508

---

## Referee Report (RR1)

"Ice fabrics in two-dimensional flows: beyond pure and simple shear"
by Richards et al.

Review comments by M. Montagnat, January 2022.

Please find below the review of the new version of the paper. Since a lot of the comments I gave for the first review were not taken into account, I put at the end of the document the previous review I did, for the editor and the authors to do comparisons. The comments in concerned are underlined in yellow.

One of my concern on this paper is the feeling it gives me of a lack of clarity. Hypotheses are done with the model used here, and this is very fine for me, but many of them are not clearly stated. For instance:
- by assuming the rotation rate of the orientation distribution with equation 5, the authors are doing the assumption of a Taylor-type of mechanical interactions between grains. Indeed, the only terms that act on the rotation rate is the strain-rate (and the vorticity that rotate the full distribution). Another way of doing, that would still be a rather crude parameterisation, would be to follow Gillet-Chaulet et al. 2006 paper (eq 13) and put a stress component into the rotation rate. In this case, the mechanical hypothesis behind the orientation distribution rotation is in between a Taylor and a Sachs hypothesis.
The discussion about that in part 3.1, lines 230-245, is very deceiving... What is said lines 238-239 appears just wrong to me in a mechanical point of view. And the justification lines 240-245 is really astonishing! One can always parameterise any model to provide the result expected, it does not mean that the good "physics" is in the model!!! Please remove this sentence.
And please let's assume the choice of the parameterisation as an "OK" hypothesis in order to simplify, especially since there exist no better way of doing so far.
From this choice depends the value of the parameters that have been tuned in Richard et al. 2021 on compression and simple shear cases, and this is fine! Providing it is clearly stated...
- Considering the migration recrystallization mechanisms (in particular lines 264-265), the authors know that what drives them is more complicated that only the "cumulated shear strain"... What drives nucleation and grain boundary migration occurring during dynamic recrystallisation is related to the STORED strain energy, that is related to geometrically necessary dislocations, the ones that help compensating the strain incompatibilities between grains, and their density is not simply related to the cumulated shear strain (some areas that are deforming "easily" cumulate a lot of shear strain and very few geometrically necessary dislocations, so very little stored energy...).
Once again, it makes sense, to my point of view, to use such a simplification in a model devoted to large scale flow modeling, but please mention it as an hypothesis and not as the truth!

- Considering figure 2, for vorticity → infinity, there should be no fabric formed since the material experiences rigid body rotation only? Where could the girdle come from? What constrains the rotation within this girdle, under rigid-body rotation? What is the relative weight of the two parameterised recrystallisation regimes in this weak girdle? And how is it impacted by slight changes in the parameters? Is it robust?

- Once again, this study lies on parameterisation performed in laboratory conditions, therefore very far from the "real world" it aims at representing. A sensibility study would therefore be necessary, to check, for instance, which of the rotation / migration recrystallization process is dominating and in which situation? Does that make sense with "real world" observations?
What happens when the parameters are shifted away from the linear fit? What is the impact on, for instance, the kinetic to steady-state?

This sensibility study is necessary to check the robustness of the modeling and therefore its ability to be predictive.

- Line 411-415: the 2D assumption is strong, and, as already mentioned in my previous review, was shown to give "correct" results only is some specific parts of the ice cores. This part is not an explanation of the limitations of the 2D assumption and the impact it could have on the results, it is more the expression of the authors' opinion "is a good first step"... It may be, but please, explain us why and under which limitations.

More specific comments:

- Line 21: what does "validated" means? When can we consider a model to be fully validated? In particular, this model has not been validated in other deformation regime than simple shear and compression, while you are going to use it in very different conditions.

- Part 2.2.1: VERY IMPORTANT!!! This paragraph contains explanations that are contrary to what is known for ice and recrystallization mechanisms. It should be re-written and bibliography may be more correctly used. For instance: **Chauve et al. 2017 do not show that non-basal slip is active! They just show that under some specific conditions, geometrically necessary non-basal dislocations can be observed. It has already been mentioned in my previous review, and since I am co-author of Chauve et al. 2017, it is very important for me that the authors correct it!**
The mechanisms described lines 42 and 43 are not stricto-sensus deformation mechanisms. Only crystal plasticity in the list is a deformation mechanism. Recrystallization is a process of accommodation that facilitates the deformation, but does not produce a deformation per-se... For instance, during post-dynamic or static recrystallization there are a lot of microstructure modifications without any deformation produced. Please modify.
Line 42: Migration Recrystallization does not refer to grain boundary migration. Migration Recrystallization refers to a recrystallization associated with nucleation AND grain boundary migration, on a regime where grain boundary migration is fast. But nucleation does take place also. Above all, migration recrystallization refers to a mechanism that is driven by stored strain energy while grain boundary migration can occur driven by the reduction in grain boundary surface energy. It would be very important to clearly make the distinction and not mis-explain the migration recrystallization regime... Please see Humphreys and Haterly 2004 if necessary.
Line 45: the paper by Chauve et al. 2017 does not allow to say that non-basal dislocation activity is restricted to high temperature regime... This is just that the experiments presented in this paper are at high temperature. And once again, it just observes some non basal dislocations and no non-basal slip activity. So this sentence should be removed.
Lines 51-55: I am really puzzled to read this part, especially when dealing with ice! Recent results have shown (and some done by one of the co-authors, S. Piazolo), that strain heterogeneities in ice can not be resumed relatively to the main deviatoric stress (see Grennerat et al. 2012 for instance), and that strain distribution is very heterogeneous, with strain high in area of low stress, and the contrary... (see Piazolo et al. 2015, Montagnat et al. 2015, Chauve et al. 2017 Phil Trans). Similar observations exist also in other materials. So to say that "strain energy stored in a grain is directly related to the imposed strain" is somehow too vague and may be wrong... The reference given here, Gottstein and Shvindlerman 1999 is a book that I can not access to to verify. It could be stated as a working hypothesis, and justified, but not as the truth.

**Previous review, August 2021:**

This paper presents some simulations of ice fabrics in conditions relevant for the Antarctic ice sheet. The simulations are made by means of a numerical model inspired by the work of Placidi et al. (2010), that simulates the rotation of individual ice crystals included in a orientation mixture that is submitted to a given strain field, and includes a parametrisation of the effect of dynamic recrystallization on this rotation. This model has been applied recently in Richards et al. 2020 (EPSL) to reproduce laboratory observations.

This paper suffers from a lack of clear explanation of the strong assumptions that are included in the numerical simulations and the associated parametrisation.
Such assumptions, that I detail below, can have a significant impact on the results, and, since they are not clearly stated, they are not tested either, and this undermines the credibility of the study.

- It would be first necessary to recall the way the strain and stress interactions between grains are dealt with in the model.
 Unlike stated in Richards et al. 2020, the model, that derives from previous works of Faria et al. (2006-I,II,III), assumes an homogeneous strain rate, meaning that each crystal is submitted to the same strain rate. This hypothesis, apparently not clearly stated in any of those works, has been shown by Gagliardini (2008) in its response to Faria et al. (2006) to correspond to a Taylor-type of approximation, meaning uniform strain.
 Such an approximation can be clearly recognised as such, and then it is possible to evaluate its impact on the simulation of the mechanical response of the polycrystal, as done by Castelnau et al. (1996). In particular, Castelnau et al. 1996 showed that this approximation was not satisfactory for a highly anisotropic material such as ice since it requires the activation of non-basal slip systems at a non realistic level. By doing so, it strongly reduces the level of strain heterogeneities between crystals, the latter being the main driving force for dynamic recrystallisation. We can expect this approximation to impact the modelling of this mechanism.
Since Castelnau et al. work, it appeared clear that in situation where the full stress and strain field heterogeneities can not be taken into account, an homogeneous stress approximation is more adapted to simulating the mechanical response of ice (see maybe, for instance, the work of Pettit and co-authors).
In Richards et al. 2020, it is mentioned that the fact that the model considers a large number of grains for each orientation specie, reduces (or annihilates) the dependency on the grain orientation on the mechanical state and response (strain and stress). Gagliardini (2008) showed, based on Lebensohn et al. 2004 work, that this is not true and that only the dependency on the neighbourhood is reduced by considering many grains for each orientation.

- The way the dynamic recrystallization is simulated is also based on important assumptions, not always in agreement with laboratory or field observations. It would be necessary to explicitly mention these approximations, and justify their use.
First, in the main part of ice sheets, where temperature and strain rates are low, the main recrystallization mechanisms is continuous (or rotation) recrystallization, characterized by a low driving force for grain boundary migration (see for instance De la Chapelle et al. 1998). In such a regime, the fabric is supposed to evolve only slightly owing to recrystallization, and to remain mainly dominated by deformation (see also Montagnat et al. 2012, for the Talos Dome core).
It would therefore be important to evaluate, in some appropriate locations, the relative influence of the simulated rotation recrystallization versus migration recrystallization in the obtained fabrics. If migration recrystallization, the way it is simulated here, has too much weight on the resulting fabric in location where rotation recrystallization is expected to dominate, the model can be questioned.
In areas where migration recrystallization dominates (high temperature / high strain rate), the grain boundary migration kinematic dominates the softening process, so that the fabric and microstructure end up resulting from the stress state, and loose track of the deformation history (see what happens at the bottom of the GRIP, NEEM, Dome C ice cores for instance, or also in high shear conditions,

Hudleston 1977 for instance, or even Hudleston 2015, see also Alley 1992). Can we expect, in such conditions, an evolution of fabric with strain?

- Second, concerning the physical mechanisms. Migration recrystallisation is supposed, in the presented model, to be governed by a "deformability" related to the total deformation accumulated in the grain. Dynamic recrystallization mechanisms (nucleation and GBM) are related to the local accumulated dislocations in the form of geometrically necessary dislocations (responsible for local misorientations) and statistically stored dislocations. The densities of SSDs and GNDs are not, by default, correlated with the total amount of strain experienced by the grains. It has been recently shown by Harte et al. 2020 for Ni-based alloy by coupled EBSD observations and Digital Image Correlation strain measurements (stored energy is different from cumulated strain).
In various experiments performed on ice, or full-field modeling, it was shown that there is no relationship between the amount of deformation (measured by Digital Image Correlation for instance) and the Schmid factor of a grain. There is therefore no "hard grains", or "soft grains", since the local behavior is much more controled by the grain interactions and the resulting stress redistribution. The uniform strain assumption neglects this aspect too.

My point of view concerning these approximations made relatively to dynamic recrystallization is that they can be useful and justified in the simplified numerical modeling approach used in this work. Nevertheless, it has to be clearly mentioned that they ARE approximations, and their effects should be tested.

- The way the boundary conditions are selected is very unclear to me. Considering that fabric is being formed during deformation in depth of the ice sheet, how can a surface velocity map be representative of the in-depth flow conditions? Can the authors be clearer about that?
The 2D approximation is also strong. It was shown by the Elmer-Ice community to be OK in the case of specific types of flow, like divides (where there is little divergence or convergence). Can it holds for more complex situations such as fast ice streams? What effect could it produce on the fabric evolution? This should be justified and tested.

- What "highly-rotational" conditions represent "in reality"? Does that correspond to area where a block of ice rotates freely on itself? Can that happen in the depth of ice sheets? If yes, where?

- About the capacity of the model to predict steady-state fabrics. Steady-state fabrics depend strongly on the mechanical state the ice is experiencing, and the flow history. I therefore don't understand how could the model be realistically predictive considering the strong assumptions made (1) on the mechanical state (Taylor-type of approximation) and (2) on the recrystallization mechanisms.
In order to test the predictability of the model, it would be necessary to test how robust it is to variations in the parameters, and to the 2D approximation, and to the use of surface velocity vorticity. Such a robustness test was already missing in Richards et al. 2020.

Specific comments:

- Abstract: "a definitive classification of all fabric patterns". This sentence lacks humility... in particular owing to the lack of clarity of the text regarding the assumptions made (see my comments above), and their effects on the obtained simulation results. On top of that, the 2D simulations highly limits the ability to provide this full classification, and also the fact that strain states were deduced from surface observations, very likely not relevant for flow in the depth of the ice sheet.
"Highly-rotational fabrics... produce a weak fabric". Can we expect a fabric to produce a fabric? Not clear to me.

- Part 2.1: The presentation of the processes made in this part is simplistic regarding the many other observations and analyses that exist in the literature (see my comments above). It is OK if it is clearly presented as assumptions made to simplify the processes and better introduce them into the modeling approach. It is a very classical approach to simplify the physics in order to be able to take it into account in a modeling approach. But it needs therefore to be clearly stated, justified, and tested when the results are presented.
What is the "real situation" responsible for some "rigid-body rotation"?

- Part 2.2: Various studies were done in the past that include torsion and compression, or shear and compression, and therefore that consider a more complex scheme that pure or simple shear. None of them are mentioned in part 2. I can suggest Budd et al. (2013), Duval 1981 for instance, but others are mentioned in Hudleston 2015.
At domes, in fact close to domes since deep ice cores are never exactly at the dome location, if girdle is observed it is that not only compression occurs, but also lateral extension. This can signify that the core was cored slightly on the flank, or that dome has moved with time (see for instance NEEM, Vostok, EDML, NorthGRIP). For nearly every deep ice core drilled close to a dome, a shear component was observed close to the bedrock, that participated to strengthen the single-max fabric (see for instance Talos Dome).
Can we consider ice deep in the ice sheet to be fully unconfined?
Please cite Gusmeroli et al. 2012 for sonic measurements of fabrics.

Part 2.2.2: How do you extrapolate surface velocity measurements to get access to in-depth flow history? What are the limitations? Where can it be used, and where it can't, and why?

Part 3: See my comment above, please provide here the main assumptions that are made in this model, from a mechanical point of view (how are the mechanical strain and stress field distributed in the microstructure, what is the flow law considered, how are the interactions taken into account, what are the boundary conditions, etc...), and from a physical point of view (what are the assumptions made to formulate the recrystallisation mechanisms, and why).
Some assumptions made, like the parametrisation with the deformability for instance, or the one for the temperature effect, are very strong and very likely control the results. It would be clearer to emphasise them and test their relative impact.
As it is presented, it appears to me as if the model was a parametrisation of the rotation of crystals, under homogeneous imposed strain, and not a mechanical modeling (such as Elmer-Ice or VPSC) able to provide interactions between the stress and strain field and the fabric evolution (see Martin et al. 2009 for instance).

Part 4: the limitations associated with the 2D formulation are not mentioned. Can it be applied in every stress and strain configurations considered? See my comment above.

Part 4.3 and discussion: to my point of view, in order to test the robustness of the results presented, the authors should provide results within which the parametrisation is modified, and the effect of the assumptions made tested. In particular, the steady-state obtained is highly dependent on the way the recrystallisation is modeled, on the parameters that control the effect of temperature. By changing them slightly, are the steady-state still reached in the same conditions?

Part 5.2: I don't think that the model can be, as it is, predictive in terms of relation between finite strains and steady-state fabric owing to the fact that it neglects the complexity of the deformation history along flow lines, that it considers a homogeneous state of strain. Taylor-type of approximation, by neglecting the strong anisotropy of ice, very likely underestimate the fabric

development rate (see Castelnau et al. 1996). It therefore seems to me hardly transferable to ice core interpretation.

Instead of citing Faria et al. 2014, please refer to some of the original work that deserve the credit, since Faria et al. 2014 is a review.

- Part 5.4:

Before Minchew et al 2018, you could refer to Russell-Head and Budd, 1979, Alley 1988, Van der Veen and Whillans 1990, etc...

By the way, the work of Minchew et al. 2018 seems to contradict the hypothesis of an evolutive effect of migration recrystallization, and go in favor of the fact that the fabric, in conditions where this recrystallization regime is dominant, is dominated by the state of stress (also mentioned by Alley 1992). Indeed, it shows that in shear zones, the fabric is very rapidly steady.

- O. Castelnau, P. Duval, R. A. Lebensohn, and G. Canova. Viscoplastic modeling of texture development in polycrystalline ice with a self-consistent approach : Comparison with bound estimates. J. Geophys. Res., 101(6):13,851–13,868, 1996.

- P. Duval. Creep and fabrics of polycrystalline ice under shear and compression. 27(95):129–140, 1981.

- S. H. Faria. Creep and recrystallization of large polycrystalline masses. I. General continuum theory. Royal Society of London Proceedings Series A, 462(2069):1493–1514, 2006.

- S. H. Faria, G. M. Kremer, and K. Hutter. Creep and recrystallization of large polycrystalline masses. II. Constitutive theory for crystalline media with transversely isotropic grains. Royal Society of London Proceedings Series A, 462(2070):1699–1720, 2006.

- S. H. Faria. Creep and recrystallization of large polycrystalline masses. III: Continuum theory of ice sheets. Royal Society of London Proceedings Series A, 462:2797–2816, 2006.

- O. Gagliardini. Comment on the papers 'creep and recrystallization of large polycrystalline masses' by faria and co-authors. Proceedings of the Royal Society A: Mathematical, Physical and Engineering Sci- ences, 464(2090):289–291, 2021/05/11 2008.

- A. Gusmeroli, E. C. Pettit, J. H. Kennedy, and C. Ritz. The crystal fabric of ice from full-waveform borehole sonic logging. Journal of Geophysical Research: Earth Surface, 117(F3), 2021/05/12 2012.

- A. Harte, M. Atkinson, M. Preuss, and J. Quinta da Fonseca. A statistical study of the relationship between plastic strain and lattice misorientation on the surface of a deformed Ni-based superalloy. Acta Materialia, 195:555–570, 2020.

- P. J. Hudleston. Progressive development of fabrics across zones of shear in glacial ice. In S. K. Saxena and S. Bhattacharji, editors, Energetics of Geological Processes, pages 121–150. Springer-Verlag, New York, 1977.

- P. J. Hudleston. Structures and fabrics in glacial ice: A review. Journal of Structural Geology, 81:1–27, 12 2015.

- Lebensohn, R., Liu, Y., and Casta˜neda, P. (2004). Macroscopic properties and field fluctuations in model power-law polycrystals: full-field solutions versus self-consistent estimates. Proc. R. Soc. Lond. A, 460:1381–1405.

- C. Martin, G. H. Gudmundsson, H. D. Pritchard, and O. Gagliardini. On the effects of anisotropic rheol- ogy on ice flow, internal structure, and the age-depth relationship at ice divides. Journal of Geophysical Research: Earth Surface, 114(F4), 2020/10/20 2009.

- E. C. Pettit, T. Thorsteinsson, P. Jacobson, and E. D. Waddington. The role of crystal fabric in flow near an ice divide. J. Glaciol., 53(181):277–288, 2007.

---

## Referee Report (RR2)

**The Cryosphere: review report of *"Ice fabrics in two-dimensional flows: beyond pure and simple shear"* by Richards et al. (tc-2021-118)**

Dear Editor and Authors,

The manuscript uses the numerical model SpecCAF to simulate and classify crystallographic preferred orientations (CPOs) generated by a wide range of two-dimensional deformation regimes. It is a follow-up of [22]. The text is well written and self-contained. The work has good scientific quality and presents interesting results. I enjoyed reading it. There are however, several clarity issues that require careful revision. None of these issues affect the main results and conclusions of the work, which I recommend for publication after revision.

**Specific comments:**

**Lines 20–21:** To be fair, the most studied deformation regime to date in relation to ice fabrics has been uniaxial (vertical) compression, probably as much or even more than pure and simple shear.

**Line 24:** It would be nice to explain why "ice flow is commonly modelled in the two-dimensional $x – z$ plane", and which $x – z$ plane is chosen.

**Line 26:** Delete the spurious "below".

**Lines 28–32:** Concerning the four open questions to be answered: The first two have been considered by [15] through the combination of theory with experimental extrapolation. The third question is unclear: which "steady state" do you mean? Strain rate steady state? Stationary CPO? Some other kind of steady state?

**Lines 42–43:** Personally, I find the term "crystal slip" a bit misleading and recommend replacing it with "intracrystalline slip", or even better "dislocation glide" (and climb, if you wish to be general; [10]). If you want to keep term "crystal slip", then please make clear what it means (as it stands, I can only guess). Also "rigid-body rotation" sounds slightly misleading, since the material under consideration is not a rigid body. Better would be simply "rigid rotation".

**Lines 42–43:** Assuming the established definition of recrystallization as "the formation and migration of high-angle grain boundaries driven by the stored strain energy" [2, 7, 9, 13], it follows that migration and rotation recrystallization are not deformation mechanisms, but rather annealing phenomena. Admittedly, recrystallization of any kind is closely related to strain, being driven by the stored strain energy and affecting the mechanical response of the material. Nevertheless, recrystallization is not a deformation mechanism per se, since it cannot produce strain (change in shape) or rigid rotation in a stressed body [10, 18, 19, 24, 26]. Migration recrystallization

describes the motion of grain boundaries *through* the material (i.e., without material movement). Rotation recrystallization describes the formation of a new grain boundary. In this respect, it is worth mentioning that some authors confuse cause and effect by erroneously attributing a material rotation to "rotation recrystallization": Actually, the material rotates by a deformation mechanism like dislocation glide and climb, and the strain energy stored in the material by this rotation triggers rotation recrystallization, which is the formation of a new grain boundary. The fallacy that recrystallization phenomena were deformation mechanisms is an epidemic pseudodoxy perpetrated by unreliable sources.

**Line 43:** Insert "in ice" after "slip".

**Lines 60–63:** The references cited here are not the most suitable. For instance, Piazolo et al. [20] is a very interesting work, but it refers only to transient creep in laboratory and simulations, and it would be reasonable to argue that stress and strain heterogeneities may disappear after the transient phase. As it turns out, that is actually not the case in practice, rather the contrary. Kipfstuhl et al. [16, 17] have observed strong strain heterogeneities in shallow and deep polar ice, while Faria et al. [6] explained those stress/strain heterogeneities through the concept of "a highly strained mantle and a less strained core within a grain." As for the diffusion/dispersion of c-axes by rotation recrystallization, Gödert [12] presents a model that simulates the concepts and observations made by previous researchers, while the original concept can actually be traced back to Poirier [21], which was popularized in ice by Alley [1].

**Lines 66–67:** Radar should be mentioned here as well (it is mentioned only later, on Line 92).

**Line 75:** The correct citation is "Li et al., 1996". The surname is "Li", the given name is "Jun".

**Line 78 and elsewhere:** The plural expression "single maxima" is repeatedly misused in singular contexts in many points of the text. The singular is "single maximum" and its plural is "single maxima". Please do not mix them up.

**Figure 1:** Please indicate the principal directions of compression and simple shear. The pole figure (d) is incorrect. The primary cluster should be closer to the centre and the secondary cluster closer to the border of the diagram, at approx. 70° from the primary cluster [1, 15].

**Lines 104–105:** The 45° is a theoretical estimate, because observed angles are less than 45° due to the continual rotation of c-axes towards the main compression axis.

**Lines 110–114:** The explanation for the imbalance in cluster strengths seems a bit confusing. The main reason for the imbalance is neither the "vorticity" in simple shear

(i.e. gradual rotation of the principal strain axes), nor recrystallization. Rather, the imbalance is mainly derived from the fact that, for simple shear, the secondary cluster is unstable, whereas the primary cluster is stable. In other words, c-axes in the primary cluster stay there, while c-axes in the secondary cluster quickly rotate away from it by usual strain-induced lattice rotation. If migration recrystallization were causing the imbalance, more recrystallization would imply a weaker secondary cluster, which is contrary to observation (the secondary cluster actually gets weaker when there is less recrystallization). The function of migration recrystallization is to make the secondary cluster more defined, by consuming the grains with c-axes that rotate away from it and move towards the principal axis of compression ("hard-glide orientations"). The "vorticity" of simple shear generally plays a very minor role, since it is much slower than the effects of c-axis rotation and recrystallization.

**Line 118:** It could be mentioned here that Kamb [15] related ice fabrics to deformation regimes using a somewhat related measure, which he called the "stress character".

**Figure 2:** Please be consistent and use either "rigid rotation" or "pure rotation", but not both.

**Figure 4:** Why have you amplified that much the Gaussian bump? I am afraid that the high vorticity numbers reported in this figure may be derived from such an extreme amplification of the bump.

**Figure 4:** Why have you chosen $n = 1$ instead of $n = 3$ in this example? Intuitively, one would expect a realistic modelling of ice flow with high vorticity numbers to use the non-Newtonian description with $n = 3$. What would be the effect of $n = 3$ on the vorticity numbers in this simulation?

**Figure 5:** This figure intrigues me. Maybe I misunderstood it? I have doubts about the use of shallow ice approximation at the transition from grounded ice to ice shelf...Besides, we know from detailed modelling and ice-core observations that the dominant deformation regime for ice shelves is non-rotational, asymmetric horizontal extension; not simple shear as indicated in the map.

**Lines 238–239:** That is correct indeed. At this point I have to digress to do something that I very rarely do—because it causes me great displeasure—which is to correct erroneous statements by another reviewer. In this particular case I feel obliged to do so, to rectify harmful and unfair criticism to the work under review. The unfair claims by the Reviewer are:

> *The model, that derives from previous works of Faria et al. (2006-I,II,III), assumes an homogeneous strain rate, meaning that each crystal is submitted to the same strain rate. This hypothesis, apparently not clearly stated in*

*any of those works, has been shown by Gagliardini (2008) in its response to*
*Faria et al. (2006) to correspond to a Taylor-type of approximation, meaning*
*uniform strain.*

There are several errors in that statement. First, the Reviewer cites a comment by Gagliardini [11], but fails to cite the subsequent response [5] that proved the falsity of all Gagliardini's comments.

Second, it is true that the SpecCAF model is ultimately based upon the theory of Continuous Diversity developed by Faria et al. [3, 4, 8], but the Reviewer's claim that the theory of Continuous Diversity assumes a homogeneous strain rate for each grain (so-called "Taylor-type approximation" or "uniform strain") is clearly fallacious: it represents a complete disregard for the fundamental principles of continuum mechanics.

The theory of Continuous Diversity (CD) describes the large-scale ("macroscopic") flow of a glacier or ice sheet. As any other continuum theory, all fields and gradients in the CD theory are spatially defined on that large scale, which is many orders of magnitude larger than the grain scale. Therefore, just as the strain rate in fluid dynamics does not impose any constraint, hypothesis or approximation on the motion of individual molecules, the strain rate in the theory of Continuous Diversity does not impose any constraint, hypothesis or approximation on the deformation of individual grains: every grain is free to deform as inhomogeneously as needed. In plain mathematical terms, if $dx$ defines an infinitesimal distance in the continuum (upon which all spatial gradients, including the strain rate, are defined) and $D$ is the average grain size, then $dx \gg D$.

**Figure 7:** Please explain the grey arrow in the figure caption.

**Figure 7:** In the caption, please replace "principal axes of deformation" with "principal strain axes". The former expression does not make much sense when there is rigid rotation.

**Line 304:** Wrong figure reference. It should be "Fig. 7b", not "Fig. 8b".

**Equation 10:** I am confused here. The non-dimensional velocity gradient defined in (10) does not seem compatible with the non-dimensional velocity gradient derived from the definitions (8) below, for its symmetric and skew-symmetric parts. If they are compatible, please show that. If not, which one are you using in your simulations?

**Line 321:** I guess you mean $-5$ °C, not $-10$ °C, right?

**Lines 325–326:** The positions of the clusters for $\mathcal{W} = 1$ (simple shear) seem way off from the observed positions in the real world... Why? The primary cluster should be close to vertical (centre of the diagram, $\theta = 0°$) and the secondary cluster close to horizontal (at around 70° from the primary cluster, that is, $\theta \approx -70°$). Are you rotating the fabric

backwards to remove the vorticity and transform the simple shear into pure shear? Please clarify.

**Line 327:** The secondary cluster is consumed by "c-axis rotation", not "migration recrystallization".

**Lines 329–330:** This statement may need revision, depending on the reactions to the comments to Figures 4 and 5 mentioned above. In any case, "prevalent" is a too strong word.

**Lines 335–336:** The $J$-index as a stand-alone measure of anisotropy has several problems and is considered unreliable [23, 25]. The former reference proposes the use of an $M$-index based on misorientations. Within the framework of a continuum theory with continuous diversity of the type presented here, the definitions and combinations of various anisotropy indices commonly used in ice-core fabric studies are discussed in [4].

**Figure 8:** I recommend adding contour lines or colour steps, as in Fig. 9 or 12, because the smooth colour gradations vary on screen and particularly on print, making it difficult to see the oscillations in the fabric patterns.

**Figure 10:** Same question as before in Lines 325–326. I see an angle close to 30° for the primary cluster for strain = 1 (c) in simple shear ($\mathcal{W}$ = 1). Why? Are you rotating the fabric backwards to remove the vorticity and transform the simple shear into pure shear? In real observations (experiment or ice cores) this angle is close to zero. Please clarify.

**Figure 11:** Why not plotting $\mathcal{W}$ = 0? Should not double maxima occur close to $\mathcal{W}$ = 0? They already appear at $\mathcal{W}$ = 0.1 in Fig. 8!

**Figure 12:** This figure is very useful and it should come before Fig. 11.

**Lines 392–394:** In my opinion, the halfway strain is not very intuitive as a measure of fabric development, because it is normalized by the fabric intensity at steady state. That is, if the steady state fabric is strong, the halfway strain will be larger, giving the impression that it takes longer for the fabric to develop, which is not true, because it may actually develop fast, but it has a long way to reach the "fabric steady state". Therefore, a much more useful measure of fabric development is in my opinion the strain to reach a definite fabric strength. This will tell us how fast fabric develops, which is the information we really need for interpreting ice cores and simulations.

**Line 431:** I am not sure what you mean by "cone-shaped fabric"... Do you mean a single maximum or a girdle?

**Lines 434–435:** This conclusion has already been presented by Kamb [15].

**Lines 446–447:** That is not stated in the cited work by Jacka and Li [14]. In fact, their results indicate that the mechanical steady state depends on stress and temperature.

**Line 650:** Please correct this reference. The authors' list is wrong and the reference data are incomplete.

I hope the Authors and the Editor find these comments useful.
Best regards,
Sérgio Henrique Faria

[1] R. B. Alley. Flow-law hypothesis for ice-sheet modelling. *J. Glaciol.*, 38:245–256, 1992.

[2] R. D. Doherty, D. A. Hughes, F. Humphreys, J. J. Jonas, D. Juul Jensen, M. E. Kassner, W. E. King, T. R. McNelley, H. J. McQueen, and A. D. Rollet. Current issues in recrystallization: a review. *Mater. Sci. Engineer.*, 238:219–274, 1997.

[3] S. H. Faria. Creep and recrystallization of large polycrystalline masses. Part I: general continuum theory. *Proc. Roy. Soc. London A*, 462(2069):1493–1514, 2006.

[4] S. H. Faria. Creep and recrystallization of large polycrystalline masses. Part III: continuum theory of ice sheets. *Proc. Roy. Soc. London A*, 462(2073):2797–2816, 2006.

[5] S. H. Faria, K. Hutter, and G. M. Kremer. Reply to Gagliardini's comment on 'Creep and recrystallization of large polycrystalline masses' by Faria and co-authors. *Proc. Roy. Soc. London A*, 464(2099):2803–2809, 2008.

[6] S. H. Faria, S. Kipfstuhl, N. Azuma, J. Freitag, I. Hamann, M. M. Murshed, and W. F. Kuhs. The multiscale structure of Antarctica. Part I: inland ice. *Low Temp. Sci.*, 68:39–59, 2009.

[7] S. H. Faria, S. Kipfstuhl, and A. Lambrecht. *The EPICA-DML Deep Ice Core.* Springer, Berlin, 2018.

[8] S. H. Faria, G. M. Kremer, and K. Hutter. Creep and recrystallization of large polycrystalline masses. Part II: constitutive theory for crystalline media with transversely isotropic grains. *Proc. Roy. Soc. London A*, 462(2070):1699–1720, 2006.

[9] S. H. Faria, I. Weikusat, and N. Azuma. The microstructure of polar ice. Part II: state of the art. *J. Struct. Geol.*, 61:21–49, 2014.

[10] H. J. Frost and M. F. Ashby. *Deformation-mechanism Maps.* Pergamon, Oxford, 1982.

[11] O. Gagliardini. Comment on the papers 'creep and recrystallization of large polycrystalline masses' by faria and co-authors. *Proc. Roy. Soc. London A*, 464:289–291, 2008.

[12] G. Gödert. A mesoscopic approach for modelling texture evolution of polar ice including recrystallization phenomena. *Ann. Glaciol.*, 37:23–28, 2003.

[13] F. J. Humphreys and M. Hatherly. *Recrystallization and Related Annealing Phenomena.* Pergamon, Oxford, 2nd edition, 2004.

[14] T. H. Jacka and J. Li. Flow rates and crystal orientation fabrics in compression of polycrystalline ice at low temperatures and stresses. In T. Hondoh, editor, *Physics of Ice Core Records*, pages 83–102. Hokkaido University Press, Sapporo, 2000.

[15] B. Kamb. Experimental recrystallization of ice under stress. In H. C. Heard, I. Y. Borg, N. L. Carter, and C. B. Raleigh, editors, *Flow and Fracture of Rocks*, number 16 in Geophysical Monograph, pages 211–241. American Geophysical Union, Washington, DC, 1972.

[16] S. Kipfstuhl, S. H. Faria, N. Azuma, J. Freitag, I. Hamann, P. Kaufmann, H. Miller, K. Weiler, and F. Wilhelms. Evidence of dynamic recrystallization in polar firn. *J. Geophys. Res.*, 114:B05204, 2009.

[17] S. Kipfstuhl, I. Hamann, A. Lambrecht, J. Freitag, S. H. Faria, D. Grigoriev, and N. Azuma. Microstructure mapping: A new method for imaging deformation-induced microstructural features of ice on the grain scale. *J. Glaciol.*, 52(178):398–406, 2006.

[18] J. W. Martin, R. D. Doherty, and B. Cantor. *Stability of Microstructure in Metallic Systems.* Cambridge University Press, Cambridge, 2nd edition, 1997.

[19] M. S. Paterson. A granular flow theory for the deformation of partially molten rock. *Tectonophysics*, 335:51–61, 2001.

[20] S. Piazolo, M. Montagnat, F. Grennerat, H. Moulinec, and J. Wheeler. Effect of local stress heterogeneities on dislocation fields: Examples from transient creep in polycrystalline ice. *Acta Materialia*, 90:303–309, 2015.

[21] J.-P. Poirier. *Creep of Crystals.* Cambridge University Press, Cambridge, 1985.

[22] D. H. Richards, S. S. Pegler, S. Piazolo, and O. G. Harlen. The evolution of ice fabrics: A continuum modelling approach validated against laboratory experiments. *Earth Planet. Sci. Lett.*, 556:116718, 2021.

[23] P. Skemer, I. Katayama, Z. Jiang, and S. ichiro Karato. The misorientation index: Development of a new method for calculating the strength of lattice-preferred orientation. *Tectonophysics*, 411(1):157–167, 2005.

[24] A. P. Sutton and R. W. Balluffi. *Interfaces in Crystalline Materials.* Clarendon, Oxford, 1995.

[25] H.-R. Wenk. Texture and anisotropy. In S. I. Karato and H.-R. Wenk, editors, *Plastic Deformation of Minerals and Rocks*, volume 51 of *Reviews in Mineralogy and Geochemistry*, pages 291–330. Mineralogical Society of America and Geochemical Society, Washington, DC, 2004.

[26] S. White. Geological significance of recovery and recrystallization processes in quartz. *Tectonophysics*, 39(1–3):143–170, 1977.

---

## Referee Report (RR3)

**Journal:** *The Cryosphere*
**Manuscript:** tc-2021-118: Ice fabrics in two-dimensional flows: beyond pure and simple shear
**Authors:** Daniel H. M. Richards, Samuel S. Pegler, and Sandra Piazolo
**Reviewer:** Ed Waddington

**1 Overview**

I was invited to comment on this manuscript late in the review process. I have read version 3 of the manuscript, and the authors' reply to the previous reviews.

**1.1 Author responses to prior reviews**

Previous reviewers of this manuscript have identified several points that needed to be addressed.

- One of those points for clarification was the concept that SpecCAF does not directly model crystal-level processes such as dislocation densities, slip on basal planes, recrystallization, or crystal-crystal interactions. The authors have clarified that SpecCAF is an empirical continuum model for the evolution of $\rho^*(\mathbf{x}, t, \mathbf{n})$, the mass fraction of grains at position $\mathbf{x}$ at time $t$ with c axes directed into a a solid angle d$\mathbf{n}$ around direction $\mathbf{n}$.
  Slip on basal planes, rotational recrystallization, and grain-boundary rotation are all incorporated in principle as continuum processes based on gradients in the continuum description, i.e. SpecCAF is essentially an empirical model with what might be called a model *shape* set by equations (3), (4) and (5), and the coefficients $\lambda$ and $\beta$ on the terms are set empirically by comparison with fabrics observed in samples whose deformation histories are known or understood.
  To me, this seems to be the same in principle as choosing to fit an exponential *shape* to a data set, where the data determine the prefactor and the exponent. Choosing a good *shape* and a good training data set are key to establishing a good fit over a wide range of circumstances.

- Reviewers were concerned that SpecCAF used the Taylor assumption, in which all grains experienced the same strain rate. The authors have clarified that, following Faria et al. (2008), the SpecCAF model assumes only that the material holds together such that individual grains (if they were explicitly followed, which they are not) would merely retain their position relative the surrounding continuum. There is no restriction imposed on how

individual grains strain, rotate, or recrystalize, relative to their neighbors; the only restrictions are on their *species*, defined as other grains with similar orientations.

- Reviewers reminded the authors that not all previous lab tests were restricted to pure or simple shear; some previous lab experiments imposed stress and strain patterns that were combinations of pure and simple shear. The authors have incorporated the suggested references in new discussion in Section 2.1.2, and modified their claim to be the first group to study this.

- Reviewers asked how the evolving fabric was coupled to the applied strain-rate fields. The authors have clarified that they are not yet coupled; that is a goal for future work.

In my view, this paper is a commendable analysis of evoluton of fabric (CPO) in 2-D under a wide range of temperature $T$ and flow regime as characterized by a vorticity number $\mathcal{W}$ in Equation (1). In my view, the paper is suitable for publication in *The Cryosphere*, pending minor revisions that can be negotiated with the scientific editor.

**1.2   My questions**

The previous reviewers are all clearly experts in anisotropic fabric development and ice-sheet flow, and I think they have done a good job of identifying technical issues and concerns. While I have some familiarity with the field, I will address mainly the likelihood that the manuscript will speak effectively to colleagues and students who are not as well-versed in the topic as the reviewers. Since these points are less germane to the scientific integrity of the manuscript, and more germane to the readability and potential readership, I expect that you can discuss with the editor the rigor with which you should follow them.

- Page 1, Line 24:
  What is meant by *a uniform spectrum*?

- Effective strain rate $\dot{\gamma}$ is introduced in Equation (8), where it is defined in terms of the strain-rate tensor, which is derived in turn from the velocity-gradient tensor. However, strain $\gamma$ itself just appears without an explanation in the caption for Figure 7.

  This may be a concern, because the manuscript deals with some large finite strains that may not be simply related to the history of strain rate. It is not obvious how (or if) the strain rate is integrated over time to get the strain. I assume Lagangian or Eulerian finite-strain tensors are involved?

- Are the finite strains in SpecCAF calculated in a way that is compatible with the calculated finite strains from lab tests, and inferred from Antarctic data sets such as Figure 8?
  For example, in lab tests to large finite strains, the shape of the sample changes significantly, and even if the applied force or the applied stress is held constant, the strain rate is time-dependent. Is the strain history inferred directly from snapshots of the shape, rather than from integrating the strain rate?
  In the Antarctic Ice Sheet, the vertical strain rate is inferred from the horizontal velocity divergence through continuity, and then is assumed to be uniform through the upper 25% of the depth. How is the strain profile then calculated for ice as it moves downward?

- Equation (2)
  At first reading, it was unclear to me whether the *mass fraction* $\rho^*(\mathbf{x}, t, \mathbf{n})$ was a (dimensional) mass, or a (nondimensional) fraction. I figured out that it must be a mass, because it integrates to $\rho^*(\mathbf{x}, t)$, which appears to be a dimensional mass, rather than integrating to unity over the sphere; however, perhaps that could be made clearer to help your readers avoid an interruption in smooth reading.

- In order to help me read the paper more efficiently, I made a table of variables with definitions and notes about where they first appear. I expect that such a table of variables would be helpful for other readers, and could increase the readership of the paper.

- Exploring the full range of two-dimensional responses to two-dimensional loading is an important step, and I think the authors are making a useful contribution. However, I also expect that minor perturbations in that two-dimensional flow may create fabrics that generate instabilities causing growing nonzero strain rates and flow in the third dimension. This is a question that could also motivate further work.

**1.3 Copy editorial points and clarity**

- Line 44:
  The author's name is Takeo Hondoh, so the reference should be simply *Hondoh, 2000*.
  In the **References** section, at line 643, the citation should be *Hondah, T., Nature and behavior ...*, i.e. only the initial, to be consistent with all the other references.

- The manuscript uses vector notation, indicial notation, and the summation convention, but does not explain these concepts from continuum-mechanics to readers who may be unfamiliar with them. While a couple of dozen or so people in the community will understand what you are doing, this oversight is liable to dissuade other readers (such as new graduate students) from reading beyond equation (1). A couple of sentences could rectify this.

- There appears to be some oversight or misunderstanding about the difference between *maximum* and *maxima*. *Maxima* is a plural word meaning (if we were to purge the latin forms), *maximums*. Just as it makes no sense to talk about *a single maximums*, it makes no sense to talk about *a single maxima*, or *a single-maxima fabric*.
  The expression *a double maxima* is also problematic, because it could be interpreted to mean four or more peaks. A *double maximum* more clearly indicates two peaks.
  The top row in Figure 1 has it right - *single-maximum* fabric, and *double-maximum* fabric.

- Line 240:
  What is meant by *fully resolved experiments*?

- Figure 5
  To my eyes, there appears to be a slight change in the character of the vorticity number inside a ghost circle at 80 degrees South. Is this a relic of *the Pole hole* caused by polar orbits that turn at 80 South? Does this affect the quality of the data shown?

I will spare you a complete line-by-line list of other grammatical suggestions; however, I hope you will see the merit of checking throughout the text for other examples of these points. Making your text easier to read can only enhance your readership numbers.

The English language is fraught with many rules that often don't appear to make a lot of sense, and there are differences of opinion among groups who have differing communication aims, such as journalists, popular-media editors, poets, novelists, and scientists; however, some rules can eliminate ambiguities and make scientific text easier to read. The following points address recommendations on using hyphens and strings of ideas, in order to make your text more accessible to readers, and therefore helping you to create a more easily understandable, and ultimately more memorable and important paper.

- Hyphenation
  A hyphen should be used in a compound adjective (an adjective and a noun) that modifies another noun,
  e.g. line 15 and elsewhere - *ice-flow dynamics.*

  A hyphen should *not* be used between a stand-alone noun (subject or object) and an adjective that modifies the noun,
  e.g. line: 10 and elsewhere - no hyphen in *strain scales.* e.g. Figure 1 caption and elsewhere - no hyphen in *(d) shows a single maximum with ...*

  Generally there should be no hyphen after an adjective or adverb that ends in 'y' . e.g. line 9 and elsewhere - *highly rotational*

- Lists
  When comparing a string of ideas in text, the ideas are easier for readers to grasp quickly when they have equivalent and parallel grammatical structures.

  For example, in the Abstract,
  *The use of our model in large-scale ice flow models as well as for interpreting fabrics observed in ice cores and seismic anisotropy,*
  introduces two ideas, but the first is written as the phrase *in ice flow models,* while the second is written as the clause *for interpreting fabrics observed in ice cores and seismic anisotropy.*
  Can you rewrite both ideas as phrases, or both ideas as clauses, i.e. neither or both should contain a verb form?

- Page 1, line 4
  *... in both compression and simple shear, ...* is unclear.
  Do you mean *... in both pure shear and simple shear, ...*?
  You are describing *deformational regimes* in terms of strain rather than stress. While there can be compressive stress in all directions (pressure), there can be no volumetric compressive strain for incompressible ice (neglecting elasticity). Perhaps as a community we are sloppy in our terminology, by calling it a *compression test* when we set a weight on top of an ice slab, because that slab experiences compressive deviatoric stress on one axis, but extensile deviatoric stress on other axes. (While we can't change the world, we can each make our own writing clearer.)
  It would be better to choose one wording, then stick with that throughout the manuscript. (I think the text gets it right later at line 21.)

---

## Author Response (AR2)

**Ice fabrics in two-dimensional flows: beyond pure and simple shear**

Authors: Daniel H. Richards, Samuel S. Pegler, and Sandra Piazolo

**Collated Response to Reviewers**

Our responses are in blue throughout.

**Review 1: Ed Waddington**

I was invited to comment on this manuscript late in the review process. I have read version 3 of the manuscript, and the authors' reply to the previous reviews.

1.1     Author responses to prior reviews

Previous reviewers of this manuscript have identified several points that needed to be addressed.

One of those points for clarification was the concept that SpecCAF does not directly model crystal-level processes such as dislocation densities, slip on basal planes, recrystallization, or crystal-crystal interactions. The authors have clarified that SpecCAF is an empirical continuum model for the evolution of (x; t; n), the mass fraction of grains at position x at time t with c axes directed into a solid angle dn around direction n.

Slip on basal planes, rotational recrystallization, and grain-boundary rotation are all incorporated in principle as continuum processes based on gradients in the continuum description, i.e. SpecCAF is essentially an empirical model with what might be called a model shape set by equations (3), (4) and (5), and the coefficients and on the terms are set empirically by comparison with fabrics observed in samples whose deformation histories are known or understood.

To me, this seems to be the same in principle as choosing to fit an exponential shape to a data set, where the data determine the prefactor and the exponent. Choosing a good shape and a good training data set are key to establishing a good t over a wide range of circumstances.

Reviewers were concerned that SpecCAF used the Taylor assumption, in which all grains experienced the same strain rate. The authors have clarified that, following Faria et al. (2008), the SpecCAF model assumes only that the material holds together such that individual grains (if they were explicitly followed, which they are not) would merely retain their position relative the surrounding continuum. There is no restriction imposed on how individual grains strain,

rotate, or recrystallize, relative to their neighbors; the only restrictions are on their species, defined as other grains with similar orientations.

Reviewers reminded the authors that not all previous lab tests were restricted to pure or simple shear; some previous lab experiments imposed stress and strain patterns that were combinations of pure and simple shear. The authors have incorporated the suggested references in new discussion in Section 2.1.2, and modifed their claim to be the first group to study this.

Reviewers asked how the evolving fabric was coupled to the applied strain-rate elds. The authors have clarified that they are not yet coupled; that is a goal for future work.

In my view, this paper is a commendable analysis of evolution of fabric (CPO) in 2-D under a wide range of temperature T and ow regime as characterized by a vorticity number W in Equation (1). In my view, the paper is suitable for publication in The Cryosphere, pending minor revisions that can be negotiated with the scientific editor.

We thank the reviewer for the detailed consideration of our manuscript and positive comments. Please find our description of minor revisions below.

1.2     My questions

The previous reviewers are all clearly experts in anisotropic fabric development and ice-sheet flow, and I think they have done a good job of identifying technical issues and concerns. While I have some familiarity with the field, I will address mainly the likelihood that the manuscript will speak effectively to colleagues and students who are not as well-versed in the topic as the reviewers. Since these points are less germane to the scientific integrity of the manuscript, and more germane to the readability and potential readership, I expect that you can discuss with the editor the rigor with which you should follow them.

Page 1, Line 24: What is meant by a uniform spectrum?

We mean to highlight by this that we are exploring the continuous space of deformation regimes as vorticity number increases, rather than the isolated cases of pure and simple shear. In light of this comment we have reformulated this part to be clearer:

"The objective of the present paper is to use the fabric evolution model SpecCAF (Richards et al. 2021) to take a step away from the isolated conditions of irrotational deformation and simple shear where the model has been validated, and explore the continuous space of deformation regimes lying between these cases. We also extrapolate to deformation regimes more rotational than simple shear.

Effective strain rate is introduced in Equation (8), where it is defined in terms of the strain-rate tensor, which is derived in turn from the velocity-gradient tensor. However, strain itself just appears without an explanation in the caption for Figure 7. This may be a concern, because the

manuscript deals with some large finite strains that may not be simply related to the history of strain rate. It is not obvious how (or if) the strain rate is integrated over time to get the strain. I assume Lagranian or Eulerian finite-strain tensors are involved?

Are the finite strains in SpecCAF calculated in a way that is compatible with the calculated finite strains from lab tests, and inferred from Antarctic data sets such as Figure 8?

We calculate the strain by computing the integral of the strain-rate with time, with the strain-rate defined as in eq (8). Different measurements are used by different groups doing laboratory experiments, but they can be converted to an effective strain for comparison with the work here by using a scaling factor. We have added a paragraph on this after eq (12) (where we define the strain as explained here)

For example, in lab tests to large finite strains, the shape of the sample changes significantly, and even if the applied force or the applied stress is held constant, the strain rate is time-dependent. Is the strain history inferred directly from snapshots of the shape, rather than from integrating the strain rate?

Within the context of SpecCAF, the strain can always be evaluated as the integral of the strain rate over time. Within the context of experiments, the strain is achieved by fixed movement of pistons. For example, during uniaxial compression, the sample shape changes and therefore the strain rate will not be constant. However, the finite strain for each deformation step ("snapshot") can always be correctly retrieved from experiments. As such results are compatible.

In the Antarctic Ice Sheet, the vertical strain rate is inferred from the horizontal velocity divergence through continuity, and then is assumed to be uniform through the upper 25% of the depth. How is the strain profile then calculated for ice as it moves downward?

Figure 5 was computed to illustrate the variability of vorticity number at first order across the ice sheet based on simple assumptions. For Fig 5 of the Antarctic ice sheet, we are calculating the vorticity number of ice flow using the surface deformation field. this calculation a conservative estimate as we assume no slip at the base. We are not using this figure to calculate any strain profiles. We agree it would be interesting to use this model to track ice parcels through an ice sheet.
We have now emphasised the purpose and assumption of the figure in the text and figure captions.

Equation (2): At first reading, it was unclear to me whether the mass fraction rhostar(x; t; n) was a (dimensional) mass, or a (nondimensional) fraction. I figured out that it must be a mass, because it integrates to (x; t), which appears to be a dimensional mass, rather than integrating

to unity over the sphere; however, perhaps that could be made clearer to help your readers avoid an interruption in smooth reading.

We agree that a clarification of this is helpful. We do introduce it as a mass fraction in (2), and then non-dimensionalise it in section 3.3, so that, for the rest of the paper, it is a nondimensional fraction. To better clarify the distinction between the mass fraction and its non-dimensional version, we have introduced a new symbol $f^* = \rho^*/\rho$ to denote the non-dimensional orientation distribution function and use this throughout (previously we had continued to use the same symbol following a non-dimensionalisation).

In order to help me read the paper more efficiently, I made a table of variables with definitions and notes about where they first appear. I expect that such a table of variables would be helpful for other readers, and could increase the readership of the paper.

We have added this as Table 1.

Exploring the full range of two-dimensional responses to two-dimensional loading is an important step, and I think the authors are making a useful contribution. However, I also expect that minor perturbations in that two-dimensional flow may create fabrics that generate instabilities causing growing nonzero strain rates and ow in the third dimension. This is a question that could also motivate further work.

Yes, this is an interesting comment

1.3     Copy editorial points and clarity

Line 44: The author's name is Takeo Hondoh, so the reference should be simply Hondoh, 2000.

In the References section, at line 643, the citation should be Hondah, T., Nature and behavior . . . i.e. only the initial, to be consistent with all the other references.

Thank you for pointing this out, we have corrected this.

The manuscript uses vector notation, indicial notation, and the summation convention, but does not explain these concepts from continuum-mechanics to readers who may be unfamiliar with them. While a couple of dozen or so people in the community will understand what you are doing, this oversight is liable to dissuade other readers (such as new graduate students) from reading beyond equation (1). A couple of sentences could rectify this.

We have added a paragraph after eq (1) to explain this:

"As a note for people unfamiliar, in Eq (1) we have used both summation notation $W_{ij}$ and vector notation $W$. $W_{ij}$ is a 2nd-rank tensor (shown by the number of indices) and the operation $W_{ij}W_{ij}$, indicating summation over the repeated indices, is the tensor inner product $W:W$.

There appears to be some oversight or misunderstanding about the difference between maximum and maxima. Maxima is a plural word meaning (if we were to purge the latin forms), maximums. Just as it makes no sense to talk about a single maximums, it makes no sense to talk about a single maxima, or a single-maxima fabric.

The expression a double maxima is also problematic, because it could be interpreted to mean four or more peaks. A double maximum more clearly indicates two peaks.

The top row in Figure 1 has it right - single-maximum fabric, and double-maximum fabric.

This has been corrected throughout the document

Line 240: What is meant by fully resolved experiments?

We agree, this was unclear and replaced with "laboratory experiments for which parameters such strain, deformation rate, and temperature are known".

Figure 5: To my eyes, there appears to be a slight change in the character of the vorticity number inside a ghost circle at 80 degrees South. Is this a relic of the Pole hole caused by polar orbits that turn at 80 South? Does this affect the quality of the data shown?

This is correct, Mouginot et al., 2019 use a different method to derive the surface velocities within this circle. This figure has been updated based on comments by other reviewers and the circle is no longer visible.

With regard to data quality effects: all the data is coloured by the relative error estimated from averaging over a 10x10 block, so any low-quality data should be coloured as white (I.e. not visible)

I will spare you a complete line-by-line list of other grammatical suggestions; however, I hope you will see the merit of checking throughout the text for other examples of these points. Making your text easier to read can only enhance your readership numbers.

We have gone through the document to check over the grammar and spelling throughout

The English language is fraught with many rules that often don't appear to make a lot of sense, and there are differences of opinion among groups who have differing communication aims, such as journalists, popular-media editors, poets, novelists, and scientists; however, some rules can eliminate ambiguities and make scientific text easier to read. The following points address recommendations on using hyphens and strings of ideas, in order to make your text more accessible to readers, and therefore helping you to create a more easily understandable, and ultimately more memorable and important paper.

Hyphenation:

A hyphen should be used in a compound adjective (an adjective and a noun) that modifies another noun, e.g. line 15 and elsewhere – ice-flow dynamics.

A hyphen should not be used between a stand-alone noun (subject or object) and an adjective that modifies es the noun, e.g. line: 10 and elsewhere - no hyphen in strain scales. e.g. Figure 1 caption and elsewhere - no hyphen in (d) shows a single maximum with . . .

Generally there should be no hyphen after an adjective or adverb that ends in 'y' . e.g. line 9 and elsewhere - highly rotational

We appreciate the reviewer taking the time to make these suggestions and have gone through the document to correct this.

Lists:

When comparing a string of ideas in text, the ideas are easier for readers to grasp quickly when they have equivalent and parallel grammatical structures.

For example, in the Abstract,

*The use of our model in large-scale ice flow models as well as for interpreting fabrics observed in ice cores and seismic anisotropy,* introduces two ideas, but the first is written as the phrase *in ice flow models,* while the second is written as the clause *for interpreting fabrics observed in ice cores and seismic anisotropy*.

Can you rewrite both ideas as phrases, or both ideas as clauses, i.e. neither or both should contain a verb form?

We thank the reviewer for this suggestion. We have corrected this to "The use of our model for addition to large-scale ice flow models and for interpreting fabrics observed in ice cores and seismic anisotropy will provide new tools supporting the community in predicting ice flow in a changing climate." We have also gone through the document to improve the use of language throughout.

Page 1, line 4

. . . in both compression and simple shear, . . . is unclear. Do you mean . . . in both pure shear and simple shear, . . . ?

We have corrected this to unconfined compression and simple shear – as the experiments we compared against were performed at these conditions. We hope this clarifies the precise deformational regime. We didn't say pure shear as this is understood to be a 2D deformation.

You are describing deformational regimes in terms of strain rather than stress. While there can be compressive stress in all directions (pressure), there can be no volumetric compressive strain for incompressible ice (neglecting elasticity). Perhaps as a community we are sloppy in our terminology, by calling it a compression test when we set a weight on top of an ice slab, because that slab experiences compressive deviatoric stress on one axis, but extensile deviatoric stress on other axes. (While we can't change the world, we can each make our own writing clearer.)

It would be better to choose one wording, then stick with that throughout the manuscript. (I think the text gets it right later at line 21.)

We have gone through the document and have removed any reference to 'confined compression' or '2D compression' instead only referring to pure shear. However, we keep the term uniaxial compression to refer to compression along a single axis with the other 2 axes then experiencing extension, as this term is used as such in the literature.

**Review 2: Sergio Faria**

Dear Editor and Authors,

The manuscript uses the numerical model SpecCAF to simulate and classify crystallographic preferred orientations (CPOs) generated by a wide range of two-dimensional deformation regimes. It is a follow-up of [22]. The text is well written and self-contained. The work has good scientific quality and presents interesting results. I enjoyed reading it. There are however, several clarity issues that require careful revision. None of these issues affect the main results and conclusions of the work, which I recommend for publication after revision. **Specific comments:**

**Lines 20–21:** To be fair, the most studied deformation regime to date in relation to ice fabrics has been uniaxial (vertical) compression, probably as much or even more than pure and simple shear.

We have added this.

**Line 24:** It would be nice to explain why "ice flow is commonly modelled in the two dimensional $x - z$ plane", and which $x - z$ plane is chosen.

Added: One way to model ice flow is to simplify it to the two-dimensional x-z plane (along the flow direction and the vertical, e.g Martin el al. 2009). This is done to understand vertical variation and compare to ice core profiles.

**Line 26:** Delete the spurious "below".

Done.

**Lines 28–32:** Concerning the four open questions to be answered: The first two have been considered by [15] through the combination of theory with experimental extrapolation. The third question is unclear: which "steady state" do you mean? Strain rate steady state? Stationary CPO? Some other kind of steady state?

Regarding Kamb, we have added a paragraph in the introduction reviewing this paper. We have clarified the third question (and combined it with the fourth): "Third, how do fabrics evolve at very high strains which have remained inaccessible to laboratory experiments, and at what strain does the fabric reach a steady state?"

**Lines 42–43:** Personally, I find the term "crystal slip" a bit misleading and recommend replacing it with "intracrystalline slip", or even better "dislocation glide" (and climb, if you wish to be general; [10]). If you want to keep term "crystal slip", then please make clear what it means (as it stands, I can only guess). Also "rigid-body rotation" sounds slightly misleading, since the material under consideration is not a rigid body. Better would be simply "rigid rotation".

We have referred to dislocation glide instead of crystal slip. While we see the point of the reviewer that rigid body rotation may be misleading, it is the term used in the microstructural community and therefore we would like to keep it.

**Lines 42–43:** Assuming the established definition of recrystallization as "the formation and migration of high-angle grain boundaries driven by the stored strain energy" [2, 7, 9, 13], it follows that migration and rotation recrystallization are not deformation mechanisms, but rather annealing phenomena. Admittedly, recrystallization of any kind is closely related to strain, being driven by the stored strain energy and affecting the mechanical response of the material. Nevertheless, recrystallization is not a deformation mechanism per se, since it cannot produce strain (change in shape) or rigid rotation in a stressed body [10, 18, 19, 24, 26]. Migration recrystallization describes the motion of grain boundaries *through* the material (i.e., without material movement). Rotation recrystallization describes the formation of a new grain boundary. In this respect, it is worth mentioning that some authors confuse cause and effect by erroneously attributing a material rotation to "rotation recrystallization": Actually, the material rotates by a deformation mechanism like dislocation glide and climb, and the strain energy stored in the material by this rotation triggers rotation recrystallization, which is the formation of a new grain boundary. The fallacy that recrystallization phenomena were deformation mechanisms is an epidemic pseudodoxy perpetrated by unreliable sources.

We have reworded this to "As ice deforms, the fabric evolves both through dislocation glide along the basal plane, which causes c-axes to rotate, rigid-body rotation which simply rotates grains around the rotation axis, and recrystallization processes which rearrange the grain boundary network.

**Line 43:** Insert "in ice" after "slip".

We have reworded this section, however we have taken care to clarify that basal-slip deformation dominates in ice only.

**Lines 60–63:** The references cited here are not the most suitable. For instance, Piazolo et al. [20] is a very interesting work, but it refers only to transient creep in laboratory and simulations, and it would be reasonable to argue that stress and strain heterogeneities may disappear after the transient phase. As it turns out, that is actually not the case in practice, rather the contrary. Kipfstuhl et al. [16, 17] have observed strong strain heterogeneities in shallow and deep polar ice, while Faria et al. [6] explained those stress/strain heterogeneities through the concept of "a highly strained mantle and a less strained core within a grain." As for the diffusion/dispersion of c-axes by rotation recrystallization, Godert [12] presents a model that simulates the concepts and observations made by previous researchers, while the original concept can actually be traced back to Poirier [21], which was popularized in ice by Alley [1].

We thank the reviewer for suggesting these helpful references. We have changed this section to:

The second recrystallization process is *rotational recrystallization*. This occurs when dislocations recover into subgrain boundaries which, with increasing strain, will develop into grains (Drury et al., 1985). These dislocations tend to be concentrated closer to grain boundaries due to stress heterogeneity, observed in shallow and deep polar ice (Kipfstuhl et al., 2006, 2009) and which can be thought of as an ice grain having a stressed outer `mantle' and a less stressed inner `core' (Faria et al., 2009). Therefore, new grains developing from subgrains, dominantly occur near grain boundaries. The orientation of these new grains is similar to, but slightly different to, the parent grain. With increasing strain, the difference in orientation tends to increase (Halfpenny et al. 2006). This randomisation of orientation acts to diffuse any concentrations in the fabric (Alley, 1992).

**Lines 66–67:** Radar should be mentioned here as well (it is mentioned only later, on Line 92).

We have added this

**Line 75:** The correct citation is "Li et al., 1996". The surname is "Li", the given name is "Jun".

We have corrected this.

**Line 78 and elsewhere:** The plural expression "single maxima" is repeatedly misused in singular contexts in many points of the text. The singular is "single maximum" and its plural is "single maxima". Please do not mix them up.

This was mentioned by another reviewer as well, and we have corrected this

**Figure 1:** Please indicate the principal directions of compression and simple shear. The pole figure (d) is incorrect. The primary cluster should be closer to the centre and the secondary cluster closer to the border of the diagram, at approx. 70° from the primary cluster [1, 15].

We have plotted (d) this way so that the principal directions of deformation do not change across the subfigures. It is rotated 45 degrees from 'normal' simple shear. We have clarified the principal directions in the caption

**Lines 104–105:** The 45° is a theoretical estimate, because observed angles are less than 45° due to the continual rotation of c-axes towards the main compression axis.

We have changed this to: "This process acts to consume grains orientated towards the compression axis and, on its own, grows grains orientated in a ring 45° away from the compression axis (the orientation easiest for basal slip and hence likely to be with the least dislocations). Therefore, the balance of basal-slip deformation and migration recrystallization produces a girdle pattern, with an angle always 45° due to the interaction between the two processes"

**Lines 110–114:** The explanation for the imbalance in cluster strengths seems a bit confusing. The main reason for the imbalance is neither the "vorticity" in simple shear (i.e. gradual rotation of the principal strain axes), nor recrystallization. Rather, the imbalance is mainly derived from the fact that, for simple shear, the secondary cluster is unstable, whereas the primary cluster is stable. In other words, c-axes in the primary cluster stay there, while c-axes in the secondary cluster quickly rotate away from it by usual strain-induced lattice rotation. If migration recrystallization were causing the imbalance, more recrystallization would imply a weaker secondary cluster, which is contrary to observation (the secondary cluster actually gets weaker when there is less recrystallization). The function of migration recrystallization is to make the secondary cluster more defined, by consuming the grains with c-axes that rotate away from it and move towards the principal axis of compression ("hard-glide orientations"). The "vorticity" of simple shear generally plays a very minor role, since it is much slower than the effects of c-axis rotation and recrystallization.

The c-axes of the secondary cluster do rotate away from the orientation preferred by migration recrystallization. But this is caused by a combination of vorticity and basal-slip deformation. If you consider the velocity gradient we use:

$$\nabla u = \begin{pmatrix} 1 & W \\ -W & -1 \end{pmatrix}$$

We are keeping the deformation tensor constant and adding vorticity to move from pure to simple shear. So for the case of pure shear, there is a balance between basal-slip deformation rotating c-axes towards the compression axis, and migration recrystallization consuming crystallites towards the compression axis and producing them at 45 degrees.

With the addition of vorticity using the velocity gradient above, there is no rotation of the deformation axes. So the only change is the addition of vorticity: here the primary cluster remains stable as vorticity acts to move c-axes away from the compression axis, so it balances basal-slip deformation. For the secondary cluster the vorticity acts in the same direction as the basal-slip deformation

While indeed migration recrystallization makes the secondary cluster stronger, in our view it is important to isolate as much as possible the effect of increasing vorticity from any rotation of the deformation axes. To make this clear we have re-worded this part to:

"This pattern is similar to a double-maximum but the presence of vorticity in simple shear causes an in-balance in cluster strengths. For the stronger, primary cluster the vorticity acts to

move c-axes in the opposite direction to the basal-slip deformation, resulting in a stable position. For the weaker, secondary cluster the vorticity and basal-slip deformation both rotate c-axes towards the compression axis."

**Line 118:** It could be mentioned here that Kamb [15] related ice fabrics to deformation regimes using a somewhat related measure, which he called the "stress character".

This has been added.

**Figure 2:** Please be consistent and use either "rigid rotation" or "pure rotation", but not both.

In this paper we are using these to refer to separate concepts: rigid (-body) rotation refers to the effect of vorticity on the fabric, pure rotation refers to a deformation condition which is solely rotational. However we have updated the caption to clarify, replacing pure rotation with "a purely rotational deformation"

**Figure 4:** Why have you amplified that much the Gaussian bump? I am afraid that the high vorticity numbers reported in this figure may be derived from such an extreme amplification of the bump.

We have modified this figure so that the height of the gaussian bump is unchanged (a tenth of the flow domain height) compared to the ISMIP benchmark. We have only reduced the width of the gaussian. Upon reflection, the height of the bump in the previous figure was unrealistic but we believe that the presence of sharp bumps or features is likely common at the base of ice-sheets.

**Figure 4:** Why have you chosen $n = 1$ instead of $n = 3$ in this example? Intuitively, one would expect a realistic modelling of ice flow with high vorticity numbers to use the non-Newtonian description with $n = 3$. What would be the effect of $n = 3$ on the vorticity numbers in this simulation?

We have updated this figure to use n=3, as the reviewer points out this is more representative.

**Figure 5:** This figure intrigues me. Maybe I misunderstood it? I have doubts about the use of shallow ice approximation at the transition from grounded ice to ice shelf...Besides, we know from detailed modelling and ice-core observations that the dominant deformation regime for ice shelves is non-rotational, asymmetric horizontal extension; not simple shear as indicated in the map.

The shallow ice approximation is just used to introduce some vertical shear and makes the vorticity number lower, giving a conservative estimate. Therefore, we say that for regions of grounded ice the vorticity number shown is valid to 25% depth, for regions with greater basal sliding it should be valid to a greater depth. We have added this explanation in the text.

We agree that extension is present in these locations, however this has been seen in combination with a simple shear component (with the shear plane perpendicular to the flow direction) e.g. (Lutz et al., 2020) for the Ross ice shelf.

However, based on your comments we double-checked this figure and saw that it disagreed with measurements in some locations as you say, e.g. the Amery ice shelf. We have updated it by recalculating the velocity gradients based on differentiating the averaged values rather than the local ones, and this gives results more in line with what is reported and would be expected. Consequently, we have also updated the text in referenced to this.

**Lines 238–239:** That is correct indeed. At this point I have to digress to do something that I very rarely do—because it causes me great displeasure—which is to correct erroneous statements by another reviewer. In this particular case I feel obliged to do so, to rectify harmful and unfair criticism to the work under review. The unfair claims by the Reviewer are:

> *The model, that derives from previous works of Faria et al. (2006-I,II,III), assumes an homogeneous strain rate, meaning that each crystal is submitted to the same strain rate. This hypothesis, apparently not clearly stated in any of those works, has been shown by Gagliardini (2008) in its response to Faria et al. (2006) to correspond to a Taylor-type of approximation, meaning uniform strain.*

There are several errors in that statement. First, the Reviewer cites a comment by Gagliardini [11], but fails to cite the subsequent response [5] that proved the falsity of all Gagliardini's comments.

Second, it is true that the SpecCAF model is ultimately based upon the theory of Continuous Diversity developed by Faria et al. [3, 4, 8], but the Reviewer's claim that the theory of Continuous Diversity assumes a homogeneous strain rate for each grain (so-called "Taylor-type approximation" or "uniform strain") is clearly fallacious: it represents a complete disregard for the fundamental principles of continuum mechanics.

The theory of Continuous Diversity (CD) describes the large-scale ("macroscopic") flow of a glacier or ice sheet. As any other continuum theory, all fields and gradients in the CD theory are spatially defined on that large scale, which is many orders of magnitude larger than the grain scale. Therefore, just as the strain rate in fluid dynamics does not impose any constraint, hypothesis or approximation on the motion of individual molecules, the strain rate in the theory of Continuous Diversity does not impose any constraint, hypothesis or approximation on the deformation of individual grains: every grain is free to deform as inhomogeneously as needed. In plain mathematical terms, if $dx$ defines an infinitesimal distance in the continuum (upon which all spatial gradients, including the strain rate, are defined) and $D$ is the average grain size, then $dx \gg D$.

We thank the reviewer for supporting our line of reasoning here.

**Figure 7:** Please explain the grey arrow in the figure caption.

We have added to the caption: "The grey arrow shows the secondary cluster transposed from the pole figure to the γ - θ diagram."

**Figure 7:** In the caption, please replace "principal axes of deformation" with "principal strain axes". The former expression does not make much sense when there is rigid rotation.

Thank you for spotting this, we have corrected it.

**Line 304:** Wrong figure reference. It should be "Fig. 7b", not "Fig. 8b".

Corrected

**Equation 10:** I am confused here. The non-dimensional velocity gradient defined in (10) does not seem compatible with the non-dimensional velocity gradient derived from the definitions (8) below, for its symmetric and skew-symmetric parts. If they are compatible, please show that. If not, which one are you using in your simulations?

The velocity gradient is compatible. However from this question it is clear it was not sufficiently well explained. Therefore, we have modified the text to define a dimensional velocity gradient first which we then non-dimensionalise. We have also included the symmetric and skew-symmetric parts for clarity. We hope this clears up any confusion that may arise for a reader.

**Line 321:** I guess you mean −5C, not −10C, right?

Thank you, we have corrected this.

**Lines 325–326:** The positions of the clusters for W= 1 (simple shear) seem way off from the observed positions in the real world...Why? The primary cluster should be close to vertical (centre of the diagram, $\vartheta = 0$ ) and the secondary cluster close to horizontal ≈-70(at around 70 from the primary cluster, that is, $\vartheta$= 70 ). Are you rotating the fabric backwards to remove the vorticity and transform the simple shear into pure shear? Please clarify.

Yes, we have rotated the fabric such that the principal strain axes remain constant as we increase the vorticity number. We have clarified this after eq (10).

**Line 327:** The secondary cluster is consumed by "c-axis rotation", not "migration recrystallization".

Indeed, c-axis rotation rotates these grains to an 'unfavourable' orientation, and at this orientation these grains are consumed by migration recrystallization to grow grains more favourably orientated.

We realize that the wording is misleading. Hence we now write: "At high temperatures with a double-maximum, as W increases the clusters are moved by the rotational component of the

deformation. In combination with basal-slip deformation, this results in one stable and one unstable cluster (see section 2.1.3). The c-axes of the unstable cluster rotate under basal-slip deformation and vorticity such that they are at an orientation where they are consumed by migration recrystallization as strain increases, leading to a single-maximum at high strains…"

**Lines 329–330:** This statement may need revision, depending on the reactions to the comments to Figures 4 and 5 mentioned above. In any case, "prevalent" is a too strong word.

We have changed this to "We also show in Fig. 8 analysis of fabrics produced in highly rotational (W > 1) deformation regimes, which we have shown to occur (Figs. 4,5)."

**Lines 335–336:** The *J*-index as a stand-alone measure of anisotropy has several problems and is considered unreliable [23, 25]. The former reference proposes the use of an *M*-index based on misorientations. Within the framework of a continuum theory with continuous diversity of the type presented here, the definitions and combinations of various anisotropy indices commonly used in ice-core fabric studies are discussed in [4].

We thank the reviewer for suggesting this, we agree that the M-index is a more useful measure. It could be possible to calculate this directly from the odf, though to our knowledge it has not been done before. However, our attempts to do so were very numerically expensive as the calculation involves integrating numerically over the orientation space twice. This is particularly a problem as to plot the steady state figure in this paper, we need to calculate the J or M index for each strain value and each vorticity number and temperature, so approximately 50x50x1000 times. Because of this we think it is best to remain with the simplicity and numerical efficiency of calculating the J-index. However, we have added some comments on the reliability of the J-index after its definition:

"Although the M-index can also be used to measure fabric strength and may be more reliable (Skemer et al., 2005) the J-index can be calculated very efficiently, enabling exploration of the parameter space used in this paper."

**Figure 8:** I recommend adding contour lines or colour steps, as in Fig. 9 or 12, because the smooth colour gradations vary on screen and particularly on print, making it difficult to see the oscillations in the fabric patterns.

Done

**Figure 10:** Same question as before in Lines 325–326. I see an angle close to 30° for the primary cluster for strain = 1 (c) in simple shear (W = 1). Why? Are you rotating the fabric backwards to remove the vorticity and transform the simple shear into pure shear? In real observations (experiment or ice cores) this angle is close to zero. Please clarify.

We have kept the deformation constant, such that it is rotated 45 degrees from 'normal' simple shear.

For a velocity gradient of $\begin{bmatrix} 0 & 1 \\ 0 & 0 \end{bmatrix}$ the deformation tensor is $\begin{bmatrix} 0 & 1/2 \\ 1/2 & 0 \end{bmatrix}$ hence the deformation axes are orientated 45 degrees from the x and y axes.

**Figure 11:** Why not plotting W=0? Should not double maxima occur close to 0? They already appear at W==0.1 in Fig. 8!

We appreciate the reviewer's suggestion and considered changing the figures but we have decided to keep the scale as it is (0.1 to 10). We have not plotted W=0 as we are using a log-scale, and we do not want to break the scale as this may make interpretation of the figure more difficult. A double-maxima will occur for W=0, however in the limit of very high strains the double-maxima is only stable for W=0 exactly.

**Figure 12:** This figure is very useful and it should come before Fig. 11.

Done

**Lines 392–394:** In my opinion, the halfway strain is not very intuitive as a measure of fabric development, because it is normalized by the fabric intensity at steady state. That is, if the steady state fabric is strong, the halfway strain will be larger, giving the impression that it takes longer for the fabric to develop, which is not true, because it may actually develop fast, but it has a long way to reach the "fabric steady state". Therefore, a much more useful measure of fabric development is in my opinion the strain to reach a definite fabric strength. This will tell us how fast fabric develops, which is the information we really need for interpreting ice cores and simulations.

We appreciate the reviewer's suggestion, we tried to plot this but, as there are such different fabric intensities across the space, the plotted picture changes with changing the value of fabric strength chosen. However, based on this we have updated the figure to include a plot of the fabric strength at a given strain of 0.5. This allows comparison between this value and the steady-state.

**Line 431:** I am not sure what you mean by "cone-shaped fabric"...Do you mean a single maximum or a girdle?

We mean a girdle, and we have changed this throughout to be consistent

**Lines 434–435:** This conclusion has already been presented by Kamb [15].

We have added this referenced here.

**Lines 446–447:** That is not stated in the cited work by Jacka and Li [14]. In fact, their results indicate that the mechanical steady state depends on stress and temperature.

This reference has been removed and replaced with (Fan et al., 2020) as an example, where they refer to a steady-state stress at a strain of 0.2.

**Line 650:** Please correct this reference. The authors' list is wrong and the reference data are incomplete.

Thank you, we have corrected this.

I hope the Authors and the Editor find these comments useful.
Best regards,
S´ergio Henrique Faria

[1] R. B. Alley. Flow-law hypothesis for ice-sheet modelling. *J. Glaciol.*, 38:245–256, 1992.

[2] R. D. Doherty, D. A. Hughes, F. Humphreys, J. J. Jonas, D. Juul Jensen, M. E. Kassner, W. E. King, T. R. McNelley, H. J. McQueen, and A. D. Rollet. Current issues in recrystallization: a review. *Mater. Sci. Engineer.*, 238:219–274, 1997.

[3] S. H. Faria. Creep and recrystallization of large polycrystalline masses. Part I: general continuum theory. *Proc. Roy. Soc. London A*, 462(2069):1493–1514, 2006.

[4] S. H. Faria. Creep and recrystallization of large polycrystalline masses. Part III: continuum theory of ice sheets. *Proc. Roy. Soc. London A*, 462(2073):2797–2816, 2006.

[5] S. H. Faria, K. Hutter, and G. M. Kremer. Reply to Gagliardini's comment on 'Creep and recrystallization of large polycrystalline masses' by Faria and co-authors. *Proc. Roy. Soc. London A*, 464(2099):2803–2809, 2008.

[6] S. H. Faria, S. Kipfstuhl, N. Azuma, J. Freitag, I. Hamann, M. M. Murshed, and W. F. Kuhs. The multiscale structure of Antarctica. Part I: inland ice. *Low Temp. Sci.*, 68:39–59, 2009.

[7] S. H. Faria, S. Kipfstuhl, and A. Lambrecht. *The EPICA-DML Deep Ice Core*. Springer, Berlin, 2018.

[8] S. H. Faria, G. M. Kremer, and K. Hutter. Creep and recrystallization of large polycrystalline masses. Part II: constitutive theory for crystalline media with transversely isotropic grains. *Proc. Roy. Soc. London A*, 462(2070):1699–1720, 2006.

[9] S. H. Faria, I. Weikusat, and N. Azuma. The microstructure of polar ice. Part II: state of the art. *J. Struct. Geol.*, 61:21–49, 2014.

[10] H. J. Frost and M. F. Ashby. *Deformation-mechanism Maps*. Pergamon, Oxford, 1982.
[11] O. Gagliardini. Comment on the papers 'creep and recrystallization of large polycrystalline masses' by faria and co-authors. *Proc. Roy. Soc. London A*, 464:289–291, 2008.

[12] G. Godert. A mesoscopic approach for modelling texture evolution of polar ice including recrystallization phenomena. *Ann. Glaciol.*, 37:23–28, 2003.

[13] F. J. Humphreys and M. Hatherly. *Recrystallization and Related Annealing Phenomena*. Pergamon, Oxford, 2nd edition, 2004.

[14] T. H. Jacka and J. Li. Flow rates and crystal orientation fabrics in compression of polycrystalline ice at low temperatures and stresses. In T. Hondoh, editor, *Physics of Ice Core Records*, pages 83–102. Hokkaido University Press, Sapporo, 2000.

[15] B. Kamb. Experimental recrystallization of ice under stress. In H. C. Heard, I. Y. Borg, N. L. Carter, and C. B. Raleigh, editors, *Flow and Fracture of Rocks*, number 16 in Geophysical Monograph, pages 211–241. American Geophysical Union, Washington, DC, 1972.

[16] S. Kipfstuhl, S. H. Faria, N. Azuma, J. Freitag, I. Hamann, P. Kaufmann, H. Miller, K. Weiler, and F. Wilhelms. Evidence of dynamic recrystallization in polar firn. *J. Geophys. Res.*, 114:B05204, 2009.

[17] S. Kipfstuhl, I. Hamann, A. Lambrecht, J. Freitag, S. H. Faria, D. Grigoriev, and N. Azuma. Microstructure mapping: A new method for imaging deformation-induced microstructural features of ice on the grain scale. *J. Glaciol.*, 52(178):398–406, 2006.

[18] J. W. Martin, R. D. Doherty, and B. Cantor. *Stability of Microstructure in Metallic Systems*. Cambridge University Press, Cambridge, 2nd edition, 1997.

[19] M. S. Paterson. A granular flow theory for the deformation of partially molten rock. *Tectonophysics*, 335:51–61, 2001.

[20] S. Piazolo, M. Montagnat, F. Grennerat, H. Moulinec, and J. Wheeler. Effect of local stress heterogeneities on dislocation fields: Examples from transient creep in polycrystalline ice. *Acta Materialia*, 90:303–309, 2015.

[21] J.-P. Poirier. *Creep of Crystals*. Cambridge University Press, Cambridge, 1985.

[22] D. H. Richards, S. S. Pegler, S. Piazolo, and O. G. Harlen. The evolution of ice fabrics: A continuum modelling approach validated against laboratory experiments. *Earth Planet. Sci. Lett.*, 556:116718, 2021.

[23] P. Skemer, I. Katayama, Z. Jiang, and S. ichiro Karato. The misorientation index: Development of a new method for calculating the strength of lattice-preferred orientation. *Tectonophysics*, 411(1):157–167, 2005.

[24] A. P. Sutton and R. W. Balluffi. *Interfaces in Crystalline Materials*. Clarendon, Oxford, 1995.

[25] H.-R. Wenk. Texture and anisotropy. In S. I. Karato and H.-R. Wenk, editors, *Plastic Deformation of Minerals and Rocks*, volume 51 of *Reviews in Mineralogy and Geochemistry*, pages 291–330. Mineralogical Society of America and Geochemical Society, Washington, DC, 2004.

[26] S. White. Geological significance of recovery and recrystallization processes in quartz. *Tectonophysics*, 39(1–3):143–170, 1977.

Fan, S., Hager, T.F., Prior, D.J., Cross, A.J., Goldsby, D.L., Qi, C., Negrini, M., Wheeler, J., 2020. Temperature and strain controls on ice deformation mechanisms: insights from the

microstructures of samples deformed to progressively higher strains at −10, −20 and −30°C. The Cryosphere 14, 3875–3905. https://doi.org/10.5194/tc-14-3875-2020

Lutz, F., Eccles, J., Prior, D.J., Craw, L., Fan, S., Hulbe, C., Forbes, M., Still, H., Pyne, A., Mandeno, D., 2020. Constraining Ice Shelf Anisotropy Using Shear Wave Splitting Measurements from Active-Source Borehole Seismics. J. Geophys. Res. Earth Surf. 125, e2020JF005707. https://doi.org/10.1029/2020JF005707

Skemer, P., Katayama, I., Jiang, Z., Karato, S., 2005. The misorientation index: Development of a new method for calculating the strength of lattice-preferred orientation. Tectonophysics 411, 157–167. https://doi.org/10.1016/j.tecto.2005.08.023

Review 3: Maurine Montagnat

Please find below the review of the new version of the paper. Since a lot of the comments I gave for the first review were not taken into account, I put at the end of the document the previous review I did, for the editor and the authors to do comparisons. The comments in concerned are underlined in yellow.

One of my concern on this paper is the feeling it gives me of a lack of clarity. Hypotheses are done with the model used here, and this is very fine for me, but many of them are not clearly stated. For instance:

-        by assuming the rotation rate of the orientation distribution with equation 5, the authors are doing the assumption of a Taylor-type of mechanical interactions between grains. Indeed, the only terms that act on the rotation rate is the strain-rate (and the vorticity that rotate the full distribution). Another way of doing, that would still be a rather crude parameterisation, would be to follow GilletChaulet et al. 2006 paper (eq 13) and put a stress component into the rotation rate. In this case, the mechanical hypothesis behind the orientation distribution rotation is in between a Taylor and a Sachs hypothesis.

We respectfully disagree with the reviewer that this model constitutes a Taylor type assumption.

We would like to remark here that just adding the macroscopic stress into the macroscopic fabric evolution equation does not entail any constraints on the stress or strain-rate experienced by individual grains (i.e. Taylor or Sachs). In fact, if Glens flow law is used, for the typical parameters used this parametrisation is equivalent to just multiplying the strain-rate tensor by a factor of 1.54. The point being that it is important to be mathematically rigorous with what assumptions are used, and not just say: strain-rate tensor in the equation means Taylor, stress tensor in the equation means Sachs.

We note that this is supported by the third reviewer of this article, quoted below in green.

The discussion about that in part 3.1, lines 230-245, is very deceiving... What is said lines 238-239 appears just wrong to me in a mechanical point of view. And the justification lines 240-245 is really astonishing! One can always parameterise any model to provide the result expected, it does not mean that the good "physics" is in the model!!! Please remove this sentence.

We refer here to the third reviewer's comments, in support of this section (in green):

**Lines 238–239:** That is correct indeed. At this point I have to digress to do something that I very rarely do—because it causes me great displeasure—which is to correct erroneous statements by another reviewer. In this particular case I feel obliged to do so, to rectify harmful and unfair criticism to the work under review. The unfair claims by the Reviewer are:

> *The model, that derives from previous works of Faria et al. (2006-I,II,III), assumes an homogeneous strain rate, meaning that each crystal is submitted to the same strain rate. This hypothesis, apparently not clearly stated in any of those works, has been shown by Gagliardini (2008) in its response to Faria et al. (2006) to correspond to a Taylor-type of approximation, meaning uniform strain.*

There are several errors in that statement. First, the Reviewer cites a comment by Gagliardini [11], but fails to cite the subsequent response [5] that proved the falsity of all Gagliardini's comments.

Second, it is true that the SpecCAF model is ultimately based upon the theory of Continuous Diversity developed by Faria et al. [3, 4, 8], but the Reviewer's claim that the theory of Continuous Diversity assumes a homogeneous strain rate for each grain (so-called "Taylor-type approximation" or "uniform strain") is clearly fallacious: it represents a complete disregard for the fundamental principles of continuum mechanics.

The theory of Continuous Diversity (CD) describes the large-scale ("macroscopic") flow of a glacier or ice sheet. As any other continuum theory, all fields and gradients in the CD theory are spatially defined on that large scale, which is many orders of magnitude larger than the grain scale. Therefore, just as the strain rate in fluid dynamics does not impose any constraint, hypothesis or approximation on the motion of individual molecules, the strain rate in the theory of Continuous Diversity does not impose any constraint, hypothesis or approximation on the deformation of individual grains: every grain is free to deform as inhomogeneously as needed. In plain mathematical terms, if d$x$ defines an

infinitesimal distance in the continuum (upon which all spatial gradients, including the strain rate, are defined) and $D$ is the average grain size, then d$x \gg D$.

[3] S. H. Faria. Creep and recrystallization of large polycrystalline masses. Part I: general continuum theory. *Proc. Roy. Soc. London A*, 462(2069):1493–1514, 2006.

[4] S. H. Faria. Creep and recrystallization of large polycrystalline masses. Part III: continuum theory of ice sheets. *Proc. Roy. Soc. London A*, 462(2073):2797–2816, 2006.

[5] S. H. Faria, K. Hutter, and G. M. Kremer. Reply to Gagliardini's comment on 'Creep and recrystallization of large polycrystalline masses' by Faria and co-authors. *Proc. Roy. Soc. London A*, 464(2099):2803–2809, 2008.

[8] S. H. Faria, G. M. Kremer, and K. Hutter. Creep and recrystallization of large polycrystalline masses. Part II: constitutive theory for crystalline media with transversely isotropic grains. *Proc. Roy. Soc. London A*, 462(2070):1699–1720, 2006.

[11] O. Gagliardini. Comment on the papers 'creep and recrystallization of large polycrystalline masses' by faria and co-authors. *Proc. Roy. Soc. London A*, 464:289–291, 2008.

And please let's assume the choice of the parameterisation as an "OK" hypothesis in order to simplify, especially since there exist no better way of doing so far.

From this choice depends the value of the parameters that have been tuned in Richard et al. 2021 on compression and simple shear cases, and this is fine! Providing it is clearly stated...

-        Considering the migration recrystallization mechanisms (in particular lines 264-265), the authors know that what drives them is more complicated that only the "cumulated shear strain"... What drives nucleation and grain boundary migration occurring during dynamic recrystallisation is related to the STORED strain energy, that is related to geometrically necessary dislocations, the ones that help compensating the strain incompatibilities between grains, and their density is not simply related to the cumulated shear strain (some areas that are deforming "easily" cumulate a lot of shear strain and very few geometrically necessary dislocations, so very little stored energy...). Once again, it makes sense, to my point of view, to use such a simplification in a model devoted to large scale flow modelling, but please mention it as an hypothesis and not as the truth!

Firstly, we would like to point out that we never refer to "accumulated shear strain" in the manuscript or in this location, we refer to the resolved shear strain-**rate** acting on the basal plane, such that this deformability acts to quantify "easy" and "hard" orientations. However, we appreciate that this may need further explanation. We have reworded this section to be clearer and to highlight that this is an approximation for the stored strain energy:

"For a given stretching tensor given by $\boldsymbol{D}$, for a basal plane with normal $\boldsymbol{n}$ this function represents the normalised strain-rate (or stretching) acting on the basal plane. Therefore, $\mathcal{D}*$ will be greater at orientations where it is easier to slip along the basal plane. Because ice deforms primarily by slip along the basal plane, this is a good approximation for the accumulation of deformation energy in a physical grain, which drives migration recrystallization."

Considering figure 2, for vorticity → infinity, there should be no fabric formed since the material experiences rigid body rotation only? Where could the girdle come from? What constrains the rotation within this girdle, under rigid-body rotation? What is the relative weight of the two parameterised recrystallisation regimes in this weak girdle? And how is it impacted by slight changes in the parameters? Is it robust?

We appreciate the reviewers' questions about the robustness and have added this figure to the parameter sensitivity supplement, and have referenced this in the text: Sensitivity analysis in the supplement reveals no change in the fabric pattern and only a small change in fabric strength, ranging between 1.11-1.23 for the min and max parameters.

Since we can only approximate infinity, exclusive rigid body rotation cannot be reached. Mathematically, the vorticity number is defined as $\mathcal{W} = \frac{O(W^2)}{O(1)}$ i.e. as we approach large vorticity numbers the magnitude of the deformation remains constant (it does not go to 0). Therefore there is always some deformation acting on the ice parcel, even if it may rotate many many times. We also remark in the text "The J-index of this fabric is 1.16, very close to completely isotropic (J=1). It is unlikely this weak girdle fabric would be distinguishable from an isotropic fabric in a physical sample, where the fabric is determined by sampling a limited number of grain orientations.

-        Once again, this study lies on parameterisation performed in laboratory conditions, therefore very far from the "real world" it aims at representing.  A sensibility study would therefore be necessary, to check, for instance, which of the rotation / migration recrystallization process is dominating and in which situation? Does that make sense with "real world" observations?
What happens when the parameters are shifted away from the linear fit? What is the impact on, for instance, the kinetic to steady-state?

This sensibility study is necessary to check the robustness of the modelling and therefore its ability to be predictive.

We introduced a supplement based on the reviewer similar comments in the previous review, which may have been missed by the reviewer. This contains 10 images exploring how all the figures in the article change based on the confidence intervals in the parameters used.

-        Line 411-415: the 2D assumption is strong, and, as already mentioned in my previous review, was shown to give "correct" results only is some specific parts of the ice cores. This part is not an explanation of the limitations of the 2D assumption and the impact it could have on the results, it is more the expression of the authors' opinion "is a good first step"... It may be, but please, explain us why and under which limitations.

We have added a sentence here explaining the limitations of the 2D assumption: "However, caution must be used when applying the conclusions of this paper to areas with highly three-dimensional deformations, such as curved ice streams or other areas where there is vertical and horizontal deformation."

More specific comments:

-        Line 21: what does "validated" means? When can we consider a model to be fully validated? In particular, this model has not been validated in other deformation regime than simple shear and compression, while you are going to use it in very different conditions.

Thank you for this helpful comment, we have changed this to: "The objective of the present paper is to use the fabric evolution model SpecCAF (Richards et al. 2021) to take a step away from the isolated conditions of irrotational deformation and simple shear where the model has been validated, and explore the continuous space of deformation regimes lying between these cases, and extrapolate beyond to deformation regimes more rotational than simple shear."

-        Part 2.2.1: VERY IMPORTANT!!! This paragraph contains explanations that are contrary to what is known for ice and recrystallization mechanisms. It should be re-written and bibliography may be more correctly used. For instance: **Chauve et al. 2017 do not show that non-basal slip is active! They just show that under some specific conditions, geometrically necessary non-basal dislocations can be observed. It has already been mentioned in my previous review, and since I am co-author of Chauve et al. 2017, it is very important for me that the authors correct it!**

We thank the reviewer for noticing this and have removed this.

The mechanisms described lines 42 and 43 are not stricto-sensus deformation mechanisms. Only crystal plasticity in the list is a deformation mechanism. Recrystallization is a process of accommodation that facilitates the deformation, but does not produce a deformation per-se...

For instance, during post-dynamic or static recrystallization there are a lot of microstructure modifications without any deformation produced. Please modify.

We thank the reviewer for pointing this out and have corrected this:

"T Both the intensity and pattern of the fabric produced is dependent on the conditions of deformation, which will influence the relative activity of different mechanisms. As ice deforms, the fabric evolves through dislocation glide along the basal plane, which causes c-axes to rotate (Steinemann, 1958; Hondoh, 2000), rigid-body rotation, which simply rotates grains around the rotation axis, and recrystallization processes, which rearrange the grain boundary network.

Line 42: Migration Recrystallization does not refer to grain boundary migration. Migration Recrystallization refers to a recrystallization associated with nucleation AND grain boundary migration, on a regime where grain boundary migration is fast. But nucleation does take place also. Above all, migration recrystallization refers to a mechanism that is driven by stored strain energy while grain boundary migration can occur driven by the reduction in grain boundary surface energy. It would be very important to clearly make the distinction and not mis-explain the migration recrystallization regime... Please see Humphreys and Haterly 2004 if necessary.

The reviewer is correct that there is at least to some researchers indeed a distinction between migration recrystallization and grain boundary migration. Within the metallurgical and geological communities the terms are used slightly differently. We have updated the text here "The first is migration recrystallization, which can include a combination of strain-induced grain boundary migration and nucleation (Dorothy et al. 1997)."

Grain boundary migration may be driven to reduce the stored strain energy and grain boundary surface energy of the whole system. In a system that is deforming by intracrystalline slip the driving force for grain boundary migration driven by stored energy reduction 1-2 orders of magnitude higher than grain boundary migration driven by grain boundary surface reduction (Gottstein and Shvindlerman, 1999). As a result of migration recrystallization grains with high stored strain energy will statistically be reduced in number (e.g. Czaplinska et al. 2017).

Line 45: the paper by Chauve et al. 2017 does not allow to say that non-basal dislocation activity is restricted to high temperature regime... This is just that the experiments presented in this paper are at high temperature. And once again, it just observes some non basal dislocations and no non-basal slip activity. So this sentence should be removed.

We realize that this was misleading and hence have removed this.

Lines 51-55: I am really puzzled to read this part, especially when dealing with ice! Recent results have shown (and some done by one of the co-authors, S. Piazolo), that strain heterogeneities in ice can not be resumed relatively to the main deviatoric stress (see Grennerat et al. 2012 for instance), and that strain distribution is very heterogeneous, with strain high in area of low stress, and the contrary... (see Piazolo et al. 2015, Montagnat et al. 2015, Chauve et al. 2017 Phil Trans). Similar observations exist also in other materials. So to say that "strain energy stored in a grain is directly related to the imposed strain" is somehow too vague and may be wrong... The reference given here, Gottstein and Shvindlerman 1999 is a book that I can not access to to verify. It could be stated as a working hypothesis, and justified, but not as the truth.

We respectfully note that, for the second time, the reviewer misquotes the manuscript in their response. We actually wrote "the strain energy stored in a grain is directly related to the imposed **stress**". However, we can see that this sentence may be misleading. The reviewer is worried about the distinction between far field stresses versus local stresses. In this section we refer to local stresses mainly, this has now been corrected in the section. We also add: "While the local stress axes are highly influenced by the environment around the respective grain and therefore stress and strain are heterogeneously distributed within a polycrystal (e.g. Piazolo et al. 2015, Grennerat et al. 2012), grains with c-axis oriented less favourably for slip relative to the far field stress axes will statistically have higher stored strain energy"

**Regarding highlighted previous review comments, August 2021: (*we assume the reviewer felt these were not addressed in our new version*)**

Reviewer: My point of view concerning these approximations made relatively to dynamic recrystallization is that they can be useful and justified in the simplified numerical modeling approach used in this work. Nevertheless, it has to be clearly mentioned that they ARE approximations, and their effects should be tested.

In response to this in the previous review, we added discussion of these assumptions in section 2.1.1 and 2.3. We hope that addressing the reviewers comments above, specifically regarding the formulation of $\mathcal{D}^*$, it is more clear what assumptions are used and what detail the model can represent.

The 2D approximation is also strong. It was shown by the Elmer-Ice community to be OK in the case of specific types of flow, like divides (where there is little divergence or convergence). Can it

holds for more complex situations such as fast ice streams? What effect could it produce on the

==fabric evolution? This should be justified and tested.==

Our previous response:

We are using the 2D approximation only as a stepping-stone to explore new fabric patterns and features beyond and intermediate to 'pure shear' and 'simple shear' (and to rotational deformations, which lie on the same spectrum). This is a deliberate choice for the scope of the present paper as a focus on a well-defined continuous space of fabrics indexed by a single parameter W (the vorticity number) and temperature T. In principle the model could be extended to more general deformations, but this is not the aim of this contribution, and would require more parameters to classify (e.g. an extra parameter representing the relative importance of vertical shear would be a natural next step). In this regard, two-dimensionality is not an approximation or limitation, but a focus to allow systematic and controlled exploration of a new research question as an initial step in the exploration of ice fabric evolutions. We appreciate that the title of the paper and the abstract may have suggested otherwise. Considering the comments by the reviewer we propose softening these statements, and to incorporate the words "two-dimensional" into the title.

We believe that extending the analysis to three-dimensions to test the 2D approximation is beyond the scope of this paper. Starting with a 2D approximation is a well established first step in science.

==In order to test the predictability of the model, it would be necessary to test how robust it is to variations in the parameters, and to the 2D approximation.==

Our previous response:

We have added this parameter sensitivity as a supplement. To do this we have taken the parameter fit from the inversion from both compression and simple shear in Richards et al. 2020 (rather than just simple shear as before) and calculated the 80% and 95% confidence intervals around this – shown in a new Fig 6. We have then reproduced the figures with the strongest possible fabric (maximum basal-slip deformation and migration recrystallization parameters, minimum rotation recrystallization parameter) and weakest possible fabric. This is then used to reproduce all the main figures in the results. We thank the reviewer for this suggestion.

In light of the parameter sensitivity investigation we decided to slightly modify our the focus of our results relating to the strain to reach steady-states. Rather than plotting steady-state time based on 90% of the convergence, which we found to be sensitive to parameter variations due to their effect on the tight criterion at which a steady state is reached, we instead report the halfway-strain to reach steady-state. This can be thought of as a half-life for fabric evolution, as explained in section 4.3.

**Please note: We believe the reviewer may have missed the parameter sensitivity supplement we included?**

By including the parameters sensitivity supplement, we have tested how robust the results are to variations in parameters. As discussed above, we believe testing the robustness of the 2D approximation (I.e. extending analysis into 3D) is beyond the scope of this paper.

---

## Author Response (AR3)

Dear Prof. Keegan

Thank you for accepting this paper for the cryosphere. We are now submitting the final paper and associated files for publication. We have also read through the manuscript and made some minor typo corrections, which we detail below:

Line 24: deleted superfluous *at*

Line 53: Changed reference Duval 1981 to Pimienta and Duval 1987

Line 110: deleted comma, corrected Figure to Fig.

Line 146: corrected *ten* to *Ten*

Line 195 and 200: corrected *orientation density function* to *orientation distribution function*

Line 376: corrected *it defines* to *it is defined*

Line 493: corrected *reached to* to *reached until*

Line 505: corrected *constraints such as knowledge the* to *constraints are available, such as knowledge that the*

Line 532: corrected *which is key processes* to *which is a key process*

Line 573: corrected *...of ice is an important...* to *… of ice. This is an important...*

Yours sincerely,

Daniel Richards

Sam Pegler

Sandra Piazolo